# Low-Rank Tensor Transitions (LoRT) for Transferable Tensor Regression

Andong Wang [1]    Yuning Qiu [1]    Zhong Jin [2]    Guoxu Zhou [3 4]    Qibin Zhao [1]

## Abstract

Tensor regression is a powerful tool for analyzing complex multi-dimensional data in fields such as neuroimaging and spatiotemporal analysis, but its effectiveness is often hindered by insufficient sample sizes. To overcome this limitation, we adopt a transfer learning strategy that leverages knowledge from related source tasks to improve performance in data-scarce target tasks. This approach, however, introduces additional challenges including model shifts, covariate shifts, and decentralized data management. We propose the Low-Rank Tensor Transitions (LoRT) framework, which incorporates a novel fusion regularizer and a two-step refinement to enable robust adaptation while preserving low-tubal-rank structure. To support decentralized scenarios, we extend LoRT to D-LoRT, a distributed variant that maintains statistical efficiency with minimal communication overhead. Theoretical analysis and experiments on tensor regression tasks, including compressed sensing and completion, validate the robustness and versatility of the proposed methods. These findings indicate the potential of LoRT as a robust method for tensor regression in settings with limited data and complex distributional structures.

## 1. Introduction

The increasing complexity of data in fields such as neuroimaging, chemometrics, and spatiotemporal analysis has amplified the need for expressive modeling tools. *Tensors*—multi-dimensional generalizations of matrices—have become essential for representing structured data (Kolda &

Bader, 2009). Building on this foundation, *tensor regression* extends classical regression to directly model relationships between tensor-valued inputs and outputs (Zhou et al., 2013; Sun & Li, 2017; Liu et al., 2021; Papadogeorgou et al., 2021; Taki et al., 2023; Wang et al., 2024a; Luo & Zhang, 2024; Zhou et al., 2024; Billio et al., 2024), preserving multi-way structure and unifying tasks like tensor completion and compressed sensing under a common framework (Wang et al., 2021; Lu et al., 2018). This modeling paradigm has enabled advances across neuroimaging (Zhou et al., 2013), social networks (Romera-Paredes et al., 2013), computer vision (Llosa-Vite & Maitra, 2022), phenotype prediction (Dos Santos et al., 2023), and climate science (Yu & Liu, 2016).

Despite notable progress, tensor regression still faces a fundamental limitation: *limited sample size* (*Challenge 1* in Fig. 1), especially when applied to complex structured tensor data (Liu et al., 2021; Zhou et al., 2013; Lock, 2018). Here, the sample size refers to the number of available input–output pairs, where each input is a high-dimensional tensor and the output is a response variable. For instance, in functional magnetic resonance imaging (fMRI) studies (Song & Lu, 2017), tensor predictors encode spatiotemporal brain activity, while responses reflect behavioral or clinical measures. Due to high data acquisition costs, the number of subjects—i.e., training samples—is often small. Similarly, in tensor compressed sensing and completion, the sample size is defined by the number of observed entries, which is typically much smaller than the total number of tensor elements (Wang et al., 2021). These scenarios create a severe underdetermined problem, as the number of model parameters increases rapidly with tensor order and size. This leads to a central challenge:

*How can tensor regression models remain effective under severe sample scarcity?*

To address this challenge, *transfer learning* provides a promising strategy by enabling models to leverage knowledge from related tasks or datasets (Zhuang et al., 2020; Sun et al., 2019; He et al., 2024a). Specifically, information can be transferred from "source tasks" with abundant or complementary data to the "target task" with insufficient samples, offering the potential to mitigate data scarcity and enhance model performance.

[1]RIKEN AIP, Tokyo, Japan. [2] Department of Computer, China University of Petroleum – Beijing at Karamay, Karamay, China. [3]School of Automation, Guangdong University of Technology, Guangzhou, China. [4]Key Laboratory of Intelligent Detection and the Internet of Things in Manufacturing, Ministry of Education, Guangdong University of Technology, Guangzhou, China. Correspondence to: Qibin Zhao <qibin.zhao@riken.jp>.

*Proceedings of the $42^{nd}$ International Conference on Machine Learning*, Vancouver, Canada. PMLR 267, 2025. Copyright 2025 by the author(s).

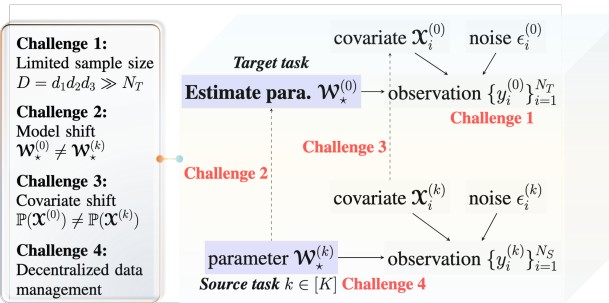

**Challenge 1:**
Limited sample size
$D = d_1 d_2 d_3 \gg N_T$

**Challenge 2:**
Model shift
$\mathcal{W}_\star^{(0)} \neq \mathcal{W}_\star^{(k)}$

**Challenge 3:**
Covariate shift
$\mathbb{P}(\mathcal{X}^{(0)}) \neq \mathbb{P}(\mathcal{X}^{(k)})$

**Challenge 4:**
Decentralized data
management

Figure 1: Illustration of the transferable tensor regression problem, featuring four key challenges: Limited sample size (Challenge 1), model shifts between tasks (Challenge 2), covariate shifts (Challenge 3), and decentralized data management (Challenge 4). The objective is to estimate the target task parameter $\mathcal{W}_\star^{(0)}$ by leveraging information from $K$ source tasks while addressing these challenges, as detailed in Section 3.

However, in the context of tensor regression, transfer learning also introduces additional challenges due to the inherent structure of tensor data and the presence of *distribution shifts* between source and target samples (He et al., 2024a;b; Duan & Wang, 2023). These shifts manifest as *model shifts* (*Challenge 2* in Fig. 1), where the relationship between tensor predictors and responses changes across domains, and *covariate shifts* (*Challenge 3* in Fig. 1), which occur when the distribution of tensor predictors differs. For instance, in tensor regression tasks like fMRI analysis, model shifts might reflect differing brain activity-response patterns across populations, while covariate shifts could arise from variations in imaging conditions or demographic factors. Similarly, in tensor sensing and completion, covariate shifts might represent differences in the sampling pattern of observations across the source and target tasks, and model shifts could indicate changes in the underlying low-rank structures. These distribution shifts can significantly degrade the performance of transfer learning models, complicating the already challenging task of tensor regression.

Beyond distribution shifts, another critical challenge in applying transfer learning to tensor regression is *decentralized data management* (*Challenge 4* in Fig. 1) (He et al., 2024a). Data often cannot be centralized due to privacy concerns, logistical constraints, or its large scale. This necessitates developing efficient strategies for implementing transfer learning in distributed settings, with a focus on minimizing communication overhead.

To address the above-mentioned challenges (summarized in Fig. 1), this paper proposes the Low-Rank Tensor Transitions (LoRT) framework for transferable tensor learning with the following contributions:

- **A novel framework for transferable tensor learning:** LoRT introduces a new framework that enhances transfer learning in tensor regression by leveraging low-tubal-rank structures with a novel fusion regularizer designed to identify shared patterns across tasks. By employing a two-step estimation strategy, LoRT first integrates information from both source and target tasks, addressing the challenge of limited sample size (Challenge 1). The refinement step further ensures accurate estimation of target tensor parameters, mitigating the impact of model shifts (Challenge 2).

- **Theoretical insights:** This paper provides a comprehensive theoretical analysis of the LoRT framework, deriving non-asymptotic error bounds that establish the conditions under which LoRT can effectively leverage information from source tasks. These results demonstrate LoRT's robustness to model and covariate shifts (Challenges 2 & 3).

- **Decentralized data management:** To tackle the challenge of decentralized data management (Challenges 4), the LoRT framework is extended to a distributed setting with the development of D-LoRT. This distributed variant retains the statistical efficiency of the centralized LoRT while significantly reducing communication overhead, making it particularly suitable for large-scale, privacy-sensitive, or distributed environments.

To the best of our knowledge, this work is the first to systematically address the four key challenges of transfer tensor regression through the innovative LoRT framework. The structure of the paper is as follows: we start with notations and preliminaries in Section 2, followed by the problem formulation in Section 3. The LoRT framework and its distributed extension, D-LoRT, are then introduced in Section 4, along with their implementation and theoretical analysis. We proceed with empirical results in Section 5 to validate the proposed methods and conclude with a discussion of limitations in Section 6. Related work, proofs and experimental details can be found in the Appendix.

## 2. Notations and Preliminaries

**Notations.** For any positive integer $d$, let $[d] := \{1, \cdots, d\}$. We use lowercase bold letters (e.g., $\mathbf{a}$) for vectors, a uppercase bold letters (e.g., $\mathbf{A}$) for matrices, underlined uppercase letters for 3-way tensor (e.g., $\mathcal{A}$). We use $c$, and its variants like $c_1, C$, etc., to denote positive constants whose value may vary between lines. For 3-way tensors of size $d_1 \times d_2 \times d_3$, we assume $d_1 \geq d_2$ without loss of generality. In this paper, "*with high probability (w.h.p.)*" means with probability at least $1 - c_1 e^{-c_2 N_T} - c_3 e^{-c_4 \log(d_1 d_3 + d_2 d_3)}$, where $\{c_i\}_{i=1}^4$ are constants and $N_T$ is the sample size of the target task.

For any matrix $\mathbf{A}$, its spectral norm $\|\mathbf{A}\|_{\mathrm{sp}}$ and nuclear norm $\|\mathbf{A}\|_*$ are defined as its maximum and sum of its singular values, respectively. Given a tensor $\mathcal{A}$, its $\ell_p$-norm is defined as $\|\mathcal{A}\|_p := \|\mathrm{vec}(\mathcal{A})\|_p$, and its F-norm is defined as $\|\mathcal{A}\|_{\mathrm{F}} := \|\mathrm{vec}(\mathcal{A})\|_2$, where $\mathrm{vec}(\cdot)$ denotes the vectorization operation of a tensor (Kolda & Bader, 2009). The inner product between two tensors $\mathcal{A}$ and $\mathcal{B}$ is defined as $\langle \mathcal{A}, \mathcal{B} \rangle := \mathrm{vec}(\mathcal{A})^\top \mathrm{vec}(\mathcal{B})$. For $\mathcal{A} \in \mathbb{R}^{d_1 \times d_2 \times d_3}$, we use $\mathcal{A}_{:,:,i}$ to denote its $i$-th frontal slice.

**The t-SVD Framework.** The tensor singular value decomposition (t-SVD) framework is built upon the t-product operation under an invertible linear transform $M$ (Kernfeld et al., 2015). This approach stems from the observation that certain linear transformations can enhance the low-rank characteristics of tensors, effectively exploiting intrinsic correlations within the data (Zhang & Ng, 2021; Wang et al., 2021). Recent research has focused on utilizing orthogonal matrices to define the transform $M$, due to their advantageous properties (Lu, 2021; Wang et al., 2023a). This convention is also adopted in this paper. Formally, given an *orthogonal matrix* $\mathbf{M} \in \mathbb{R}^{d_3 \times d_3}$, we define the associated linear transform $M(\cdot)$ and its inverse $M^{-1}(\cdot)$ on any tensor $\mathcal{T} \in \mathbb{R}^{d_1 \times d_2 \times d_3}$ as follows:

$$M(\mathcal{T}) := \mathcal{T} \times_3 \mathbf{M}, \quad \text{and} \quad M^{-1}(\mathcal{T}) := \mathcal{T} \times_3 \mathbf{M}^{-1} \quad (1)$$

where $\times_3$ denotes the mode-3 tensor-matrix product (Kernfeld et al., 2015).

Building upon this framework, we can now introduce the fundamental concepts of t-SVD:

**Definition 2.1** (t-product (Kernfeld et al., 2015)). The t-product of any $\mathcal{A} \in \mathbb{R}^{d_1 \times d_2 \times d_3}$ and $\mathcal{B} \in \mathbb{R}^{d_2 \times d_4 \times d_3}$ under the transform $M$ in Eq. (1) is denoted and defined as $\mathcal{A} *_M \mathcal{B} = \mathcal{C} \in \mathbb{R}^{d_1 \times d_4 \times d_3}$ such that

$$M(\mathcal{C}) = M(\mathcal{A}) \odot M(\mathcal{B})$$

in the transformed domain, where $\odot$ denotes the tensor frontal-slice-wise product.

Following the definitions of t-transpose, t-identity tensor, t-orthogonal tensor, and f-diagonal tensor given[1] by (Kernfeld et al., 2015), we can now introduce the t-SVD:

**Definition 2.2** (t-SVD, tensor tubal rank (Kernfeld et al., 2015)). The tensor Singular Value Decomposition (t-SVD) of any $\mathcal{T} \in \mathbb{R}^{d_1 \times d_2 \times d_3}$ under the invertible linear transform $M$ in Eq. (1) is given as follows:

$$\mathcal{T} = \mathcal{U} *_M \mathcal{S} *_M \mathcal{V}^\top \quad (2)$$

where $\mathcal{U} \in \mathbb{R}^{d_1 \times d_1 \times d_3}$ and $\mathcal{V} \in \mathbb{R}^{d_2 \times d_2 \times d_3}$ are t-orthogonal, and $\mathcal{S} \in \mathbb{R}^{d_1 \times d_2 \times d_3}$ is f-diagonal.

The tubal rank of $\mathcal{T}$ is defined as the number of non-zero tubes in tensor $\mathcal{S}$ in the t-SVD in Eq. (2), i.e.,

$$r_{\mathrm{t}}(\mathcal{T}) := \#\{i \mid \mathcal{S}_{i,i,:} \neq \mathbf{0}, i \leq \min\{d_1, d_2\}\}.$$

To further quantify the low-rank structure of tensors in the transformed domain, we introduce the concepts of tensor tubal nuclear norm and tensor t-spectral norm:

**Definition 2.3** (Tensor tubal nuclear norm, tensor t-spectral norm (Lu et al., 2019b)). The tubal nuclear norm (TNN) and tensor spectral norm of any tensor $\mathcal{T} \in \mathbb{R}^{d_1 \times d_2 \times d_3}$ under any $M$ in Eq. (1) are defined as the sum of the nuclear norms and the maximum of the spectral norms of the frontal slices of $M(\mathcal{T})$, respectively, i.e.,

$$\|\mathcal{T}\|_\star := \sum_{i=1}^{d_3} \|M(\mathcal{T})_{:,:,i}\|_*, \ \|\mathcal{T}\|_{\mathrm{tsp}} := \max_{i \in [d_3]} \|M(\mathcal{T})_{:,:,i}\|_{\mathrm{sp}}.$$

The significance of tubal rank and TNN lies in their ability to capture and exploit the low-rank structure of tensors in the transformed domain. Tubal rank quantifies the degree of low-rankness by counting non-zero tubes in the t-SVD representation, while the tubal nuclear norm serves as its convex surrogate. These concepts have found wide-ranging applications in tensor estimation tasks, enabling efficient representation and processing of multi-dimensional data (Lu, 2021; Zhang & Ng, 2021; Hou et al., 2021).

## 3. Problem Formulation

This section proposes a problem setting that aligns with and addresses the four challenges illustrated in Fig. 1. We first introduce the target task as follows:

**Target Task.** For the target task[2], we observe $N_T$ input-output pairs $(\mathcal{X}_i^{(0)}, y_i^{(0)})$ generated from:

$$y_i^{(0)} = \langle \mathcal{X}_i^{(0)}, \mathcal{W}_\star^{(0)} \rangle + \epsilon_i^{(0)}, \quad i \in [N_T] \quad (3)$$

where $\mathcal{W}_\star^{(0)} \in \mathbb{R}^{d_1 \times d_2 \times d_3}$ is our primary parameter of interest and $\epsilon_i^{(0)}$ is the observation noise. The primary goal is to estimate the target parameter tensor $\mathcal{W}_\star^{(0)}$ from noisy observations, as commonly encountered in tasks like tensor compressed sensing and completion (Wang et al., 2021).

A key difficulty in this setting is the *limited sample size* (Challenge 1), where the total number of parameters $D = d_1 d_2 d_3$ far exceeds the available target samples $N_T$, making reliable estimation of $\mathcal{W}_\star^{(0)}$ inherently challenging.

Motivated by the empirical and theoretical success of low-rank models in tensor completion (Wang et al., 2021; Lu

---

[1]Due to space constraints, please refer to the appendix for detailed definitions.

[2]Following He et al. (2024a), we use the superscript 0 to denote the target task, i.e., task 0.

et al., 2019b; Liu et al., 2020), we assume that $\boldsymbol{\mathcal{W}}_\star^{(0)}$ has a low tubal rank, i.e.,

$$r := r_t(\boldsymbol{\mathcal{W}}_\star^{(0)}) \ll d_2. \tag{4}$$

This assumption helps reduce the effective number of parameters and enables statistically efficient estimation under limited sample sizes. It is also supported by prior studies showing the effectiveness of t-SVD-based low-rank structures in capturing intrinsic correlations in tensor data (Zhang & Ng, 2021; Qiu et al., 2022a; Lu, 2021; Wang et al., 2023a; 2021).

To further address the limited sample size issue, we adopt transfer learning, utilizing data from related source tasks to enhance target task performance.

**Source Tasks.** Similarly, each of the $K$ source tasks, potentially in *decentralized* scenarios, has $N_S$ samples:

$$y_i^{(k)} = \langle \boldsymbol{\mathcal{X}}_i^{(k)}, \boldsymbol{\mathcal{W}}_\star^{(k)} \rangle + \epsilon_i^{(k)}, \;\; i \in [N_S], k \in [K] \tag{5}$$

where $\boldsymbol{\mathcal{W}}_\star^{(k)} \in \mathbb{R}^{d_1 \times d_2 \times d_3}$ is the $k$-th source task parameter.

While transfer learning offers a promising solution to mitigate the limited sample size issue, it also brings additional challenges. Notably, differences between source and target tasks—characterized by *model shift and covariate shift* (Challenges 2 and 3 in Fig. 1)—become significant hurdles.

First, model shift is quantified by the difference between each source model and the target model, i.e., $\boldsymbol{\mathcal{W}}_\star^{(k)} - \boldsymbol{\mathcal{W}}_\star^{(0)}$ for $k \in [K]$. To formalize the notion of informative source tasks, we introduce the *parameter space for all the tasks*:

$$\mathbb{W}(r, \mathbf{h}) \tag{6}$$
$$:= \left\{ (\boldsymbol{\mathcal{W}}^{(k)})_{k=0}^{K} : r_t(\boldsymbol{\mathcal{W}}^{(0)}) \leq r, \|\boldsymbol{\mathcal{W}}^{(k)} - \boldsymbol{\mathcal{W}}^{(0)}\|_\star \leq h_k \right\}$$

with $\mathbf{h} := (h_1, \ldots, h_K)^\top$. Here, $h_k \geq 0$ quantifies the informative level of the $k$-th source task. In $\mathbb{W}(r, \mathbf{h})$, we posit that the model shift between source and target tasks follows an *approximately low-rank structure* imposed by TNN to capture shared patterns across tasks, facilitating effective knowledge transfer.

*Remark* 3.1 (Rationale of approximately low-rank model shift). The approximately low-rank shift in our parameter space $\mathbb{W}(r, \mathbf{h})$ is motivated by empirical findings in Low-Rank Adaptation (LoRA) (Hu et al., 2022) and its extensions (Agiza et al., 2024; Wang et al., 2024b). In LoRA, it is assumed that the target model parameters differ from those of the pre-trained or source model by a low-rank increment. This assumption aligns conceptually with our approximately low-rank constraint in $\mathbb{W}(r, \mathbf{h})$, which captures shared patterns while accommodating task-specific differences.

Then, to account for covariate shift, we make the following assumption on the distribution of covariates $\boldsymbol{\mathcal{X}}_i^{(k)}$:

**Assumption 3.2** (Gaussian designs). For any $0 \leq k \leq K$, the entries of the covariate $\boldsymbol{\mathcal{X}}_i^{(k)} \in \mathbb{R}^{d_1 \times d_2 \times d_3}$ for the $k$-th task are *i.i.d.* drawn from $\mathcal{N}(0, \sigma_k^2)$. Furthermore, there exists a universal constant $c_x$ such that $c_x^{-1} \leq \min_{0 \leq k \leq K}\{\sigma_k\} \leq \max_{0 \leq k \leq K}\{\sigma_k\} \leq c_x$.

This assumption allows for different covariate distributions across tasks while ensuring well-behaved tail properties. While Gaussian design is assumed for simplicity, Assumption 3.2 can be extended to sub-Gaussian designs under mild conditions. Finally, for the observation noise, we assume:

**Assumption 3.3** (Gaussian random noises). For all $0 \leq k \leq K$, the noises $\epsilon_i^{(k)}$ are independent Gaussian with zero mean and variance uniformly upper bounded by a universal constant $c_\epsilon^2$, and are independent of the covariates $\boldsymbol{\mathcal{X}}_i^{(k)}$.

**Problem Objective.** Given the task models, we may ask:

- **Q1**: *How can we estimate the target parameter $\boldsymbol{\mathcal{W}}_\star^{(0)}$ using both target and source samples while facing Challenges 1-3 in Fig. 1?*

- **Q2**: *How can we further address the decentralized data management challenge (Challenge 4 in Fig. 1) given potentially decentralized source tasks?*

Answering this question is the primary focus of our subsequent sections, where we introduce our proposed method, LoRT, and its distributed variant, D-LoRT.

## 4. Low-Rank Tensor Transitions

We introduce LoRT (Low-Rank Tensor Transitions) and its distributed variant, D-LoRT, to address **Q1** and **Q2**. An illustration of both methods is provided in Fig. 2.

### 4.1. LoRT for Centralized Tensor Transfer Learning

Our method is motivated by the recent *TransFusion* framework (He et al., 2024a), which studies fusion-based transfer learning for sparse vector regression under covariate shift. In *TransFusion*, a shared low-complexity component is jointly estimated across tasks using $\ell_1$-based fusion regularization, enabling robustness to distributional changes. LoRT extends this idea to the tensor regression setting, where both the predictors and parameters are tensors. This extension poses new challenges, such as modeling low-rank structure along multiple tensor modes, handling tensor-specific regularization, and ensuring identifiability under limited samples.

To address these, LoRT introduces a novel fusion regularizer (Eq. (8)) based on the TNN, which promotes low tubal-rank structure in both the target parameter and its differences with the source parameters to leverage the parameter structures

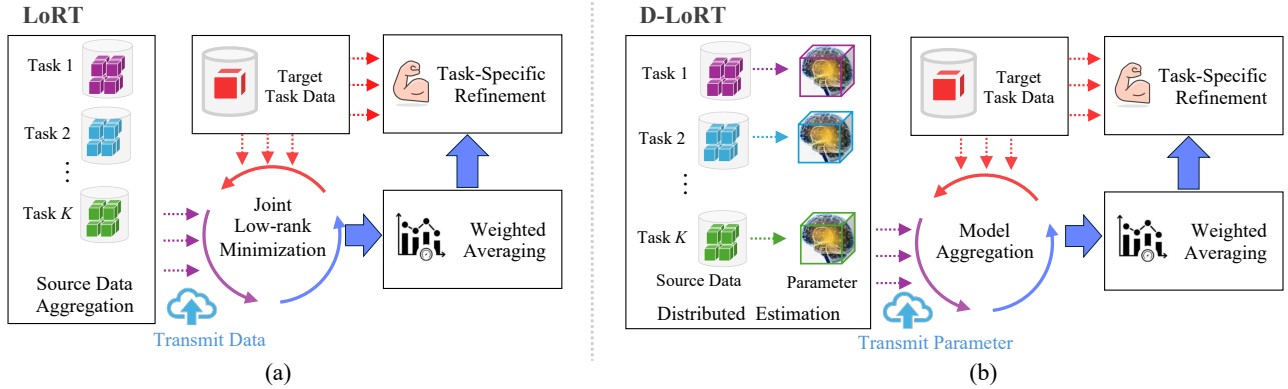

Figure 2: Illustration of the proposed LoRT and D-LoRT frameworks: (a) LoRT: A centralized method that combines joint low-rank learning and target-specific refinement to address challenges such as limited sample sizes, model shifts, and covariate shifts in tensor regression. (b) D-LoRT: A distributed extension of LoRT that handles decentralized data by aggregating locally estimated models from source tasks, ensuring statistical and communication overhead.

in $\mathbb{W}(r, \mathbf{h})$. More specifically, LoRT adopts a two-step estimation framework:

- a *joint learning* step that integrates source and target data under low-rank constraints to alleviate sample scarcity, and

- a *target-specific refinement* step that adapts the solution to the target domain and mitigates negative transfer caused by heterogeneity.

This design enables LoRT to systematically address the key challenges of transferable tensor regression, including limited target data, model shift, and covariate shift.

**Step 1: Joint Low-rank Learning.** It consists of three sub-steps:

S1 *Data Aggregation:* We collect $N_T$ input-output pairs for the target task and $N_S$ pairs for each of the $K$ source tasks, resulting in a total sample size of $N = N_T + K N_S$. This aggregation enables the integration of information from both source and target domains.

S2 *Joint Low-Rank Estimation:* Given the structure of the parameter space $\mathbb{W}(r, \mathbf{h})$ in Eq. (6), our goal is to estimate the ground truth parameters $\vec{\mathcal{W}}_\star = (\mathcal{W}_\star^{(0)}, \mathcal{W}_\star^{(1)}, \ldots, \mathcal{W}_\star^{(K)})$ by minimizing a regularized empirical loss across all tasks. Specifically, we solve the following optimization problem:

$$\min_{\vec{\mathcal{W}}} \frac{1}{2N} \sum_{k=0}^{K} \sum_{i=1}^{N_k} \left( y_i^{(k)} - \langle \mathcal{W}^{(k)}, \mathcal{X}_i^{(k)} \rangle \right)^2 + \lambda_0 \, \mathcal{R}(\vec{\mathcal{W}}),$$
$$(7)$$

where $N_k = N_T$ for the target task ($k = 0$) and $N_k = N_S$ for each source task ($k \in [K]$). Here,

$\mathcal{X}_i^{(k)} \in \mathbb{R}^{d_1 \times d_2 \times d_3}$ and $y_i^{(k)} \in \mathbb{R}$ denote the $i$-th input-output pair from task $k$, and $\mathcal{W}^{(k)}$ is the corresponding task-specific parameter. The optimization variable in Problem (7) is $\vec{\mathcal{W}} = (\mathcal{W}^{(0)}, \mathcal{W}^{(1)}, \ldots, \mathcal{W}^{(K)})$, and $\widehat{\vec{\mathcal{W}}} = (\hat{\mathcal{W}}^{(0)}, \hat{\mathcal{W}}^{(1)}, \ldots, \hat{\mathcal{W}}^{(K)})$ denotes the corresponding solution.

The regularizer $\mathcal{R}(\vec{\mathcal{W}})$ enforces both low-rank structure in the target and structured similarity across tasks:

$$\mathcal{R}(\vec{\mathcal{W}}) := \|\mathcal{W}^{(0)}\|_\star + \sum_{k=1}^{K} a_k \|\mathcal{W}^{(0)} - \mathcal{W}^{(k)}\|_\star, \quad (8)$$

where $\|\cdot\|_\star$ denotes the TNN. The hyperparameter $\lambda_0$ controls overall regularization strength, while $\{a_k\}_{k=1}^{K}$ modulate the contribution of each source task in the fusion.

Problem (7) integrates an average squared loss term to ensure a good fit across all tasks, and a fusion regularization term $\mathcal{R}(\vec{\mathcal{W}})$ with two primary functions. First, $\|\mathcal{W}^{(0)}\|_\star$ encourages a low-rank structure in the target task parameters, addressing the high-dimensionality challenge. Second, $\sum_{k=1}^{K} a_k \|\mathcal{W}^{(0)} - \mathcal{W}^{(k)}\|_\star$ captures the contrast between the target and source tasks, enabling effective knowledge transfer while accounting for potential model shifts. The use of the TNN is crucial as it effectively captures low-rank structures in tensor parameters and model shifts.

S3 *Weighted Averaging:* Define the first-step estimator:

$$\hat{\mathcal{W}}^{\mathrm{a}} := N^{-1} \left( N_S \sum_{k=1}^{K} \hat{\mathcal{W}}^{(k)} + N_T \hat{\mathcal{W}}^{(0)} \right). \quad (9)$$

This averaging step combines information from all tasks, weighting each task's contribution by its sample size. It potentially reduces variance and improves overall accuracy by leveraging the collective knowledge from both source and target domains. The resulting $\hat{\mathcal{W}}^{\mathrm{a}}$ serves as a robust initial estimate that balances the influence of all tasks.

**Step 2: Target-Specific Refinement.** While joint low-rank learning leverages information from all tasks, it may introduce bias due to differences between source and target tasks. To mitigate this, we introduce a refinement step focused solely on the target task:

$$\hat{\mathcal{W}}^{(0)}_{\mathrm{lort}} = \hat{\mathcal{W}}^{\mathrm{a}} + \hat{\mathcal{C}} \qquad (10)$$

with correction $\hat{\mathcal{C}}$ computed by

$$\hat{\mathcal{C}} \in \underset{\mathcal{C}}{\arg\min} \left\{ \frac{1}{2N_T} \sum_{i=1}^{N_T} (y_i^{(0)} - \langle \mathcal{X}_i^{(0)}, \hat{\mathcal{W}}^{\mathrm{a}} + \mathcal{C} \rangle)^2 + \tilde{\lambda} \|\mathcal{C}\|_\star \right\}.$$

This refinement step can be interpreted as a denoising procedure. It allows us to fine-tune our initial estimator using only the target sample, potentially mitigating any negative transfer effects from the source tasks. The use of the TNN regularization in this step ensures that the refinement maintains the low-rank structure of the estimator.

**Parameter Selection.** The performance of LoRT depends on the appropriate selection of tuning parameters. Our analysis suggests the following parameter selection strategy:

$$\lambda_0 = c_0 \xi_1, \quad a_k = c_1 \xi_2, \quad \tilde{\lambda} = c_2 \sqrt{d_1/N_T}. \qquad (11)$$

Here, $\xi_1$ and $\xi_2$ are adaptive factors for regularization and source task weighting, respectively, defined as: $\xi_1 = \sqrt{d_1/N}\mathbb{I}_{\mathbf{A}} + \sqrt{d_1/N_S}\mathbb{I}_{\mathbf{A}^c}$ and $\xi_2 = \sqrt{N_S/N}\mathbb{I}_{\mathbf{A}} + (N_S/N)\mathbb{I}_{\mathbf{A}^c}$, where $\mathbb{I}_{\mathbf{A}}$ and $\mathbb{I}_{\mathbf{A}^c}$ are indicator functions for the event $\mathbf{A}$ and its complement. The event $\mathbf{A}$ is defined as $\mathbf{A} := \left\{ rd_1d_3/N_S \geq \bar{h}\sqrt{d_1/N_T} \right\}$, with $\bar{h} := N^{-1}N_S \sum_{k=1}^{K} h_k$ representing average task *heterogeneity* (i.e., the differences between the model parameters of the source tasks and the target task). This strategy balances source and target influences, enabling LoRT to leverage informative source tasks or rely on target data when source tasks are less relevant.

### 4.2. Theoretical Analysis of LoRT

We provide theoretical guarantees for LoRT, demonstrating its effectiveness in addressing the challenges of high-dimensional tensor regression in transfer learning settings.

We first introduce a key concept that quantifies the alignment of source tasks with the target task:

**Definition 4.1** (Source task alignment). Given the ground truth parameter $\vec{\mathcal{W}}_\star \in \mathbb{W}(r, \mathbf{h})$, we quantify the alignment

across source tasks with the metric

$$N^{-1} \left\| N_S \sum_{k=1}^{K} (\mathcal{W}_\star^{(k)} - \mathcal{W}_\star^{(0)}) \right\|_\star \leq \delta_\star.$$

A small $\delta_\star$ indicates that the source tasks $\{\mathcal{W}_\star^{(k)}\}_{k=1}^{K}$ are well-aligned with the target task $\mathcal{W}_\star^{(0)}$, facilitating effective knowledge transfer. We now present our main theoretical results, starting with the convergence rate for the one-step estimator. Recall that we have assumed $d_1 \leq d_2$ without loss of generality.

**Theorem 4.2** (One-step LoRT). *Under Assumptions 3.2 and 3.3, if $N_S \gg rd_1d_3$, then by choosing $\lambda_0 = c_0\sqrt{d_1/N}$ and $a_k = c_1\sqrt{N_S/N}$ for some constant $c_0$ and $c_1$, we have*

$$\|\hat{\mathcal{W}}^{\mathrm{a}} - \mathcal{W}_\star^{(0)}\|_{\mathrm{F}}^2 \lesssim \frac{rd_1d_3}{N} + (1 + v_n)\bar{h}\sqrt{\frac{d_1}{N_S}} + \delta_\star^2 \quad (12)$$

*w.h.p., where $v_n := \bar{h}K\sqrt{d_1/N_S}$, quantifying the impact of task heterogeneity on the estimation error.*

This result reveals how the estimation error depends on the sample sizes, task differences, and source task alignment. The first term represents the standard error rate for estimating a low-rank tensor, while the latter terms capture the impact of transfer learning.

For the refined estimator, we have the following guarantee:

**Theorem 4.3** (Refined LoRT). *Under the assumptions of Theorem 4.2, if $N_T \gtrsim rd_1d_3$, $N_S \gtrsim K^2rd_1d_3$ and $\bar{h}\sqrt{d_1/N_T} + Krd_1d_3/N_S = o(1)$, then by choosing parameters as specified in Eq. (11), the solution of the two-step method satisfies w.h.p. that*

$$\|\hat{\mathcal{W}}^{(0)}_{\mathrm{lort}} - \mathcal{W}_\star^{(0)}\|_{\mathrm{F}}^2 \lesssim \frac{rd_1d_3}{N} + \bar{h}\sqrt{\frac{d_1}{N_T}}. \qquad (13)$$

This theorem demonstrates that the two-step method can effectively mitigate the impact of non-diverse source tasks, as the dependence on $\delta_\star$ in Eq. (12) is removed.

These results underscore key strengths of LoRT, effectively addressing **Q1**. When source tasks are both informative (small $\bar{h}$) and closely aligned with the target task (small $\delta_\star$), LoRT achieves significantly *smaller errors* compared to the $O(rd_1d_3N_T^{-1})$ bound of t-SVD-based tensor sensing (Wang et al., 2021), which depends solely on the target task data. The two-step refinement design further ensures *robustness* by protecting against negative transfer, maintaining reliable performance even when source tasks are uninformative or poorly aligned. Additionally, the dependence on $rd_1d_3$ rather than the full dimension $d_1d_2d_3$ demonstrates *efficiency* in handling of high-dimensional data through low-rank structure.

### 4.3. D-LoRT: A Distributed Variant of LoRT

While LoRT effectively addresses **Q1**, it assumes centralized data access, thus fails to address the decentralization challenge in **Q2**. To overcome this, we introduce D-LoRT, a distributed variant of LoRT.

**Algorithm Overview.** D-LoRT adapts the joint learning step of LoRT to operate on summary statistics from source tasks, proceeding as follows:

S1 *Distributed Estimation:* Each source node $k$ computes a local estimator $\tilde{\mathcal{W}}^{(k)}$ using its data $(\mathcal{X}_i^{(k)}, y_i^{(k)})_{i=1}^{N_S}$ in a distributed manner.

S2 *Model Aggregation:* Source nodes transmit their local estimators to the target node and the target node computes $\widehat{\mathcal{W}}_{\mathsf{d}}$ via the following minimization problem:

$$\widehat{\mathcal{W}}_{\mathsf{d}} \in \arg\min_{\vec{\mathcal{W}}} \frac{N_S}{2N} \sum_{k=1}^{K} \|(\tilde{\mathcal{W}}^{(k)} - \mathcal{W}^{(k)})\|_{\mathrm{F}}^2 \qquad (14)$$
$$+ \lambda_0 \mathcal{R}(\vec{\mathcal{W}}) + \frac{1}{2N} \sum_{i=1}^{N_T} (y_i^{(0)} - \langle \mathcal{X}_i^{(0)}, \mathcal{W}^{(0)} \rangle)^2.$$

S3 *Weighted Averaging:* The target node computes the averaging parameter $\hat{\mathcal{W}}_{\mathsf{d}}^{\mathsf{a}}$ based on $\widehat{\mathcal{W}}_{\mathsf{d}}$ using Eq. (9).

S4 *Task-Specific Refinement:* The target node refines $\hat{\mathcal{W}}_{\mathsf{d}}^{\mathsf{a}}$ locally via Eq. (10) to obtain the estimator $\hat{\mathcal{W}}_{\mathsf{dlort}}^{(0)}$.

**Theoretical Guarantees for D-LoRT.** We establish the following statistical guarantees for D-LoRT. Recall that we have assumed $d_1 \leq d_2$ without loss of generality.

**Theorem 4.4** (One-step D-LoRT). *Suppose Assumptions 3.2 and 3.3 hold. For each source task, the debiased estimator is defined in Eq. (E.2) and satisfies standard approximation conditions. Assume $N_S \gg K r^2 d_1 d_3^2$, $N_S \gtrsim (\bar{h}^2 \vee K^2) r d_1 d_3$, and $h_k \asymp \bar{h} = O(1)$. Then, for some chosen parameters[3], w.h.p.*

$$\|\hat{\mathcal{W}}_{\mathsf{d}}^{\mathsf{a}} - \mathcal{W}_{\star}^{(0)}\|_{\mathrm{F}}^2 \lesssim \frac{r d_1 d_3}{N} + \bar{h}\sqrt{\frac{d_1}{N_S}} + \delta_{\star}^2. \qquad (15)$$

This theorem demonstrates that D-LoRT achieves statistical accuracy comparable to centralized LoRT, particularly when source tasks are similarly informative and sample sizes are sufficiently large. The error bound captures the influence of task heterogeneity ($\bar{h}$), sample sizes ($N$ and $N_S$), and source task alignment ($\delta_{\star}$).

For the two-step D-LoRT method, we obtain:

---

[3]See Appendix for detailed parameter settings.

**Theorem 4.5** (Refined D-LoRT). *Under the conditions of Theorem 4.4, and assuming $N_T \gtrsim r d_1 d_3$, $\bar{h}\sqrt{d_1/N_T} = o(1)$, then, for some chosen parameters[4], w.h.p.*

$$\|\hat{\mathcal{W}}_{\mathsf{dlort}}^{(0)} - \mathcal{W}_{\star}^{(0)}\|_{\mathrm{F}}^2 \lesssim \frac{r d_1 d_3}{N} + \bar{h}\sqrt{\frac{d_1}{N_T}}. \qquad (16)$$

This theorem shows that the two-step D-LoRT method achieves a statistical rate comparable to the centralized LoRT under specific conditions. It effectively balances statistical efficiency and communication cost, making it ideal for high-dimensional tensor regression in decentralized environments, thereby addressing **Q2**.

## 5. Experiments

We evaluate the proposed methods, LoRT and D-LoRT, on both synthetic and real-world datasets in the context of transferable tensor regression[5]. The objective is to evaluate whether transfer learning can significantly improve tensor regression performance under limited sample conditions, using tensor compressed sensing and tensor completion as test scenarios.

### 5.1. Synthetic Data for Tensor Compressed Sensing

In the synthetic experiments, we investigate the impact of transfer learning on tensor compressed sensing under Gaussian design tensor regression models. Following the setup in § 3, we generate low-tubal-rank target task parameters $\mathcal{W}_{\star}^{(0)}$ using $N_T = 200$ target samples and $N_S = 2000$ source samples per task. We vary the number of source tasks ($K$), model shift magnitude ($h_k$), and covariate shift level ($\sigma_S$) to evaluate the robustness of the proposed methods. Due to space constraints, we present only the key results here; comprehensive experimental setups can be found in the Appendix A.1.

*Performance Across Source Tasks.* Fig. 3(a) demonstrates that as the number of source tasks[6] $K$ increases from 1 to 9, LoRT exhibits consistent performance gains, particularly in Step 2. This highlights the effectiveness of the refinement step in leveraging diverse source data. D-LoRT also benefits from additional source tasks but plateaus earlier, likely due to the limitations of decentralized learning. In contrast, baselines TNN (Lu et al., 2018) and $k$-Sup (Wang et al., 2021) show no improvement as they rely solely on the target

---

[4]See Appendix for detailed parameter settings.

[5]A simulated implementation is available at: https://github.com/pingzaiwang/LoRT, including an example for simulating distributed computation.

[6]While we do not explicitly ablate individual terms in the fusion regularizer, their effects are indirectly reflected through varying the number of source tasks (see Fig. 3 for tensor compressed sensing and Table 1(a) for tensor completion). These variations serve as empirical ablations.

task.

*Robustness to Model Shifts.* Fig. 3(b) shows the impact of model shifts. While both steps of LoRT experience increasing errors with larger shifts, Step 2 maintains superior accuracy, underscoring the benefits of refinement in mitigating model discrepancies. D-LoRT follows a similar trend but with slightly higher errors, while TNN and $k$-Sup remain stable yet suboptimal due to their inability to leverage source task data.

*Impact of Covariate Shifts.* Fig. 3(c) highlights the robustness of LoRT to covariate shifts. As $\sigma_S$ varies from 0.3 to 1.8, LoRT remains stable, with Step 2 outperforming Step 1 in most cases. D-LoRT exhibits higher sensitivity to covariate shifts but still outperforms the baselines, which show consistently high errors.

### 5.2. Real-World Data for Tensor Completion

Tensor completion can be viewed as a special case of tensor regression where the design tensor consists of standard tensor bases (Qiu et al., 2022b). In real-world experiments, we investigate the impact of transfer learning on tensor completion tasks. We evaluate LoRT and D-LoRT using YUV RGB video datasets (*akiyo*, *bridge*, *grandma*, and *hall*), where each video frame is represented as a $128 \times 128 \times 3$ tensor. Completion performance is measured using Peak Signal-to-Noise Ratio (PSNR) [7] . The target task aims to reconstruct a video frame from a very limited number of observed entries. To enhance performance, preceding frames are treated as related source tasks, enabling the model to transfer and utilize information from these tasks for improved reconstruction of the target frame. TNN (Lu et al., 2019b) applied solely to the target task is used as the baseline[8] . Detailed setups and additional results are provided in Appendix A.2.

*Performance Across Source Tasks.* Table 1(a) illustrates the trend that LoRT benefits from an increasing number of source tasks ($K = 2$ to 8). For instance, on the *akiyo*

---

[7]Other perceptual metrics such as SSIM (Wang et al., 2004) and LPIPS (Snell et al., 2017) were also examined. Since our focus is on transfer effectiveness rather than perceptual fidelity, we report PSNR and RE. Preliminary checks indicated that SSIM trends largely mirrored PSNR across Table 7 and Table 8, offering limited additional insight. Given this and the nontrivial cost of full evaluation, we omit these metrics in the current version. Additional results will be made available at https://github.com/pingzaiwang/LoRT.

[8]To our knowledge, no existing transfer learning methods directly apply to our setting, which involves tensor completion under extreme sparsity and distribution shifts. In this regime, target-only baselines across tensor formats (e.g., t-SVD, Tucker, CP, TT) generally perform poorly due to insufficient observations. As these methods show no substantial difference or comparative value under such conditions, we report TNN (Lu et al., 2019b) as a representative baseline to illustrate the benefit of transfer. Additional comparisons are provided in Table 6.

dataset, the PSNR gradually improves with more source tasks, reflecting LoRT's ability to aggregate complementary information from diverse sources. LoRT-Step2 consistently shows stronger trends compared to D-LoRT and TNN, indicating the importance of refinement in utilizing source-task knowledge.

*Impact of Sampling Rates.* Tables 1(b) and 1(c) reveal how source and target sampling rates (SR) affect performance on the *akiyo* dataset. Higher source SR leads to a more stable improvement in PSNR, while higher target SR amplifies the reconstruction accuracy. These trends suggest LoRT's robustness in handling varying data availability and its capacity to balance source and target contributions effectively.

Table 1: Performance comparison of LoRT, D-LoRT, and TNN on the *akiyo* dataset. Results are reported in PSNR.

(a) Impact of the number of source tasks ($K$).

| $K$ | LoRT Step1 | LoRT Step2 | D-LoRT | TNN |
|---|---|---|---|---|
| 2 | 28.60 | 28.67 | 26.64 | 14.49 |
| 4 | 35.44 | 35.59 | 27.12 | 14.50 |
| 6 | 37.58 | 37.61 | 27.18 | 14.30 |
| 8 | 40.29 | 40.32 | 27.37 | 14.24 |

(b) Effect of source sampling rates (SR).

| SR | LoRT Step1 | LoRT Step2 | D-LoRT | TNN |
|---|---|---|---|---|
| 50% | 31.88 | 31.92 | 27.41 | 14.50 |
| 70% | 34.43 | 34.51 | 27.56 | 14.20 |
| 90% | 36.42 | 36.60 | 27.65 | 14.38 |

(c) Effect of target sampling rates (SR).

| SR | LoRT Step1 | LoRT Step2 | D-LoRT | TNN |
|---|---|---|---|---|
| 5% | 35.10 | 35.31 | 27.50 | 14.50 |
| 10% | 37.13 | 37.22 | 27.51 | 17.91 |
| 15% | 37.35 | 37.71 | 27.51 | 19.70 |

**Summary of Experimental Results.** The experiments demonstrate that the proposed transfer learning framework, LoRT, effectively enhances tensor compressed sensing and completion tasks under limited sample conditions. By utilizing multiple source tasks and a two-step refinement design, LoRT achieves consistent improvements over baseline methods across various settings, showcasing its potential as a practical tool for tensor compressed sensing and completion in data-constrained scenarios.

## 6. Conclusion and Extensions

To address the challenge of limited sample sizes in tensor regression, we propose LoRT, a framework that leverages transfer learning to enhance target tasks by utilizing knowledge from related source tasks. LoRT tackles key challenges such as model and covariate shifts through a novel fusion regularizer and a two-step refinement process that adapts to distributional differences. Furthermore, we extend LoRT to

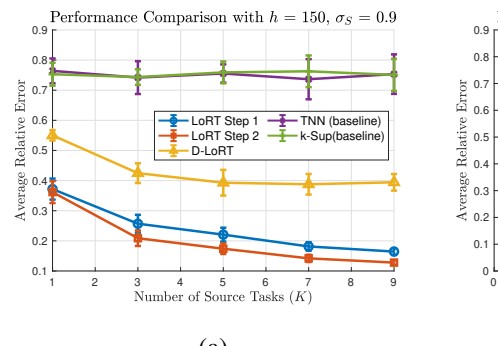
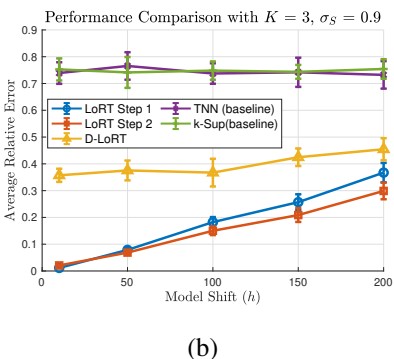
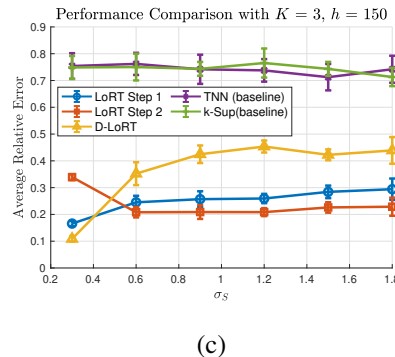

(a)          (b)          (c)

Figure 3: Comparison of average relative error for LoRT, D-LoRT, TNN, and $k$-Sup under varying conditions. (a) Average Relative Error vs. Number of Source Tasks ($K$) for LoRT, D-LoRT, TNN, and $k$-Sup ($h = 150, \sigma_S = 0.9$). (b) Average Relative Error vs. Model Shift ($h$) for LoRT, D-LoRT, TNN, and $k$-Sup ($K = 3, \sigma_S = 0.9$). (c) Average Relative Error vs. Covariate Shift ($\sigma_S$) for LoRT, D-LoRT, TNN, and $k$-Sup ($K = 3, h = 150$).

decentralized settings with D-LoRT, which reduces communication costs while retaining statistical efficiency. Through rigorous theoretical analysis and empirical validation on tasks like tensor compressed sensing and completion, LoRT proves to be a robust and effective solution for advancing tensor regression in data-scarce and complex scenarios.

**Limitations and Extensions.** This work offers a first theoretical framework for transfer tensor regression under low-sample and distribution-shifted settings. Several directions merit further exploration:

- *Computational complexity.* The LoRT framework involves repeated applications of the TNN proximal operator (Lu et al., 2019b). Each such call requires $\mathcal{O}(d_3 \cdot \min\{d_1, d_2\} \cdot d_1 d_2)$ operations due to per-slice SVDs, making large-scale deployment computationally demanding. Further work may explore randomized tensor decompositions to reduce runtime.

- *Extension to broader tensor formulations.* While LoRT is instantiated with the t-SVD, its design is broadly compatible with alternative tensor decompositions, including Tucker, Tensor Train, and Tensor Ring (Liu et al., 2013; Imaizumi et al., 2017; Qiu et al., 2022b). Beyond the t-product family, formulations based on HOSVD (De Lathauwer et al., 2000) or Tensor Regression Networks (TRN) (Kossaifi et al., 2020) offer complementary modeling paradigms that can be integrated within the LoRT transfer pipeline.

- *Beyond the i.i.d. setting.* The current analysis assumes *i.i.d.* sampling within tasks, which simplifies estimation bounds. Real-world tensor data often involve dependencies (e.g., spatiotemporal structure), motivating future extensions to *non-i.i.d.* regimes. See preliminary experiments in Table 7.

- *Real-world TCS applications.* Our synthetic TCS experiments follow theoretical convention. Applying LoRT to real sensing setups (e.g., MRI, hyperspectral imaging) requires modeling domain-specific structures and noise, which we leave for future work. We provide preliminary results in Table 8.

- *Real-world data without ground truth.* A key gap in our study is the lack of experiments on real-world tensor data without ground truth. We follow common practice in theory-driven work (Lu et al., 2018; Zhang et al., 2020; Wang et al., 2021) by focusing on controlled settings to directly validate theoretical predictions. Nonetheless, evaluating LoRT on real datasets without known parameters is a valuable direction for future work.

- *Adaptive control of negative transfer.* Our theoretical analysis ensures improvement under moderate heterogeneity, consistent with the observation in He et al. (2024a) that effective transfer generally requires adaptation. The modular fusion mechanism in LoRT enables task-specific weighting based on alignment metrics, thereby downweighting uninformative or misaligned sources. This design naturally supports heterogeneity-aware regularization strategies such as He et al. (2024b); Duan & Wang (2023), which interpolate between full-source fusion and target-only estimation.

## Acknowledgements

The authors sincerely thank the Area Chair and the four anonymous reviewers for their detailed and constructive feedback. Their suggestions have greatly enhanced the quality and clarity of this paper. This work was supported in part by the National Natural Science Foundation of

China under Grant 62203124, and the JSPS KAKENHI Grant Numbers JP25K21283, JP24K20849, JP23K28109, JP24K03005, JP25K21288, and RIKEN Incentive Research Project 100847-202301062011. Yuning Qiu was supported by the RIKEN Special Postdoctoral Researcher Program.

## Impact Statement

This paper develops the Low-Rank Tensor Transitions (LoRT) framework, which systematically addresses the challenges of transfer learning in tensor regression, particularly under limited data and distributional shifts. While the proposed methods offer potential benefits for applications in scientific domains such as neuroimaging and spatiotemporal analysis, the primary contribution is theoretical. Given its focus on methodological advancements rather than direct application deployment, the societal impact of this work is expected to be indirect, primarily influencing future research in tensor-based learning and transfer learning theory.

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

# Supplemental Material for
## *Low-Rank Tensor Transitions (LoRT) for Transferable Tensor Regression*

This supplemental material provides a comprehensive set of theoretical derivations, proofs, and extended experimental results to support the main findings of our paper on Low-Rank Tensor Transitions (LoRT) for transferable tensor regression. The content is organized to offer a thorough understanding of the proposed methodology, its theoretical underpinnings, and its empirical performance across diverse scenarios. Below, we outline the structure of the appendix.

Section A presents an extensive set of additional experimental results to validate the performance of LoRT and its distributed variant, D-LoRT, on both synthetic and real-world datasets. These experiments evaluate the methods' robustness to model and covariate distribution shifts, the impact of varying source task numbers, and performance under different sampling rates in tensor compressed sensing and completion tasks. Comprehensive comparisons against baselines, such as TNN-based regression, highlight the advantages of LoRT and D-LoRT in addressing transfer learning challenges in tensor regression. Additionally, this section includes detailed experimental settings, results on further datasets (e.g., UCF-101 video sequences), and comparisons with other tensor completion methods.

Section B reviews related work, positioning our contributions within the broader context of tensor regression, transfer learning, and multitask learning. This section discusses advancements in low-rank tensor methods and their applications, emphasizing how LoRT integrates transfer learning principles to handle distribution shifts and limited sample sizes effectively.

Section C introduces essential preliminaries, providing a detailed explanation of the t-SVD and key tensor operations that underpin our theoretical framework. This section ensures clarity for readers by defining fundamental concepts and properties critical to understanding the subsequent theoretical analysis.

Section D delves into the theoretical analysis of LoRT, presenting rigorous proofs of the main theorems and supporting lemmas. This section establishes performance guarantees for LoRT, demonstrating its effectiveness in high-dimensional tensor regression under model and covariate shifts, with a focus on the joint low-rank learning and target-specific refinement steps.

Section E provides the theoretical analysis of D-LoRT, including proofs of convergence for the distributed framework. It highlights the method's ability to leverage local model aggregation while maintaining performance comparable to centralized LoRT, particularly in privacy-sensitive or bandwidth-constrained settings.

Section F details the algorithmic design and comparison of LoRT and D-LoRT, focusing on the proximal gradient descent (PGD) implementations for both methods. This section also compares their computational complexity and communication overhead.

A comprehensive list of notations and symbols used throughout the paper and appendix is provided at the end, ensuring consistency and accessibility for readers.

**Ongoing Challenges and Future Directions.** While this appendix consolidates the basic theoretical and empirical analysis of LoRT and D-LoRT, we emphasize that this work marks only an initial step toward transferable tensor learning. Developing a fully mature framework in this direction requires addressing several open challenges, such as the design of tighter generalization bounds under more adversarial shifts, adaptive task selection mechanisms in large-scale heterogeneous environments, and communication-efficient protocols for federated tensor optimization. We hope this work lays the groundwork for further research at the intersection of tensor modeling, transferability, and decentralized learning.

# Contents

# A. Additional Experimental Results

This section provides comprehensive experimental evidence supporting the effectiveness of LoRT and its distributed variant, D-LoRT. We assess their robustness to model and covariate shifts, performance under varying source task numbers and sampling rates, and their advantages over classical baselines in both tensor compressed sensing and completion tasks. The results span synthetic and real-world datasets, including YUV and UCF-101 video sequences, and include detailed implementation settings and extended comparisons with representative tensor completion methods.

## A.1. Synthetic Data for Tensor Compressed Sensing

In this section, we evaluate the performance of our proposed methods, LoRT and D-LoRT, from the perspective of tensor compressed sensing. The experiments aim to assess their ability to recover low-rank tensor structures and leverage transfer learning to improve performance under limited sample size. We also compare them against two baselines: TNN-based regression (Lu et al., 2018) and $k$-Support-norm-based ($k = 2$) regression (Wang et al., 2021), referred to as *TNN (baseline)* and *$k$-Sup (baseline)*.

The target parameter tensor $\mathcal{W}_\star^{(0)} \in \mathbb{R}^{d_1 \times d_2 \times d_3}$ with a tubal rank $r$ is generated as $\mathcal{W}_\star^{(0)} = \mathcal{P} *_M \mathcal{Q}$, where $\mathcal{P} \in \mathbb{R}^{d_1 \times r \times d_3}$ and $\mathcal{Q} \in \mathbb{R}^{r \times d_2 \times d_3}$ are *i.i.d.* samples from $\mathcal{N}(0, 1)$. We consider a high-dimensional tensor regression problem with dimensions $d_1 = d_2 = 20$, $d_3 = 3$, and a low-rank level $r = 2$. We generate $N_T = 200$ independent target samples $(y_i^{(0)}, \mathcal{X}_i^{(0)})$ using $y_i^{(0)} = \langle \mathcal{X}_i^{(0)}, \mathcal{W}_\star^{(0)} \rangle + \epsilon_i^{(0)}$, where $\mathrm{vec}(\mathcal{X}_i^{(0)}) \sim \mathcal{N}(0, \mathbf{I})$ and $\epsilon_i^{(0)} \sim \mathcal{N}(0, 0.1)$.

The source sample size is set to $N_S = 2000$, with the number of source tasks $K$ varying from 1 to 9. The parameter $h_k$ is chosen from values ranging between 10 and 200. To simulate model and covariate shifts, the source tasks are configured as follows:

- *Model Shift:* Model shifts are simulated by setting $\mathcal{W}_\star^{(k)} = \mathcal{W}_\star^{(0)} + \mathcal{E}^{(k)}$ for $k \in [K]$, where $\mathcal{E}^{(k)} = \mathcal{P}_k *_M \mathcal{Q}_k$, with $\mathcal{P}_k \in \mathbb{R}^{d_1 \times r \times d_3}$ and $\mathcal{Q}_k \in \mathbb{R}^{r \times d_2 \times d_3}$ sampled *i.i.d.* from $\mathcal{N}(0, 1)$. If $\|\mathcal{E}^{(k)}\|_\star > h_k$, then $\mathcal{E}^{(k)}$ is rescaled as $\mathcal{E}^{(k)} = h_k \cdot \mathcal{E}^{(k)} / \|\mathcal{E}^{(k)}\|_\star$.

- *Covariate Shift:* To assess robustness to covariate shifts, a heterogeneous design is used where $\mathrm{vec}(\mathcal{X}_i^{(k)}) \sim \mathcal{N}(0, \sigma_S^2 \mathbf{I})$ for all $k \in [K]$. The value of $\sigma_S$ is selected from the set $\{0.3, 0.6, 0.9, 1.2, 1.5, 1.8\}$. For each source task, $N_S$ independent samples are generated.

**Experimental Results.** The experimental results are shown in Fig. 3. Fig. 3-(a) demonstrates the effect of the number of source tasks on method performance. As the number of tasks increases from one to nine, LoRT exhibits significant improvement, with Step 2 consistently outperforming Step 1. The widening gap between Step 1 and Step 2 indicates that the refinement step becomes more effective with diverse source data. D-LoRT also shows improvement with increasing tasks but plateaus faster than LoRT, possibly due to limitations in decentralized learning. TNN and $k$-Sup, which only utilize target task data, are unable to leverage multiple source tasks, resulting in no improvement.

The impact of model shifts is illustrated in Fig. 3-(b). As the model shift increases, both steps of LoRT show increasing error, with Step 2 maintaining superior performance. The growing gap between Step 1 and Step 2 at larger shifts underscores the increased benefit of the refinement step under significant model discrepancies. D-LoRT follows a similar trend to LoRT but with higher error rates, while maintaining relative stability across shifts. TNN and $k$-Sup exhibit stable yet suboptimal performance due to their inability to harness information from multiple source tasks.

Fig. 3-(c) showcases the influence of covariate shifts. As the covariate shift parameter ranges from 0.3 to 1.8, LoRT demonstrates stable performance, with Step 2 outperforming Step 1 most of the time. This stability highlights LoRT robustness to changing data distributions. D-LoRT displays more sensitivity to covariate shifts, with increasing error as the shift parameter grows, but still outperforms TNN and $k$-Sup for most shift values. TNN and $k$-Sup show the highest and most stable error across all covariate shifts.

**Summary of Findings.** The experimental results demonstrate the performance characteristics of LoRT, D-LoRT, TNN and $k$-Sup under various conditions:

- *Performance across tasks*: LoRT and D-LoRT demonstrate the advantages of transfer learning as the number of source tasks increases, with LoRT showing more consistent gains. In contrast, TNN and $k$-Sup maintain stable but suboptimal

performance due to their inability to utilize information from multiple source tasks.

- *Robustness to distribution shifts*: LoRT and D-LoRT effectively manage both model and covariate shifts, with LoRT showing more consistent performance across different conditions. D-LoRT also demonstrates robustness, though it may experience slight variability in stability under certain shifts.

- *Centralized vs. Distributed learning*: The centralized LoRT method generally outperforms its distributed counterpart, D-LoRT. However, D-LoRT still shows improved performance over TNN and $k$-Sup, potentially offering a balance between performance and data privacy.

### A.2. Real-World Data for Tensor Completion

Tensor completion can be regarded as a significant special case of tensor regression, characterized by covariates that are standard tensor basis elements (Qiu et al., 2024). In this setting, the relationship between tensor covariates and a response is modeled as

$$y_i = \langle \mathbfcal{X}_i, \mathbfcal{W}_\star \rangle + \varepsilon_i, \tag{A.1}$$

where $\mathbfcal{X}_i$ represents *i.i.d.* random tensor basis elements as covariates, $\mathbfcal{W}_\star$ is the parameter tensor encoding the complete data, and $y_i$ denotes the observed entries of the incomplete tensor. The objective is to recover $\mathbfcal{W}_\star$ using only a limited number of observed responses $y_i$.

This formulation highlights the structural similarities between tensor completion and general tensor regression while focusing on the unique challenges posed by missing data and sparse observations. By explicitly leveraging the low-rank structure of $\mathbfcal{W}_\star$, tensor completion provides an ideal platform for exploring the advantages of transfer learning. Specifically, when the sample size is limited, incorporating information from related tasks (e.g., similar incomplete tensors) can substantially improve the estimation accuracy of $\mathbfcal{W}_\star$. This transfer learning approach not only addresses data sparsity but also enhances robustness to distributional shifts, thereby demonstrating its potential in tensor regression scenarios.

**Experiment Setup.** In our real-world experiments, we investigate the effect of transfer learning on tensor completion tasks, focusing on YUV RGB video datasets[9] (*akiyo*, *bridge*, *grandma*, and *hall*). Each frame of the videos is represented as a $128 \times 128 \times 3$ tensor. The target task involves reconstructing the current video frame from a very sparsely observed subset of its entries. To enhance this reconstruction process, transfer learning is leveraged by treating prior video frames as related source tasks. These source frames share similar spatiotemporal structures with the target frame, providing valuable contextual information. The LoRT and D-LoRT models utilize this information by transferring knowledge from the source tasks to the target task. This enables the models to integrate patterns and relationships learned from earlier frames, resulting in more accurate and robust completion of the target frame, even under conditions of sparse observations. A series of experiments were designed to investigate the impact of different factors on the tensor completion task, including the number of source tasks, sampling rates on source tasks and target tasks, and noise levels.

To evaluate the experimental results, we adopt the widely used relatively error (RE) and Peak Signal-to-Noise Ratio (PSNR) metrics, where the RE is defined as

$$\text{RE} = \frac{\|\hat{\mathbfcal{W}} - \mathbfcal{W}_\star\|_\text{F}}{\|\mathbfcal{W}_\star\|_\text{F}} \tag{A.2}$$

where $\hat{\mathbfcal{W}}$ and $\mathbfcal{W}_\star$ are the estimated tensor and ground truth tensor, respectively, and PSNR is given by

$$\text{PSNR} = 10 \log_{10} \left( \frac{\text{MAX}^2}{\text{MSE}} \right) \tag{A.3}$$

where MAX and MSE denote the maximum element and the mean squared error between the ground truth and reconstructed tensors.

**Low-Rank Inductive Bias of the Tubal Nuclear Norm.** The tubal nuclear norm (TNN), defined as the convex surrogate of the tubal rank under the t-SVD framework (Definition 2.3), effectively promotes low-rank structure in third-order tensors. By minimizing the sum of nuclear norms across frontal slices in the Fourier (or DCT) domain, TNN induces spectral sparsity and suppresses uninformative components. This behavior is analogous to the matrix nuclear norm in classical low-rank

---

[9]Available at http://trace.eas.asu.edu/yuv/

recovery, but adapted to preserve the multi-dimensional correlations inherent in tensor data. Empirical results (e.g., Fig. 4) demonstrate that TNN regularization leads to a sharp decay in singular values and energy concentration among a small number of spectral components, validating its role as a powerful inductive bias for modeling low-tubal-rank structure. This property underlies the effectiveness of our approach in compressive sensing and transfer learning settings, where sample efficiency and structural fidelity are critical.

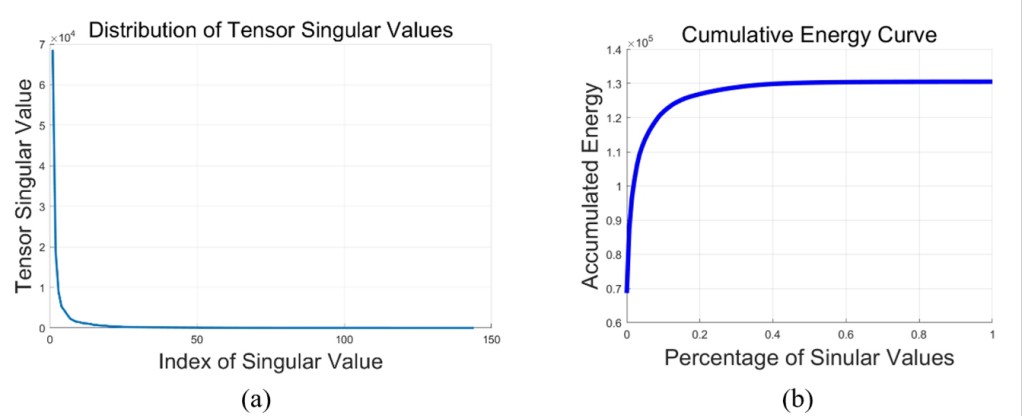

Figure 4: Empirical evidence of the low-tubal-rank structure induced by the tubal nuclear norm. The plots correspond to the recovered tensor *(ninth frame of the Akiyo video)* and exhibit clear low-rank characteristics: (a) a sharp spectral decay of the tensor singular values, and (b) a rapid saturation in the cumulative energy curve.

### A.2.1. EXPERIMENTAL RESULTS ON THE VARYING NUMBER OF SOURCE TASKS

We conducted a series of controlled experiments to investigate the performance of our algorithms under various different source tasks. The number of tasks is selected from the candidate set $\{1, 2, 3, 4, 5, 6, 7, 8\}$. For simplicity, we let the first $K$ frame be the source tasks tensor and the $(K+1)$-th frame as the target tensor. The sampling rate source tensors and target tensor are set to $80\%$ and $5\%$, respectively.

The experimental results are reported in Table 2 and 3. The results demonstrate the efficacy of the proposed Low-rank Tensor (LoRT) method in leveraging multiple source tasks for improved tensor completion. As the number of source tasks $K$ increases, LoRT exhibits a consistent performance improvement, with PSNR values showing substantial gains and RE values decreasing significantly across all video datasets. This trend underscores the method's ability to harness information from multiple source tasks effectively. Notably, LoRT, particularly in its LoRT-Step2 implementation, consistently outperforms other methods, including D-LoRT and the TNN baseline. The *akiyo* video demonstrates the most pronounced improvement with LoRT, showcasing a remarkable PSNR increase from approximately 22dB at *K=1* to over 40dB at *K=8*. While D-LoRT generally surpasses the TNN baseline, it falls short of LoRT's performance, suggesting potential for refinement in its fusion strategy. The TNN baseline, as expected, maintains relatively constant performance across different $K$ values due to its sole reliance on target task data. These findings not only validate the proposed LoRT approach but also highlight its superiority in transfer tensor completion scenarios.

Figure 5 illustrates the reconstruction quality achieved by various algorithms, including LoRT-Step1, LoRT-Step2, LoRT-D, and TNN approaches, under $K = 4$ and $K = 8$ settings. The visual results demonstrate the efficacy of the proposed LoRT approach in recovering missing data across diverse video sequences, highlighting the impact of different $K$ values on the completion accuracy.

### A.2.2. EXPERIMENTAL RESULTS ON VARYING SOURCE TASKS SR

We designed a comprehensive set of experiments to evaluate the performance of our algorithms under varying source sampling rates (SR). The SR values for source tasks were selected from the set $\{50\%, 60\%, 70\%, 80\%, 90\%\}$, and the SR for the target task is fixed to $5\%$. In our experimental framework, we designated the first 4 frames as the source task tensor and the 5th frame as the target tensor.

The experimental results presented in Tables 4 and 5 demonstrate the effectiveness of the proposed LoRT method in

Table 2: PSNR value comparison on *akiyo*, *bridge*, *grandma*, and *hall* videos under varying source tasks.

| | | *akiyo* | | | | *bridge* | | |
|---|---|---|---|---|---|---|---|---|
| $K$ | LoRT Step1 | LoRT Step2 | D-LoRT | TNN (baseline) | LoRT Step1 | LoRT Step2 | D-LoRT | TNN (baseline) |
| 1 | 22.15 | 22.19 | 25.71 | 14.49 | 33.03 | 33.04 | 30.33 | 23.59 |
| 2 | 28.60 | 28.67 | 26.64 | 14.49 | 34.99 | 35.04 | 31.08 | 23.82 |
| 3 | 32.90 | 33.01 | 26.97 | 14.49 | 36.06 | 36.14 | 31.22 | 23.73 |
| 4 | 35.44 | 35.59 | 27.12 | 14.50 | 36.53 | 36.62 | 31.38 | 23.68 |
| 5 | 35.60 | 35.77 | 27.13 | 14.51 | 36.96 | 37.06 | 31.31 | 23.45 |
| 6 | 37.58 | 37.61 | 27.18 | 14.30 | 37.31 | 37.42 | 31.46 | 23.34 |
| 7 | 39.73 | 39.77 | 27.29 | 14.38 | 37.63 | 37.76 | 31.61 | 23.65 |
| 8 | 40.29 | 40.32 | 27.37 | 14.24 | 37.57 | 37.70 | 31.60 | 23.52 |
| | | *grandma* | | | | *hall* | | |
| $K$ | LoRT Step1 | LoRT Step2 | D-LoRT | TNN (baseline) | LoRT Step1 | LoRT Step2 | D-LoRT | TNN (baseline) |
| 1 | 29.10 | 29.10 | 26.60 | 17.45 | 28.52 | 28.52 | 25.65 | 16.10 |
| 2 | 32.59 | 32.59 | 27.24 | 17.44 | 31.88 | 31.89 | 26.47 | 16.08 |
| 3 | 34.88 | 34.88 | 27.47 | 17.43 | 33.04 | 33.06 | 26.68 | 16.02 |
| 4 | 36.60 | 36.61 | 27.62 | 17.42 | 35.06 | 35.08 | 27.04 | 15.95 |
| 5 | 37.79 | 37.80 | 27.65 | 17.34 | 35.75 | 35.77 | 27.11 | 16.11 |
| 6 | 38.84 | 38.85 | 27.79 | 17.48 | 36.40 | 36.44 | 27.28 | 16.17 |
| 7 | 39.61 | 39.63 | 27.72 | 17.24 | 36.73 | 36.77 | 27.28 | 16.12 |
| 8 | 40.37 | 40.39 | 27.78 | 18.13 | 36.24 | 36.29 | 27.17 | 16.13 |

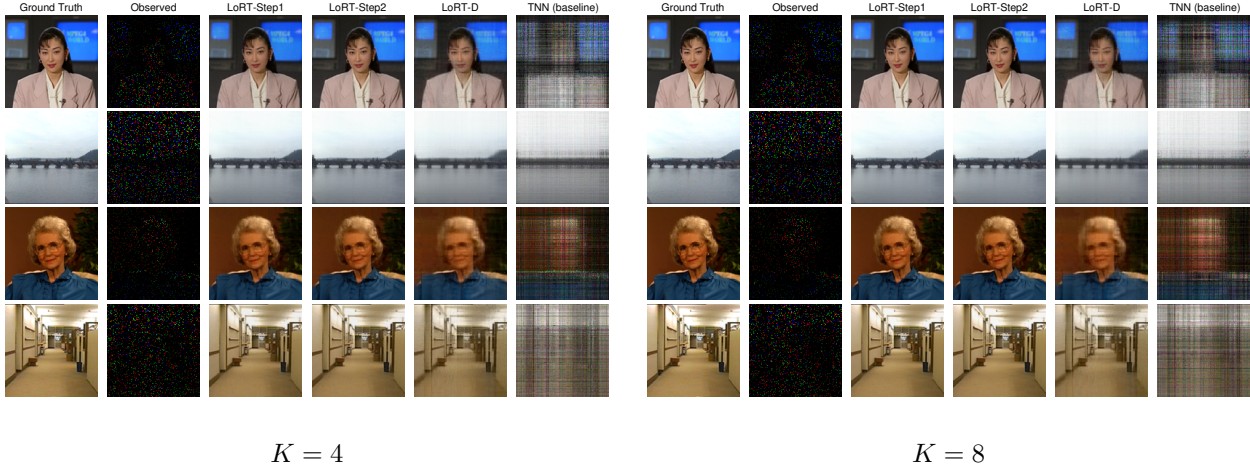

$K = 4$          $K = 8$

Figure 5: Comparison of completion performance across different methods for four video datasets under (a) $K = 4$ and (b) $K = 8$ settings.

Table 3: RE value comparison on *akiyo*, *bridge*, *grandma*, and *hall* videos under varying source tasks.

| | akiyo | | | | bridge | | | |
|---|---|---|---|---|---|---|---|---|
| $K$ | LoRT Step1 | LoRT Step2 | D-LoRT | TNN (baseline) | LoRT Step1 | LoRT Step2 | D-LoRT | TNN (baseline) |
| 1 | 0.1593 | 0.1588 | 0.1058 | 0.3851 | 0.0292 | 0.0291 | 0.0398 | 0.0865 |
| 2 | 0.0759 | 0.0753 | 0.0951 | 0.3850 | 0.0233 | 0.0231 | 0.0365 | 0.0842 |
| 3 | 0.0462 | 0.0457 | 0.0915 | 0.3851 | 0.0206 | 0.0204 | 0.0360 | 0.0851 |
| 4 | 0.0345 | 0.0339 | 0.0899 | 0.3848 | 0.0195 | 0.0193 | 0.0353 | 0.0857 |
| 5 | 0.0339 | 0.0332 | 0.0899 | 0.3839 | 0.0186 | 0.0183 | 0.0356 | 0.0879 |
| 6 | 0.0270 | 0.0269 | 0.0893 | 0.3935 | 0.0178 | 0.0176 | 0.0350 | 0.0890 |
| 7 | 0.0211 | 0.0210 | 0.0882 | 0.3901 | 0.0172 | 0.0169 | 0.0344 | 0.0860 |
| 8 | 0.0197 | 0.0197 | 0.0875 | 0.3962 | 0.0173 | 0.0171 | 0.0344 | 0.0872 |
| | grandma | | | | hall | | | |
| $K$ | LoRT Step1 | LoRT Step2 | D-LoRT | TNN (baseline) | LoRT Step1 | LoRT Step2 | D-LoRT | TNN (baseline) |
| 1 | 0.1259 | 0.1260 | 0.1679 | 0.4818 | 0.0659 | 0.0659 | 0.0918 | 0.2757 |
| 2 | 0.0845 | 0.0846 | 0.1566 | 0.4834 | 0.0448 | 0.0448 | 0.0836 | 0.2765 |
| 3 | 0.0647 | 0.0647 | 0.1518 | 0.4824 | 0.0392 | 0.0391 | 0.0815 | 0.2779 |
| 4 | 0.0532 | 0.0531 | 0.1494 | 0.4836 | 0.0311 | 0.0310 | 0.0782 | 0.2806 |
| 5 | 0.0464 | 0.0464 | 0.1492 | 0.4891 | 0.0287 | 0.0286 | 0.0775 | 0.2754 |
| 6 | 0.0412 | 0.0412 | 0.1472 | 0.4827 | 0.0266 | 0.0265 | 0.0761 | 0.2732 |
| 7 | 0.0376 | 0.0375 | 0.1477 | 0.4936 | 0.0256 | 0.0255 | 0.0760 | 0.2748 |
| 8 | 0.0345 | 0.0345 | 0.1471 | 0.4469 | 0.0271 | 0.0270 | 0.0771 | 0.2746 |

leveraging source tasks with varying SR for improved tensor completion. As the SR of the source task increases from 50% to 90%, LoRT exhibits a consistent performance improvement across all video datasets, with PSNR values showing substantial gains and RE values decreasing significantly. Notably, LoRT, particularly in its LoRT-Step2 implementation, consistently outperforms other methods, including D-LoRT and the TNN baseline, across all SR values. The TNN baseline, as expected, shows relatively constant performance across different SR values due to its sole reliance on target task data. This highlights the advantage of LoRT in leveraging source task information.

Table 4: PSNR value comparison on *akiyo*, *bridge*, *grandma*, and *hall* videos under different SR.

| | akiyo | | | | bridge | | | |
|---|---|---|---|---|---|---|---|---|
| SR | LoRT Step1 | LoRT Step2 | D-LoRT | TNN (baseline) | LoRT Step1 | LoRT Step2 | D-LoRT | TNN (baseline) |
| 50% | 31.88 | 31.92 | 27.41 | 14.50 | 30.44 | 30.56 | 31.60 | 23.79 |
| 60% | 33.27 | 33.33 | 27.47 | 14.23 | 31.71 | 31.89 | 31.69 | 23.14 |
| 70% | 34.43 | 34.51 | 27.56 | 14.20 | 32.93 | 33.21 | 31.75 | 23.63 |
| 80% | 35.41 | 35.54 | 27.58 | 14.56 | 33.94 | 34.31 | 31.80 | 24.01 |
| 90% | 36.42 | 36.60 | 27.65 | 14.38 | 34.68 | 35.08 | 31.85 | 23.69 |
| | grandma | | | | hall | | | |
| SR | LoRT Step1 | LoRT Step2 | D-LoRT | TNN (baseline) | LoRT Step1 | LoRT Step2 | D-LoRT | TNN (baseline) |
| 50% | 34.95 | 34.98 | 27.87 | 17.61 | 30.72 | 30.77 | 27.43 | 16.05 |
| 60% | 35.76 | 35.81 | 27.89 | 17.40 | 31.62 | 31.71 | 27.42 | 16.17 |
| 70% | 36.38 | 36.43 | 27.90 | 17.26 | 32.60 | 32.71 | 27.49 | 16.17 |
| 80% | 37.26 | 37.32 | 27.96 | 17.62 | 33.50 | 33.63 | 27.56 | 16.11 |
| 90% | 37.92 | 37.99 | 28.00 | 17.40 | 34.31 | 34.50 | 27.59 | 15.91 |

### A.2.3. EXPERIMENTAL RESULTS ON VARYING TARGET TASK SR

We conducted a series of comprehensive experiments to evaluate the performance of our algorithms under varying target SR. The SR values for the target task were selected from the set {5%, 8%, 10%, 12% 15%, 18%, 20%}, representing a range from sparse to relatively dense sampling of the target task. In our experimental framework, we consistently used the first 4 frames as the source task tensors and the 5th frame as the target tensor across all datasets. For the source tasks, we maintained a fixed sampling rate of 80% for all tensors.

The experimental results presented in Figures 6 demonstrate the effectiveness of the proposed LoRT method in leveraging

Table 5: RE value comparison on *akiyo*, *bridge*, *grandma*, and *hall* videos under different SR.

| | *akiyo* | | | | *bridge* | | | |
|---|---|---|---|---|---|---|---|---|
| SR | LoRT Step1 | LoRT Step2 | D-LoRT | TNN (baseline) | LoRT Step1 | LoRT Step2 | D-LoRT | TNN (baseline) |
| 50% | 0.0520 | 0.0518 | 0.0870 | 0.3848 | 0.0393 | 0.0388 | 0.0344 | 0.0846 |
| 60% | 0.0443 | 0.0440 | 0.0864 | 0.3968 | 0.0340 | 0.0333 | 0.0341 | 0.0912 |
| 70% | 0.0388 | 0.0384 | 0.0855 | 0.3982 | 0.0295 | 0.0286 | 0.0338 | 0.0862 |
| 80% | 0.0346 | 0.0341 | 0.0853 | 0.3819 | 0.0263 | 0.0252 | 0.0336 | 0.0825 |
| 90% | 0.0309 | 0.0302 | 0.0846 | 0.3901 | 0.0241 | 0.0231 | 0.0335 | 0.0856 |

| | *grandma* | | | | *hall* | | | |
|---|---|---|---|---|---|---|---|---|
| SR | LoRT Step1 | LoRT Step2 | D-LoRT | TNN | LoRT Step1 | LoRT Step2 | D-LoRT | TNN |
| 50% | 0.0643 | 0.0641 | 0.1452 | 0.4729 | 0.0512 | 0.0509 | 0.0748 | 0.2773 |
| 60% | 0.0586 | 0.0582 | 0.1448 | 0.4849 | 0.0462 | 0.0457 | 0.0749 | 0.2735 |
| 70% | 0.0545 | 0.0542 | 0.1447 | 0.4925 | 0.0412 | 0.0407 | 0.0743 | 0.2736 |
| 80% | 0.0493 | 0.0489 | 0.1437 | 0.4724 | 0.0372 | 0.0366 | 0.0737 | 0.2753 |
| 90% | 0.0456 | 0.0453 | 0.1431 | 0.4848 | 0.0339 | 0.0331 | 0.0734 | 0.2819 |

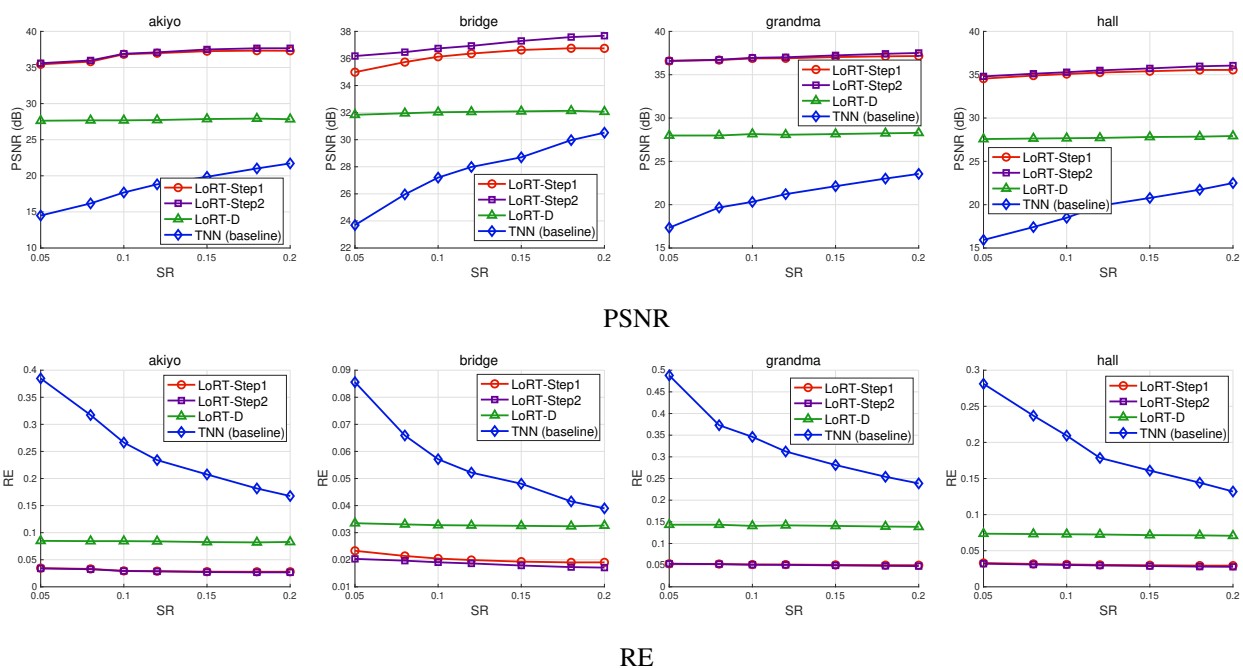

PSNR

RE

Figure 6: Comparison of completion performance across different methods for four video datasets under (a) PSNR and (b) RE settings.

target tasks with varying SR for improved tensor completion across different video datasets. As the SR of the target task increases from 5% to 20%, LoRT exhibits a consistent performance improvement across all video datasets, with PSNR values showing substantial gains and RE values decreasing significantly. Notably, LoRT, particularly in its LoRT-Step2 implementation, consistently outperforms other methods, including LoRT-D and the TNN baseline, across all SR values. For instance, in the *akiyo* dataset, LoRT-Step2 achieves the highest PSNR values, ranging from approximately 35 dB at 5% SR to nearly 40 dB at 20% SR.

Figure 7 visually corroborates these quantitative findings, showcasing the superior reconstruction quality of LoRT-Step1, LoRT-Step2, and LoRT-D compared to the baseline, especially at lower SR values (10%). The visual quality improvement is particularly noticeable in complex scenes like *hall* and detailed facial features in *akiyo* and *grandma* datasets. In conclusion, these results highlight the significant advantage of the LoRT method, especially its two-step implementation, in effectively leveraging source task information for improved tensor completion across various sampling rates and video datasets.

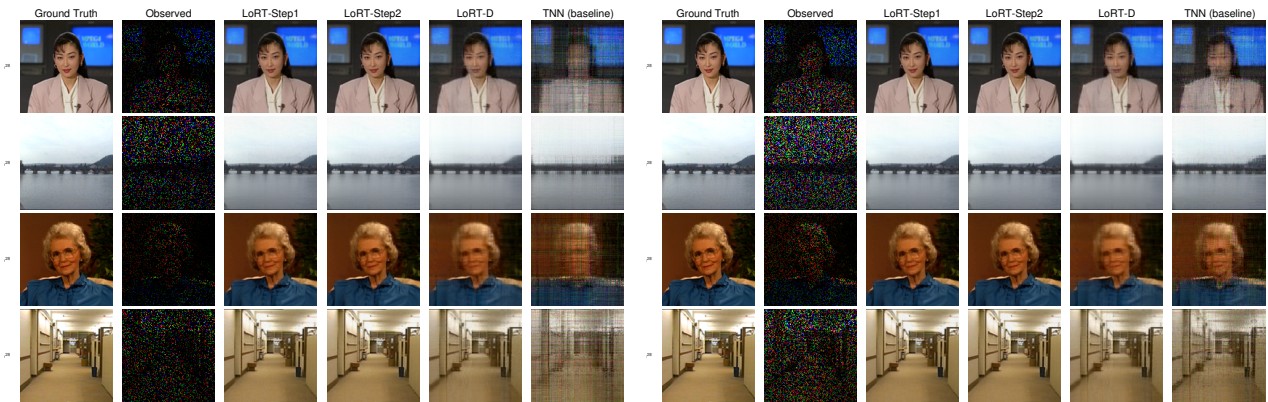

$$\text{SR} = 10\% \text{ for Target Tensor} \qquad\qquad \text{SR} = 20\% \text{ for Target Tensor}$$

Figure 7: Comparison of completion performance across different methods for four video datasets under (a) SR = 10% and (b) SR = 20% settings.

### A.3. Additional Datasets, Baselines, and Experimental Settings

To better demonstrate the robustness and applicability of the proposed LoRT framework, we provide additional experimental results and implementation details beyond the main paper. These additions aim to offer a more complete understanding of the proposed method's empirical behavior across diverse application contexts, especially in visual tensor recovery tasks.

In particular, we incorporate two additional video sequences—*Apply Eye Make-up* and *Blowing Candles*—from the UCF-101 benchmark, as used in Wang & Zhao (2024), to evaluate the generalization performance of LoRT in spatiotemporal tensor completion settings. These datasets are widely used for benchmarking and exhibit diverse motion dynamics and textural structures, making them suitable for testing the adaptability of transfer-based tensor learning algorithms.

#### A.3.1. ADDITIONAL BASELINES

To the best of our knowledge, no existing transfer learning methods directly apply to our setting, which involves tensor completion under extreme sparsity and inter-task distribution shifts. In such regimes, target-only baselines across all tensor formats—including t-SVD, Tucker, CP, and TT—consistently perform poorly due to insufficient observations. Due to the lack of substantive differences among target-only baselines under extreme sparsity, we choose to report the t-SVD-based TNN method (Lu et al., 2019b) as a representative baseline to highlight the benefit of transfer in the main experiments.

To further support this design choice, we also compare LoRT against several representative tensor completion methods proposed. These include Tucker-based methods (*Convex* in Raskutti et al. (2019) and *Tucker* in Li et al. (2018)), *CP*-based methods (Zhou et al., 2013), mixed t-SVD/TR-based approache (*Balanced TNN* in Qiu et al. (2024)), the orientation-invariant TNN method (*OITNN* in Wang et al. (2023b)), and the tensor $\ell_1 - \ell_2$ regularized approach (Tan et al., 2023). However, their target-only nature limits their effectiveness in highly sparse and heterogeneous transfer settings. In fact, their empirical

performance closely resembles that of TNN under identical sampling conditions, offering limited distinction, comparative value, or practical insight from the perspective of transfer learning. Due to this and the associated computational overhead, we report only representative results rather than repeating comparisons across all datasets. Detailed results are provided in Table 6.

Table 6: PSNR results on UCF-101 video clips (*Apply Eye Make-up* and *Blowing Candles*) with varying $K$.

| Dataset | $K$ | LoRT Step 1 | LoRT Step 2 | D-LoRT | TNN | Balanced TNN | OITNN | $\ell_1 - \ell_2$ | CP | Tucker | Convex |
|---|---|---|---|---|---|---|---|---|---|---|---|
| *Eye Make-up* | 1 | 25.09 | **25.16** | 24.41 | 16.01 | 17.26 | 16.19 | 14.73 | 14.19 | 14.79 | 14.73 |
| | 2 | 29.16 | **29.18** | 24.98 | 16.10 | 18.01 | 16.93 | 15.33 | 13.68 | 14.43 | 15.33 |
| | 3 | 29.92 | **29.94** | 25.50 | 16.62 | 18.07 | 17.14 | 15.56 | 14.65 | 14.23 | 15.56 |
| *Blowing Candles* | 1 | 25.63 | **25.64** | 23.14 | 13.59 | 14.49 | 13.65 | 12.01 | 14.04 | 14.96 | 12.01 |
| | 2 | 25.97 | **25.98** | 23.86 | 13.27 | 14.87 | 13.77 | 12.33 | 14.15 | 13.19 | 12.33 |
| | 3 | 28.72 | **28.74** | 25.17 | 13.36 | 14.92 | 13.96 | 12.19 | 12.93 | 13.87 | 12.19 |

### A.3.2. NON-I.I.D. SAMPLING SETTINGS

We further investigate LoRT's robustness under non-i.i.d. tensor sampling patterns, including a mixture of tube-wise and element-wise missingness. These experiments are designed to validate the performance of LoRT in less idealized settings, where tensor observations may be sampled in a structured or nonuniform fashion. Results are summarized in Table 7 and show that LoRT retains its advantage over conventional baseline TNN even under these more realistic sampling regimes. In particular, we observe that LoRT Step 2 consistently yields the highest PSNR and SSIM values across all configurations. The performance gap becomes especially pronounced as the number of source tasks increases, suggesting that LoRT effectively leverages auxiliary information even when source observations are non-i.i.d. Notably, while D-LoRT offers improvements over TNN, it trails behind centralized LoRT, underscoring the benefit of joint optimization when data can be aggregated.

Table 7: Preliminary results on tensor completion under *non-i.i.d. source task measurements*. Each source tensor is partially observed with a 40% sampling rate combining tube-wise and element-wise missing patterns. The target tensor is observed at a 5% sampling rate. PSNR and SSIM are reported for each method.

| Dataset | $K$ | LoRT Step 1 | | LoRT Step 2 | | D-LoRT | | TNN | |
|---|---|---|---|---|---|---|---|---|---|
| | | PSNR | SSIM | PSNR | SSIM | PSNR | SSIM | PSNR | SSIM |
| *Eye Make-up* | 1 | 24.38 | 0.6468 | **24.47** | **0.6507** | 21.89 | 0.6138 | 16.01 | 0.2045 |
| | 2 | 28.48 | 0.7591 | **28.49** | **0.7593** | 23.70 | 0.6472 | 16.47 | 0.2131 |
| | 3 | 29.58 | 0.7854 | **29.60** | **0.7861** | 24.55 | 0.6545 | 16.26 | 0.1641 |
| *Blowing Candles* | 1 | 24.15 | 0.8984 | **24.15** | **0.8985** | 14.39 | 0.5854 | 13.22 | 0.2690 |
| | 2 | 25.17 | 0.9213 | **25.18** | **0.9216** | 18.34 | 0.7689 | 13.43 | 0.3097 |
| | 3 | 27.62 | 0.9517 | **27.63** | **0.9519** | 19.25 | 0.8073 | 13.46 | 0.2987 |

### A.3.3. PRELIMINARY RESULTS ON TENSOR COMPRESSED SENSING

To further evaluate the effectiveness of LoRT beyond tensor completion, we conduct preliminary experiments on tensor compressed sensing. In this setting, each video frame is modeled as a third-order tensor, and the entries are sensed via random Gaussian measurements. Each source task contains 3000 measurements, and the target task is measured with only 1500 observations, reflecting a highly undersampled regime.

We adopt two video sequences (*Apply Eye Make-up* and *Blowing Candles*), with each frame rescaled to $40 \times 40 \times 3$ due to computational constraints. PSNR and SSIM metrics are used to evaluate reconstruction performance across different values of $K$, the number of source tasks. The results in Table 8 show that LoRT Step 2 consistently outperforms both D-LoRT and TNN across all settings of $K$ and on both datasets. D-LoRT, despite its decentralized nature, achieves competitive performance, validating the effectiveness of local model aggregation. In contrast, TNN—trained solely on the target

Table 8: Preliminary results on tensor compressed sensing using UCF-101 video clips. Each source tensor is sensed with 3000 Gaussian projections, and the target tensor is sensed with 1500. PSNR and SSIM are reported.

| Dataset | $K$ | LoRT Step 1 | | LoRT Step 2 | | D-LoRT | | TNN | |
|---------|-----|------|------|------|------|------|------|------|------|
| | | PSNR | SSIM | PSNR | SSIM | PSNR | SSIM | PSNR | SSIM |
| | 1 | 18.29 | 0.5762 | **18.33** | **0.5774** | 14.48 | 0.3868 | 13.54 | 0.2832 |
| *Eye Make-up* | 2 | 29.87 | 0.8461 | **30.13** | **0.8472** | 16.12 | 0.4681 | 13.82 | 0.2942 |
| | 3 | 33.97 | **0.9202** | **34.69** | 0.9126 | 16.96 | 0.5183 | 13.88 | 0.2942 |
| | 1 | 15.86 | 0.6775 | **15.9** | **0.679** | 11.88 | 0.4397 | 9.53 | 0.2326 |
| *Blowing Candles* | 2 | 26.41 | 0.9531 | **26.72** | **0.9549** | 13.03 | 0.5225 | 9.32 | 0.1945 |
| | 3 | 31.42 | 0.986 | **32.03** | **0.9861** | 13.99 | 0.5959 | 9.3 | 0.2008 |

task—suffers from substantially lower PSNR and SSIM, especially under limited measurement conditions, underscoring the advantages of leveraging source task information through transfer.

# B. Related Work

We categorize related work into three key areas: tensor regression and recovery, transfer learning, and multitask learning. Our approach intersects these domains by introducing a theoretically grounded, low-rank tensor regression framework capable of addressing both limited samples and distributional heterogeneity.

**Tensor Regression and Recovery.**    Tensor-based regression and recovery methods have been extensively developed to model multi-dimensional structured data (Qiu et al., 2022b; Hou et al., 2021; Lu et al., 2019a; Zhang et al., 2020; Zhang & Ng, 2021; Zhang, 2016). Early work established foundational tensor decompositions, including CANDECOMP/PARAFAC (CP) (Carroll & Chang, 1970; Harshman et al., 1970) and Tucker decomposition (Tucker, 1966), which were later applied to practical domains such as neuroimaging and chemometrics (Zhou et al., 2013; Sun & Li, 2017). Notably, the STORE model (Sun & Li, 2017) introduced sparsity for efficient estimation, while Raskutti et al. (2019) proposed a convex regularization framework to manage high-dimensional multi-response tensor regression.

In parallel, low-rank tensor recovery has seen both convex and non-convex developments. Convex approaches include tubal nuclear norm (TNN) minimization based on the t-SVD framework (Lu et al., 2019a), as well as theoretical tools like tensor restricted isometry property (T-RIP) (Zhang et al., 2020). Non-convex regularizers have also gained attention, including Schatten-$p$ norms (Kong et al., 2018) and weighted TNN strategies (Mu et al., 2020), aiming to balance accuracy with computational scalability.

Our work builds upon these methods by extending low-tubal-rank modeling to a multi-task transfer setting, introducing a principled fusion regularizer and leveraging source-target interactions for improved tensor regression performance under distribution shift. In particular, we adapt the TNN framework to heterogeneous learning across tasks, while preserving its low-rank structural benefits. This contributes a new perspective to the low-tubal-rank recovery literature by enabling theoretically justified estimation in transfer settings, without relying on centralized access to all data.

**Transfer Learning.**    Transfer learning for regression has developed along both theoretical and algorithmic lines, especially in high-dimensional or low-sample regimes. Kuzborskij & Orabona (2013; 2017) established foundational generalization bounds using algorithmic stability, while Wang et al. (2016) analyzed the excess risk of nonparametric transfer across tasks. More recent work has extended to high-dimensional and structured models: Tian & Feng (2023); Tian et al. (2023) developed theory for generalized linear models under covariate and model shift, and Du et al. (2020); Tripuraneni et al. (2021) studied few-shot and meta-learning with provable representation transfer.

Refinements include surrogate-loss-based frameworks (Aghbalou & Staerman, 2023), smoothness-adaptive penalties (Lin & Reimherr, 2024), and distributionally robust optimization under covariate shift (He et al., 2024a). These methods, however, primarily operate on vectorized models or unstructured data. In contrast, our work incorporates transfer learning directly within the tensor regression framework by designing a decomposition-aware regularizer and two-step refinement strategy, enabling effective transfer under low-rank constraints and structured heterogeneity.

**Multitask Learning.**    Multitask learning has been a rich area of study, particularly in understanding the conditions under which joint training improves generalization. Hanneke & Kpotufe (2022) established a "no-free-lunch" theorem that characterizes when negative transfer is unavoidable. Tensor-based extensions of multitask learning were proposed by Wimalawarne et al. (2014), who used low-rank tensor factorization to encode task relationships through convex regularization.

In terms of information transfer, Wu et al. (2020) provided theoretical insights into when task information can be beneficially shared, and Duan & Wang (2023) proposed adaptive frameworks that dynamically modulate transfer strength according to task relevance. Multi-source and multi-target challenges have also been addressed theoretically: Konstantinov et al. (2020) studied PAC-learning guarantees in adversarial multi-source settings, while Deng et al. (2023) focused on mixture weight estimation in domain adaptation.

Our proposed LoRT and D-LoRT frameworks offer a preliminary step toward structured knowledge transfer in tensor regression, where each task is represented as a low-rank tensor and source-target alignment is incorporated through a fusion regularizer. While modest in scope, this design allows the method to leverage tensor structure under distribution shifts, and provides a scalable implementation for decentralized or heterogeneous data environments, with theoretical insights that complement existing multitask and transfer learning approaches.

# C. Additional Preliminaries

## C.1. Preliminaries of t-Singular Value Decomposition

Due to space limitations, some concepts related to t-SVD were omitted in the main text. We provide additional notions here.

**Definition C.1** (Frontal-slice-wise product (Lu et al., 2019a)). The frontal-slice-wise product of any two tensors $\mathcal{A} \in \mathbb{R}^{d_1 \times d_2 \times d_3}$ and $\mathcal{B} \in \mathbb{R}^{d_1 \times d_2 \times d_3}$, denoted by $\mathcal{A} \odot \mathcal{B}$, is defined as a tensor $\mathcal{T}$ such that

$$\mathcal{T}_{:,:,i} = \mathcal{A}_{:,:,i} \cdot \mathcal{B}_{:,:,i}, \ i \in [K]$$

where $\cdot$ denotes the standard matrix multiplication. The frontal-slice-wise product performs matrix multiplication on each frontal slice of the tensors, resulting in a new tensor.

**Definition C.2** ($M$-block-diagonal matrix). The $M$-block-diagonal matrix of any tensor $\mathcal{T} \in \mathbb{R}^{d_1 \times d_2 \times d_3}$, denoted by $\bar{\mathbf{T}}$, is the block diagonal matrix whose diagonal blocks are the frontal slices of $M(\mathcal{T}) := M(\mathcal{T})$:

$$\bar{\mathbf{T}} := \mathtt{bdiag}(M(\mathcal{T})) := \begin{bmatrix} M(\mathcal{T})_{:,:,1} & & & \\ & M(\mathcal{T})_{:,:,2} & & \\ & & \ddots & \\ & & & M(\mathcal{T})^{(K)} \end{bmatrix} \in \mathbb{R}^{d_1 d_3 \times d_2 d_3}.$$

This concept arranges the slices of a tensor in the frequency domain into a block diagonal matrix, facilitating the theoretical analysis of t-SVD.

We further provide some definitions and properties related to t-SVD:

**Definition C.3** ((Kernfeld et al., 2015)). The t-transpose of a tensor $\mathcal{T} \in \mathbb{R}^{d_1 \times d_2 \times d_3}$ under the $M$ transform (as shown in Eq. (1)), denoted by $\mathcal{T}^\top$, satisfies

$$M(\mathcal{T}^\top)_{:,:,i} = \left(M(\mathcal{T})_{:,:,i}\right)^\top, \ i \in [K].$$

In other words, the t-transpose performs a transpose on each slice in the frequency domain and then transforms back to the time domain. This operation is one of the foundations of t-SVD theory.

**Definition C.4** ((Kernfeld et al., 2015)). The t-identity tensor $\mathcal{I} \in \mathbb{R}^{d \times d \times d_3}$ under the $M$ transform satisfies that each frontal slice of $M(\mathcal{I})$ is an $d_3 \times d_3$ identity matrix, i.e.,

$$M(\mathcal{I})_{:,:,i} = \mathbf{I}, \ i \in [K].$$

It is easy to verify that $\mathcal{T} *_M \mathcal{I} = \mathcal{T}$ and $\mathcal{I} *_M \mathcal{T} = \mathcal{T}$ hold for appropriate dimensions. The t-identity tensor plays a role similar to the identity matrix in t-SVD.

**Definition C.5** ((Kernfeld et al., 2015)). A tensor $\mathcal{Q} \in \mathbb{R}^{d \times d \times d_3}$ is called t-orthogonal under the $M$ transform if it satisfies

$$\mathcal{Q}^\top *_M \mathcal{Q} = \mathcal{Q} *_M \mathcal{Q}^\top = \mathcal{I}.$$

T-orthogonality is an important property of tensor transformations, ensuring that the inner product and norm of tensors remain invariant before and after the transformation.

**Decomposability of Tubal Nuclear Norm**   Consider the reduced t-SVD of $\mathcal{W}_\star^{(0)}$ given by

$$\mathcal{W}_\star^{(0)} = \mathcal{U} *_M \mathcal{S} *_M \mathcal{V}^\top$$

where $\mathcal{U} \in \mathbb{R}^{d_1 \times r \times d_3}$ and $\mathcal{V} \in \mathbb{R}^{d_2 \times r \times d_3}$ are orthogonal tensors, and $\mathcal{S} \in \mathbb{R}^{r \times r \times d_3}$ is an f-diagonal tensor. We define the projection operators $\mathcal{P}_\star(\cdot)$ and $\mathcal{P}_{\star\perp}(\cdot)$ as follows:

$$\mathcal{P}_\star(\mathcal{T}) = \mathcal{U} *_M \mathcal{U}^\top *_M \mathcal{T} + \mathcal{T} *_M \mathcal{V} *_M \mathcal{V}^\top - \mathcal{U} *_M \mathcal{U}^\top *_M \mathcal{T} *_M \mathcal{V} *_M \mathcal{V}^\top \tag{C.1}$$

$$\mathcal{P}_{\star\perp}(\mathcal{T}) = (\mathcal{I} - \mathcal{U} *_M \mathcal{U}^\top) *_M \mathcal{T} *_M (\mathcal{I} - \mathcal{V} *_M \mathcal{V}^\top). \tag{C.2}$$

These operators decompose the tensor $\mathcal{T}$ into components aligned with the sub-modules t-spanned by $\mathcal{U}$ and $\mathcal{V}$, and their orthogonal complements, respectively.

As shown in the appendix of Wang et al. (2020), the following properties hold:

a). Any tensor $\mathcal{T} \in \mathbb{R}^{d_1 \times d_2 \times d_3}$ can be uniquely decomposed as $\mathcal{T} = \mathcal{P}_\star(\mathcal{T}) + \mathcal{P}_{\star\perp}(\mathcal{T})$.

b). The inner product between the projections $\mathcal{P}_\star(\mathcal{X})$ and $\mathcal{P}_{\star\perp}(\mathcal{Y})$ is zero, i.e., $\langle \mathcal{P}_\star(\mathcal{X}), \mathcal{P}_{\star\perp}(\mathcal{Y}) \rangle = 0$, for all tensors $\mathcal{X}, \mathcal{Y} \in \mathbb{R}^{d_1 \times d_2 \times d_3}$.

c). The tubal rank of the projected tensor $\mathcal{P}_\star(\mathcal{T})$ is at most twice the rank of $\mathcal{W}_\star^{(0)}$, i.e., $r_t(\mathcal{P}_\star(\mathcal{T})) \leq 2 \cdot r_t(\mathcal{W}_\star^{(0)})$, for all $\mathcal{T} \in \mathbb{R}^{d_1 \times d_2 \times d_3}$.

Additionally, the following properties related to the tubal nuclear norm (TNN) can be established:

a). **(Decomposability of TNN)** For any tensors $\mathcal{X}, \mathcal{Y} \in \mathbb{R}^{d_1 \times d_2 \times d_3}$ satisfying $\mathcal{X} *_M \mathcal{Y}^\top = 0$ and $\mathcal{X}^\top *_M \mathcal{Y} = 0$, the tubal nuclear norm decomposes additively:

$$\|\mathcal{X} + \mathcal{Y}\|_\star = \|\mathcal{P}_\star(\mathcal{X})\|_\star + \|\mathcal{P}_{\star\perp}(\mathcal{Y})\|_\star.$$

b). **(Norm compatibility inequality)** For any tensor $\mathcal{T} \in \mathbb{R}^{d_1 \times d_2 \times d_3}$, the tubal nuclear norm can be related to the tensor Frobenius norm and the tensor rank as follows:

$$\|\mathcal{T}\|_\star \leq \sqrt{r_t(\mathcal{T}) \cdot d_3} \cdot \|\mathcal{T}\|_F.$$

## C.2. Additional Notations

Throughout this appendix, we adopt the following notations: Total sample size: $N = KN_S + N_T$. Let $N_k = N_S$ for $k = 1, \ldots, K$ and $N_T$ for $k = 0$.

Following the conventions in Wainwright (2019), we adopt the following notations. For any task $k = 0, 1, \ldots, K$, define the observation vector $\mathbf{y}^{(k)}$, noise vector $\boldsymbol{\epsilon}^{(k)}$, design operator $\mathfrak{X}^{(k)}(\cdot)$ and the adjoint operator of $\mathfrak{X}^{(k)}(\cdot)$ as follows:

$$
\begin{aligned}
\mathbf{y}^{(k)} &:= (y_1^{(k)}, \cdots, y_{N_k}^{(k)})^\top \in \mathbb{R}^{N_k} \\
\boldsymbol{\epsilon}^{(k)} &:= (\epsilon_1^{(k)}, \cdots, \epsilon_{N_k}^{(k)})^\top \in \mathbb{R}^{N_k} \\
\mathfrak{X}^{(k)}(\boldsymbol{\Delta}) &:= (\langle \mathcal{X}_1^{(k)}, \boldsymbol{\Delta} \rangle, \cdots, \langle \mathcal{X}_{N_k}^{(k)}, \boldsymbol{\Delta} \rangle)^\top \in \mathbb{R}^{N_k}, \quad \forall \boldsymbol{\Delta} \in \mathbb{R}^{d_1 \times d_2 \times d_3} \\
\mathfrak{X}^{*(k)}(\mathbf{z}) &:= \sum_{i=1}^{N_k} z_i \cdot \mathcal{X}_i^{(k)} \in \mathbb{R}^{d_1 \times d_2 \times d_3}, \quad \forall \mathbf{z} \in \mathbb{R}^{N_k}.
\end{aligned}
\tag{C.3}
$$

We use several asymptotic notations to describe the relationships between functions. For the sake of clarity, we provide their definitions here:

- The notation $f(n) \lesssim g(n)$ means that there exists a positive constant $c$ and a positive integer $n_0$ such that for all $n \geq n_0$, we have $f(n) \leq c \cdot g(n)$. This is equivalent to saying $f(n) = O(g(n))$.

- Similarly, $f(n) \gtrsim g(n)$ means that there exists a positive constant $c$ and a positive integer $n_0$ such that for all $n \geq n_0$, we have $f(n) \geq c \cdot g(n)$. This is equivalent to saying $g(n) = O(f(n))$.

- We write $f(n) \asymp g(n)$ if both $f(n) \lesssim g(n)$ and $f(n) \gtrsim g(n)$ hold. This means that $f(n)$ and $g(n)$ are of the same order.

- The notation $f(n) = o(g(n))$ means that for every positive constant $\epsilon$, there exists a positive integer $n_0$ such that for all $n \geq n_0$, we have $|f(n)| \leq \epsilon \cdot |g(n)|$.

These notations allow us to express the asymptotic behavior of functions concisely, which is particularly useful in our analysis of algorithmic complexity and error bounds.

## C.3. Restricted Strong Convexity (RSC) and Restricted Smoothness (RSM)

Note that according to Assumption 3.2, the design tensors $\mathcal{X}_i^{(k)}$ in each task ($k = 0, 1, \ldots, K$) is a random Gaussian design. We first establish a key property of the Gaussian design tensor, which serves as a fundamental building block for our subsequent analysis. Lemma C.6 demonstrates that the least squares objective function possesses the restricted strong convexity (RSC) and restricted smoothness (RSM) properties.

**Lemma C.6** (RSC and RSM). *Under Assumption 3.2, for any* $\boldsymbol{\Delta} \in \mathbb{R}^{d_1 \times d_2 \times d_3}$*, with probability at least* $1 - c_1 \exp(-c_2 N_k)$*,*

$$\text{RSC:} \qquad \frac{1}{N_k} \left\| \mathcal{X}^{(k)}(\boldsymbol{\Delta}) \right\|_2^2 \geq \alpha_k \|\boldsymbol{\Delta}\|_2^2 - \beta_k \frac{d_1}{N_k} \|\boldsymbol{\Delta}\|_\star^2$$

$$\text{RSM:} \qquad \frac{1}{N_k} \left\| \mathcal{X}^{(k)}(\boldsymbol{\Delta}) \right\|_2^2 \leq \gamma_k \|\boldsymbol{\Delta}\|_2^2 + \tau_k \frac{d_1}{N_k} \|\boldsymbol{\Delta}\|_\star^2$$

*where* $\alpha_k \geq c_1 c_x^{-1}, \gamma_k \leq c_2 c_x$ *and* $\beta_k, \tau_k \leq c_3 c_x$*, where* $c_1, c_2, c_3$ *are universal postive constants.*

*Proof of Lemma C.6.* The results can be simply obtained in a similar manner to the proof of Proposition 1 in Raskutti et al. (2011). The only difference is to change the changing the $\ell_1$-norm to tubal nuclear norm, and the upper bound of $\mathbb{E}[\sup_{v \in V(r)} h^T(\Sigma^{1/2} v)]$ by

$$\mathbb{E}[\sup_{\boldsymbol{\Delta} \in \mathbf{V}(\ell)} \langle \mathcal{H}, \sigma_k \boldsymbol{\Delta} \rangle] \leq \ell \sigma_k \mathbb{E}[\|\mathcal{H}\|_{\text{tsp}}] \leq \ell \sigma_k (\sqrt{d_1} + \sqrt{d_2}) \tag{C.4}$$

where $\mathcal{H} \in \mathbb{R}^{d_1 \times d_2 \times d_3}$ is a tensor whose entries are *i.i.d.* standard Gaussian, and

$$\mathbf{V}(\ell) := \{ \boldsymbol{\Delta} \in \mathbb{R}^{d_1 \times d_2 \times d_3} : \sigma_k \|\boldsymbol{\Delta}\|_F = 1, \|\boldsymbol{\Delta}\|_\star \leq \ell \}. \tag{C.5}$$

$\square$

# D. Theoretical Analysis of LoRT

## D.1. Analysis of the Joint Low-rank Learning Step of LoRT

We begin by introducing some key notations and transformations that will facilitate our analysis.

For each source task $k \in [K]$, let $\boldsymbol{\Theta}_\star^{(k)} := \boldsymbol{\mathcal{W}}_\star^{(k)} - \boldsymbol{\mathcal{W}}_\star^{(0)}$ denote the difference between the ground truth parameters of the $k$-th source task and the target task, representing the model shift. We define:

$$\vec{\boldsymbol{\Theta}}_\star = (\boldsymbol{\Theta}_\star^{(0)}, \boldsymbol{\Theta}_\star^{(1)}, \cdots, \boldsymbol{\Theta}_\star^{(K)}) = (\boldsymbol{\mathcal{W}}_\star^{(0)}, \boldsymbol{\mathcal{W}}_\star^{(1)} - \boldsymbol{\mathcal{W}}_\star^{(0)}, \cdots, \boldsymbol{\mathcal{W}}_\star^{(K)} - \boldsymbol{\mathcal{W}}_\star^{(0)}) \in \mathbb{R}^{(K+1) \times d_1 \times d_2 \times d_3}. \qquad \text{(D.1)}$$

Here, $\boldsymbol{\mathcal{W}}_\star^{(0)} \in \mathbb{R}^{d_1 \times d_2 \times d_3}$ is the tensor parameter of the target task model, and $\boldsymbol{\mathcal{W}}_\star^{(k)} \in \mathbb{R}^{d_1 \times d_2 \times d_3}$ is the parameter of the $k$-th source task model for all $k \in [K]$.

We rewrite the loss function as:

$$\mathcal{L}(\vec{\boldsymbol{\Theta}}) := \frac{1}{2N} \left( \|\mathbf{y}^{(0)} - \mathfrak{X}^{(0)}(\boldsymbol{\Theta}^{(0)})\|_2^2 + \sum_{k=1}^{K} \|\mathbf{y}^{(k)} - \mathfrak{X}^{(k)}(\boldsymbol{\Theta}^{(k)} + \boldsymbol{\Theta}^{(0)})\|_2^2 \right) \qquad \text{(D.2)}$$

where we use the change of variable:

$$\vec{\boldsymbol{\Theta}} = (\boldsymbol{\Theta}^{(0)}, \boldsymbol{\Theta}^{(1)}, \cdots, \boldsymbol{\Theta}^{(K)}) = (\boldsymbol{\mathcal{W}}^{(0)}, \boldsymbol{\mathcal{W}}^{(1)} - \boldsymbol{\mathcal{W}}^{(0)}, \cdots, \boldsymbol{\mathcal{W}}^{(K)} - \boldsymbol{\mathcal{W}}^{(0)}) \in \mathbb{R}^{(K+1) \times d_1 \times d_2 \times d_3} \qquad \text{(D.3)}$$

With this change of variables, solving problem (7) is equivalent to solving:

$$\widehat{\vec{\boldsymbol{\Theta}}} = \underset{\vec{\boldsymbol{\Theta}}}{\operatorname{argmin}} \{\mathcal{L}(\vec{\boldsymbol{\Theta}}) + \lambda_0 \mathcal{R}(\vec{\boldsymbol{\Theta}})\} \qquad \text{(D.4)}$$

where $\lambda_0 \mathcal{R}(\vec{\boldsymbol{\Theta}}) := \sum_{k=0}^{K} \lambda_k \|\boldsymbol{\Theta}^{(k)}\|_\star$, and we set $\lambda_k = \lambda_0 a_k$ for all $k \in [K]$ for simplicity.

We define the estimation error as $\vec{\boldsymbol{\Delta}} := \widehat{\vec{\boldsymbol{\Theta}}} - \vec{\boldsymbol{\Theta}}_\star \in \mathbb{R}^{(K+1) \times d_1 \times d_2 \times d_3}$, with the corresponding $k$-th block $\boldsymbol{\Delta}^{(k)} := \widehat{\boldsymbol{\Theta}}^{(k)} - \boldsymbol{\Theta}_\star^{(k)}$, for all $k = 0, 1, \ldots, K$.

To quantify the quality of the weighted averaging estimator $\hat{\boldsymbol{\mathcal{W}}}^{\text{a}}$, we further define:

$$\boldsymbol{\Delta}^{\text{a}} := \boldsymbol{\Delta}^{(0)} + \sum_{k=1}^{K} \frac{N_S}{N} \boldsymbol{\Delta}^{(k)} = \hat{\boldsymbol{\mathcal{W}}}^{\text{a}} - \boldsymbol{\mathcal{W}}_\star^{\text{a}}$$

as the estimation error of the parameter average $\boldsymbol{\mathcal{W}}_\star^{\text{a}}$. Our goal is to establish an upper bound for $\|\boldsymbol{\Delta}^{\text{a}}\|_{\text{F}}^2$.

**Supporting Lemmas**  Theorem 4.2 provides an upper bound on the estimation error of the first step of LoRT. The proof of Theorem 4.2 relies on three key technical lemmas:

**Lemma D.1** (Concentration of Gradient). *Under Assumptions 3.2 and 3.3, if $N_S \gtrsim d_1$, then by choosing*

$$\lambda_0 = c_0 \sqrt{\frac{d_1}{N}}, \quad \lambda_k = a_k \lambda_0 = c_0 \frac{\sqrt{d_1 N_S}}{N}$$

*for some appropriate constant $c_0$, we have the following upper bound for any $\vec{\boldsymbol{\Delta}} = (\boldsymbol{\Delta}^{(0)}, \boldsymbol{\Delta}^{(1)}, \ldots, \boldsymbol{\Delta}^{(K)}) \in \mathbb{R}^{(K+1) \times d_1 \times d_2 \times d_3}$, with high probability:*

$$\left| \langle \nabla \mathcal{L}\left(\vec{\boldsymbol{\Theta}}_\star\right), \vec{\boldsymbol{\Delta}} \rangle \right| \le \sum_{k=0}^{K} \frac{\lambda_k}{2} \|\boldsymbol{\Delta}^{(k)}\|_\star.$$

This lemma bounds the inner product of the gradient of the loss function with the estimation error, which is crucial for controlling the first-order term in our analysis.

**Lemma D.2** (Restricted Set of Directions). *Under Assumptions 3.2 and 3.3, and the conditions of Lemma D.1, if we further assume $\lambda_k \geq 8\lambda_0 \frac{N_S}{N}$ and $N_S > N_T$, then the averaging estimation error $\boldsymbol{\Delta}^{\mathrm{a}}$ satisfies the following inequality with high probability:*

$$2\lambda_0\|\boldsymbol{\Delta}^{(0)}\|_\star + \sum_{k=0}^{K} \lambda_k \|\boldsymbol{\Delta}^{(k)}\|_\star \leq 8\lambda_0 \|\mathcal{P}_\star(\boldsymbol{\Delta}^{\mathrm{a}})\|_\star + 8\sum_{k=1}^{K} \lambda_k h_k$$

*where $\mathcal{P}_\star(\cdot)$ is the operator defined in Eq. (C.1).*

This lemma establishes a restricted set of directions in which the averaging error $\boldsymbol{\Delta}^{\mathrm{a}}$ lies, which is essential for our subsequent analysis.

**Lemma D.3** (Restricted Strong Convexity). *Under Assumptions 3.2 and 3.3 and the conditions of Lemma D.2, the estimation error $\boldsymbol{\Delta}^{\mathrm{a}}$ satisfies with high probability:*

$$\mathcal{L}\left(\vec{\boldsymbol{\Theta}}_\star + \vec{\boldsymbol{\Delta}}\right) - \mathcal{L}\left(\vec{\boldsymbol{\Theta}}_\star\right) - \langle \nabla\mathcal{L}\left(\vec{\boldsymbol{\Theta}}_\star\right), \vec{\boldsymbol{\Delta}}\rangle \geq (1 - u_n)\alpha_{\min}\|\boldsymbol{\Delta}^{\mathrm{a}}\|_{\mathrm{F}}^2 - v_n \sum_{k=1}^{K} \lambda_k h_k \tag{D.5}$$

*where*

$$u_n := \frac{512\beta_{\max}\lambda_0^2}{\alpha_{\min}\lambda_k^2 \wedge (\lambda_0^2/(K+1))} \frac{rd_1d_3}{N}$$

$$v_n := \frac{256\beta_{\max}}{\lambda_k^2 \wedge (\lambda_0^2/(K+1))} \frac{d_1}{N} \left(\sum_{k=1}^{K} \lambda_k h_k\right)$$

$$\alpha_{\min} := \min_{0 \leq k \leq K} \alpha_k, \quad \beta_{\max} := \max_{0 \leq k \leq K} \beta_k$$

*with RSC constants $(\alpha_k, \beta_k)$ defined in Lemma C.6.*

This lemma ensures a property analogous to restricted strong convexity for $\boldsymbol{\Delta}^{\mathrm{a}}$, which is crucial for establishing the convergence of our estimator.

Now, we proceed with the proof of Theorem 4.2.

**Proof of Theorem 4.2** We introduce the function $\mathcal{F} : \mathbb{R}^{(K+1)\times d_1 \times d_2 \times d_3} \to \mathbb{R}$, defined as:

$$\mathcal{F}(\vec{\boldsymbol{\Delta}}) = \mathcal{L}\left(\vec{\boldsymbol{\Theta}}_\star + \vec{\boldsymbol{\Delta}}\right) - \mathcal{L}\left(\vec{\boldsymbol{\Theta}}_\star\right) + \lambda_0 \mathcal{R}\left(\vec{\boldsymbol{\Theta}}_\star + \vec{\boldsymbol{\Delta}}\right) - \lambda_0 \mathcal{R}\left(\vec{\boldsymbol{\Theta}}_\star\right).$$

By applying Lemma D.1, we can establish the following inequality with high probability:

$$\begin{aligned}
\mathcal{F}(\widehat{\vec{\boldsymbol{\Theta}}}) &= \mathcal{L}\left(\vec{\boldsymbol{\Theta}}_\star + \vec{\boldsymbol{\Delta}}\right) - \mathcal{L}\left(\vec{\boldsymbol{\Theta}}_\star\right) + \lambda_0 \mathcal{R}\left(\vec{\boldsymbol{\Theta}}_\star + \vec{\boldsymbol{\Delta}}\right) - \lambda_0 \mathcal{R}\left(\vec{\boldsymbol{\Theta}}_\star\right) \\
&\overset{(i)}{\geq} -\left|\langle \nabla\mathcal{L}\left(\vec{\boldsymbol{\Theta}}_\star\right), \vec{\boldsymbol{\Delta}}\rangle\right| + \mathrm{vec}(\vec{\boldsymbol{\Delta}})^\top \nabla^2\mathcal{L}\left(\vec{\boldsymbol{\Theta}}_\star + \gamma\vec{\boldsymbol{\Delta}}\right)\mathrm{vec}(\vec{\boldsymbol{\Delta}}) \quad (\gamma \in (0,1)) \\
&\quad + \sum_{k=1}^{K} \lambda_k \left(\|\boldsymbol{\Theta}_\star^{(k)} + \boldsymbol{\Delta}^{(k)}\|_\star - \|\boldsymbol{\Theta}_\star^{(k)}\|_\star\right) + \lambda_0\|\boldsymbol{\Theta}_\star^{(0)} + \boldsymbol{\Delta}^{(0)}\|_\star - \lambda_0\|\boldsymbol{\Theta}_\star^{(0)}\|_\star \\
&\overset{(ii)}{\geq} -\sum_{k=1}^{K} \frac{\lambda_k}{2} \left\|\boldsymbol{\Delta}^{(k)}\right\|_\star - \frac{\lambda_0}{2} \left\|\boldsymbol{\Delta}^{(0)}\right\|_\star + \mathrm{vec}(\vec{\boldsymbol{\Delta}})^\top \hat{\boldsymbol{\Sigma}}\mathrm{vec}(\vec{\boldsymbol{\Delta}}) \\
&\quad + \sum_{k=1}^{K} \lambda_k \left(\left\|\boldsymbol{\Delta}^{(k)}\right\|_\star - 2\left\|\boldsymbol{\Theta}^{(k)}\right\|_\star\right) + \lambda_0 \left(\left\|\mathcal{P}_\star(\boldsymbol{\mathcal{W}}_\star^{(0)})\right\|_\star - \left\|\mathcal{P}_\star(\boldsymbol{\Delta}^{(0)})\right\|_\star + \left\|\mathcal{P}_{\star^\perp}(\boldsymbol{\Delta}^{(0)})\right\|_\star - \left\|\mathcal{P}_\star(\boldsymbol{\mathcal{W}}_\star^{(0)})\right\|_\star\right) \\
&\overset{(iii)}{\geq} \mathrm{vec}(\vec{\boldsymbol{\Delta}})^\top \hat{\boldsymbol{\Sigma}}\mathrm{vec}(\vec{\boldsymbol{\Delta}}) + \frac{\lambda_0}{2}\left(\left\|\mathcal{P}_{\star^\perp}(\boldsymbol{\Delta}^{(0)})\right\|_\star - 3\left\|\mathcal{P}_\star(\boldsymbol{\Delta}^{(0)})\right\|_\star\right) + \sum_{k=1}^{K} \frac{\lambda_k}{2}\left\|\boldsymbol{\Delta}^{(k)}\right\|_\star - 2\sum_{k=1}^{K} \lambda_k h_k
\end{aligned}$$

where (i) follows by the mean value theorem with $\gamma \in (0, 1)$; (ii) holds as a result of Lemma D.1 and we also use $\boldsymbol{\Theta}_\star^{(0)} = \boldsymbol{\mathcal{W}}_\star^{(0)}$, $\|\mathcal{P}_{\star\perp}(\boldsymbol{\Theta}_\star^{(0)})\|_\star = 0$, and $\|\boldsymbol{\mathcal{W}}_\star^{(0)} + \boldsymbol{\Delta}\|_\star = \|\boldsymbol{\mathcal{W}}_\star^{(0)} + \mathcal{P}_{\star\perp}(\boldsymbol{\Delta}) + \mathcal{P}_\star(\boldsymbol{\Delta})\|_\star \geq \|\boldsymbol{\mathcal{W}}_\star^{(0)} + \mathcal{P}_{\star\perp}(\boldsymbol{\Delta})\|_\star - \|\mathcal{P}_\star(\boldsymbol{\Delta})\|_\star = \|\boldsymbol{\mathcal{W}}_\star^{(0)}\|_\star + \|\mathcal{P}_{\star\perp}(\boldsymbol{\Delta})\|_\star - \|\mathcal{P}_\star(\boldsymbol{\Delta})\|_\star$ due to the decomposibility of TNN; (iii) holds because $\|\boldsymbol{\Theta}_\star^{(k)}\|_\star \leq h_k$ for $1 \leq k \leq K$. Here, $\hat{\boldsymbol{\Sigma}}$ represents the Hessian matrix of the loss function $\mathcal{L}(\cdot)$ evaluated at the point $(\vec{\boldsymbol{\Theta}}_\star + \gamma\vec{\boldsymbol{\Delta}})$, whose explicit form is given by:

$$\hat{\boldsymbol{\Sigma}} := \frac{1}{N} \begin{pmatrix} \sum_{k=0}^{K}\sum_{i=1}^{N_k} \text{vec}(\boldsymbol{\mathcal{X}}_i^{(k)})\text{vec}(\boldsymbol{\mathcal{X}}_i^{(k)})^\top & \sum_{i=1}^{N_k} \text{vec}(\boldsymbol{\mathcal{X}}_i^{(1)})\text{vec}(\boldsymbol{\mathcal{X}}_i^{(1)})^\top & \cdots & \sum_{i=1}^{N_k} \text{vec}(\boldsymbol{\mathcal{X}}_i^{(K)})\text{vec}(\boldsymbol{\mathcal{X}}_i^{(K)})^\top \\ \sum_{i=1}^{N_k} \text{vec}(\boldsymbol{\mathcal{X}}_i^{(1)})\text{vec}(\boldsymbol{\mathcal{X}}_i^{(1)})^\top & \sum_{i=1}^{N_k} \text{vec}(\boldsymbol{\mathcal{X}}_i^{(1)})\text{vec}(\boldsymbol{\mathcal{X}}_i^{(1)})^\top & \cdots & \mathbf{0} \\ \vdots & \vdots & \vdots & \vdots \\ \sum_{i=1}^{N_k} \text{vec}(\boldsymbol{\mathcal{X}}_i^{(K)})\text{vec}(\boldsymbol{\mathcal{X}}_i^{(K)})^\top & \mathbf{0} & \cdots & \sum_{i=1}^{N_k} \text{vec}(\boldsymbol{\mathcal{X}}_i^{(K)})\text{vec}(\boldsymbol{\mathcal{X}}_i^{(K)})^\top \end{pmatrix}$$
$$\in \mathbb{R}^{(K+1)d_1d_2d_3 \times (K+1)d_1d_2d_3}. \tag{D.6}$$

Leveraging the definition of $\boldsymbol{\Delta}^{\text{a}}$, we have $\boldsymbol{\Delta}^{(0)} = \boldsymbol{\Delta}^{\text{a}} - \sum_{k=1}^{K} \frac{N_S}{N}\boldsymbol{\Delta}^{(k)}$. Applying the triangle inequality yields:

$$\mathcal{F}(\boldsymbol{\Delta}) \geq \text{vec}(\vec{\boldsymbol{\Delta}})^\top \hat{\boldsymbol{\Sigma}}\text{vec}(\vec{\boldsymbol{\Delta}}) + \frac{1}{2}\lambda_0 \|\mathcal{P}_{\star\perp}(\boldsymbol{\Delta}^{\text{a}})\|_\star - \frac{1}{2}\lambda_0 \sum_{k=1}^{K} \frac{N_S}{N} \left\|\mathcal{P}_{\star\perp}(\boldsymbol{\Delta}^{(k)})\right\|_\star$$
$$- \frac{3}{2}\lambda_0 \|\mathcal{P}_\star(\boldsymbol{\Delta}^{\text{a}})\|_\star - \frac{3}{2}\lambda_0 \sum_{k=1}^{K} \frac{N_S}{N} \left\|\mathcal{P}_\star(\boldsymbol{\Delta}^{(k)})\right\|_\star + \sum_{k=1}^{K} \frac{\lambda_k}{2} \left\|\boldsymbol{\Delta}^{(k)}\right\|_\star - 2\sum_{k=1}^{K} \lambda_k h_k.$$

We select $\lambda_0, \ldots, \lambda_k$ such that $\frac{\lambda_k}{8} \geq \frac{3}{2}\frac{N_S}{N}\lambda_0$. This choice leads to:

$$\sum_{k=1}^{K} \frac{\lambda_k}{2} \left\|\boldsymbol{\Delta}^{(k)}\right\|_\star - \frac{3}{2}\lambda_0 \sum_{k=1}^{K} \frac{N_S}{N} \left\|\mathcal{P}_\star(\boldsymbol{\Delta}^{(k)})\right\|_\star - \frac{1}{2}\lambda_0 \sum_{k=1}^{K} \frac{N_S}{N} \left\|\mathcal{P}_{\star\perp}(\boldsymbol{\Delta}^{(k)})\right\|_\star$$
$$\overset{(i)}{\geq} \sum_{k=1}^{K} \frac{3\lambda_k}{8} \left\|\boldsymbol{\Delta}^{(k)}\right\|_\star + \sum_{k=1}^{K} \frac{\lambda_k}{8} \left(\left\|\mathcal{P}_\star(\boldsymbol{\Delta}^{(k)})\right\|_\star - \left\|\mathcal{P}_{\star\perp}(\boldsymbol{\Delta}^{(k)})\right\|_\star\right)$$
$$- \frac{3}{2}\lambda_0 \sum_{k=1}^{K} \frac{N_S}{N} \left\|\mathcal{P}_\star(\boldsymbol{\Delta}^{(k)})\right\|_\star - \frac{1}{2}\lambda_0 \sum_{k=1}^{K} \frac{N_S}{N} \left\|\mathcal{P}_{\star\perp}(\boldsymbol{\Delta}^{(k)})\right\|_\star$$
$$\overset{(ii)}{\geq} \sum_{k=1}^{K} \frac{3\lambda_k}{8} \left\|\boldsymbol{\Delta}^{(k)}\right\|_\star - \left(\frac{\lambda_k}{8} + \frac{1}{2}\lambda_0\right) \sum_{k=1}^{K} \frac{N_S}{N} \left\|\mathcal{P}_{\star\perp}(\boldsymbol{\Delta}^{(k)})\right\|_\star$$
$$\geq 0.$$

Here, (i) follows from the triangle inequality $\|\boldsymbol{\Delta}^{(k)}\|_\star = \|\mathcal{P}_\star(\boldsymbol{\Delta}^{(k)}) + \mathcal{P}_{\star\perp}(\boldsymbol{\Delta}^{(k)})\|_\star \geq \|\mathcal{P}_\star(\boldsymbol{\Delta}^{(k)})\|_\star - \|\mathcal{P}_{\star\perp}(\boldsymbol{\Delta}^{(k)})\|_\star$, and (ii) holds due to our parameter setting $\frac{\lambda_k}{8} \geq \frac{3}{2}\frac{N_S}{N}\lambda_0$ and the fact that $\|\boldsymbol{\Delta}^{(k)}\|_\star \geq \|\mathcal{P}_{\star\perp}(\boldsymbol{\Delta}^{(k)})\|_\star$ since orthogonal projection does not increase singular values.

Given that $\widehat{\vec{\boldsymbol{\Theta}}}$ is the solution to problem (D.4), it follows that $0 \geq \mathcal{F}(\vec{\boldsymbol{\Delta}})$. We can further deduce:

$$0 \geq \text{vec}(\vec{\boldsymbol{\Delta}})^\top \hat{\boldsymbol{\Sigma}}\text{vec}(\vec{\boldsymbol{\Delta}}) - \frac{3}{2}\lambda_0 \|\mathcal{P}_\star(\boldsymbol{\Delta}^{\text{a}})\|_\star + \frac{1}{2}\lambda_0 \|\mathcal{P}_{\star\perp}(\boldsymbol{\Delta}^{\text{a}})\|_\star - 2\sum_{k=1}^{K} \lambda_k h_k.$$

Now, we establish an upper bound for the error measured in Frobenius norm, $\|\boldsymbol{\Delta}^{\text{a}}\|_{\text{F}}$. Applying Lemma D.3, we obtain with high probability:

$$0 \geq (1 - u_n)\alpha_{\min} \|\boldsymbol{\Delta}^{\text{a}}\|_2^2 - \frac{3}{2}\lambda_0 \sqrt{rd_3} \|\boldsymbol{\Delta}^{\text{a}}\|_{\text{F}} - (2 + v_n)\sum_{k=1}^{K} \lambda_k h_k$$

where we use the fact that $\|\mathcal{P}_{\star^\perp}(\boldsymbol{\Delta}^{\mathrm{a}})\|_\star \geq 0$. Rearranging terms, we derive:

$$\|\boldsymbol{\Delta}^{\mathrm{a}}\|_2^2 \lesssim \frac{1}{(1-u_n)\alpha_{\min}} \left(\lambda_0^2 rd_3 + (1+v_n)\sum_{k=1}^K \lambda_k h_k\right).$$

To complete the proof, we need to show the order of $v_n$ and prove that $u_n = o(1)$ under the conditions of Theorem 4.2. Given the assumptions in Theorem 1 and our choice of $\lambda_0, \ldots, \lambda_K$, we have:

$$u_n = \frac{256\beta_{\max}\lambda_0^2}{\alpha_{\min}\lambda_k^2 \wedge (\lambda_0^2/(K+1))} \frac{rd_1 d_3}{N} \lesssim \frac{rd_1 d_3}{N_S} = o(1),$$

$$v_n = \frac{256\beta_{\max}}{\lambda_k^2 \wedge (\lambda_0^2/(K+1))} \frac{d_1}{N} \left(\sum_{k=1}^K \lambda_k h_k\right) \lesssim \sqrt{\frac{K^2 d_1}{N_S}} \bar{h}.$$

Finally, leveraging the definition of $\boldsymbol{\Delta}^{\mathrm{a}}$, we have $\boldsymbol{\Delta}^{(0)} = \boldsymbol{\Delta}^{\mathrm{a}} - \sum_{k=1}^K \frac{N_S}{N} \boldsymbol{\Delta}^{(k)}$. This leads to:

$$\|\boldsymbol{\Delta}^{(0)}\|_{\mathrm{F}}^2 \leq 2\|\boldsymbol{\Delta}^{\mathrm{a}}\|_{\mathrm{F}}^2 + 2\|\sum_{k=1}^K \frac{N_S}{N} \boldsymbol{\Delta}^{(k)}\|_{\mathrm{F}}^2$$

$$\lesssim \|\boldsymbol{\Delta}^{\mathrm{a}}\|_{\mathrm{F}}^2 + \|\sum_{k=1}^K \frac{N_S}{N} \boldsymbol{\Delta}^{(k)}\|_\star^2$$

$$\lesssim \frac{rd_1 d_3}{N} + (1+v_n)\bar{h}\sqrt{\frac{d_1}{N_s}} + \delta_\star^2.$$

This completes the proof of Theorem 4.2.

### D.2. Analysis of the Target-Specific Refinement Step of LoRT

Theorem 4.3 provides an upper bound of the estimation error of the target-specific refinement step of LoRT. We begin by defining two key functions:

$$\tilde{\mathcal{L}}(\mathcal{C}) = \frac{1}{2N_T} \left\| \mathbf{y}^{(0)} - \mathfrak{X}^{(0)}(\hat{\boldsymbol{\mathcal{W}}}^{\mathrm{a}}) - \mathfrak{X}^{(0)}(\mathcal{C}) \right\|_2^2$$

$$\tilde{\mathcal{F}}(\boldsymbol{\Delta}) = \tilde{\mathcal{L}}\left(\mathcal{C}^* + \boldsymbol{\Delta}\right) - \tilde{\mathcal{L}}\left(\mathcal{C}^*\right) + \tilde{\lambda}\left\|\mathcal{C}^* + \boldsymbol{\Delta}\right\|_\star - \tilde{\lambda}\left\|\mathcal{C}^*\right\|_\star.$$

where $\mathcal{C}^* = \boldsymbol{\mathcal{W}}_\star^{(0)} - \boldsymbol{\mathcal{W}}_\star^{\mathrm{a}}$ represents the difference between the target parameter and the averaged parameter.

Let $\boldsymbol{\Delta}^{\mathcal{C}} = \hat{\mathcal{C}} - \mathcal{C}^*$ and recall that $\boldsymbol{\Delta}^{\mathrm{a}} = \hat{\boldsymbol{\mathcal{W}}}_\star^{\mathrm{a}} - \boldsymbol{\mathcal{W}}_\star^{\mathrm{a}}$. We now turn to the proof of Theorem 4.3.

**Proof of Theorem 4.3** Applying Hölder's inequality and the triangle inequality, we obtain:

$$\left\langle \nabla \tilde{\mathcal{L}}\left(\mathcal{C}^*\right), \boldsymbol{\Delta}^{\mathcal{C}} \right\rangle = \frac{1}{N_T} \left\langle \mathfrak{X}^{*(0)} \left[\mathbf{y}^{(0)} - \mathfrak{X}^{(0)}(\boldsymbol{\mathcal{W}}_\star^{(0)}) - \mathfrak{X}^{(0)}(\hat{\boldsymbol{\mathcal{W}}}_\star^{\mathrm{a}} + \mathcal{C}^* - \boldsymbol{\mathcal{W}}_\star^{(0)})\right], \boldsymbol{\Delta}^{\mathcal{C}} \right\rangle$$

$$= \frac{1}{N_T} \left\langle \mathfrak{X}^{*(0)} \left[\boldsymbol{\epsilon}^{(0)} - \mathfrak{X}^{(0)}(\hat{\boldsymbol{\mathcal{W}}}_\star^{\mathrm{a}} - \boldsymbol{\mathcal{W}}_\star^{\mathrm{a}})\right], \boldsymbol{\Delta}^{\mathcal{C}} \right\rangle$$

$$\leq \frac{1}{N_T} \left\| \mathfrak{X}^{*(0)}(\boldsymbol{\epsilon}^{(0)}) \right\|_{\mathrm{tsp}} \|\boldsymbol{\Delta}^{\mathcal{C}}\|_\star + \frac{1}{2}\mathrm{vec}(\boldsymbol{\Delta}^{\mathcal{C}})^\top \hat{\boldsymbol{\Sigma}}^{(0)}\mathrm{vec}(\boldsymbol{\Delta}^{\mathcal{C}}) + \frac{1}{2}\mathrm{vec}(\boldsymbol{\Delta}^{\mathrm{a}})^\top \hat{\boldsymbol{\Sigma}}^{(0)}\mathrm{vec}(\boldsymbol{\Delta}^{\mathrm{a}})$$

where $\hat{\boldsymbol{\Sigma}}^{(0)} = \frac{1}{N_T} \sum_{i=1}^{N_T} \mathrm{vec}(\boldsymbol{\mathcal{X}}_i^{(0)})\mathrm{vec}(\boldsymbol{\mathcal{X}}_i^{(0)})^\top$.

By Lemma D.1, if $N_T \gtrsim d_1$, we can choose $\tilde{\lambda} = c\sqrt{\frac{d_1}{N_T}}$ for some constant $c$ such that $\frac{1}{N_T} \left\| \mathfrak{X}^{*(0)}(\boldsymbol{\epsilon}^{(0)}) \right\|_{\mathrm{tsp}} \leq \frac{\tilde{\lambda}}{2}$ with high

probability. This leads to:

$$\tilde{\mathcal{L}}\left(\boldsymbol{\Delta}^{\mathcal{C}} + \mathcal{C}^*\right) - \tilde{\mathcal{L}}(\mathcal{C}^*) = \left\langle \nabla \tilde{\mathcal{L}}\left(\mathcal{C}^*\right), \boldsymbol{\Delta}^{\mathcal{C}}\right\rangle + \text{vec}\left(\boldsymbol{\Delta}^{\mathcal{C}}\right)^{\top} \hat{\boldsymbol{\Sigma}}^{(0)} \text{vec}(\boldsymbol{\Delta}^{\mathcal{C}})$$

$$\geq -\frac{\tilde{\lambda}}{2} \|\boldsymbol{\Delta}^{\mathcal{C}}\|_{\star} + \frac{1}{2}\text{vec}\left(\boldsymbol{\Delta}^{\mathcal{C}}\right)^{\top} \hat{\boldsymbol{\Sigma}}^{(0)} \text{vec}(\boldsymbol{\Delta}^{\mathcal{C}}) - \frac{1}{2}\text{vec}(\boldsymbol{\Delta}^{\mathtt{a}})^{\top} \hat{\boldsymbol{\Sigma}}^{(0)} \text{vec}(\boldsymbol{\Delta}^{\mathtt{a}}).$$

By the optimality condition of $\hat{\mathcal{C}}$, we have:

$$\begin{aligned}
0 &\geq \tilde{\mathcal{F}}(\boldsymbol{\Delta}^{\mathcal{C}}) \\
&\geq \tilde{\mathcal{L}}\left(\boldsymbol{\Delta}^{\mathcal{C}} + \mathcal{C}^*\right) - \tilde{\mathcal{L}}(\boldsymbol{\Delta}^{\mathcal{C}}) + \tilde{\lambda}\left\|\boldsymbol{\Delta}^{\mathcal{C}} + \mathcal{C}^*\right\|_{\star} - \tilde{\lambda}\|\mathcal{C}^*\|_{\star} \\
&\geq \tilde{\mathcal{L}}\left(\boldsymbol{\Delta}^{\mathcal{C}} + \mathcal{C}^*\right) - \tilde{\mathcal{L}}(\boldsymbol{\Delta}^{\mathcal{C}}) + \tilde{\lambda}\|\boldsymbol{\Delta}^{\mathcal{C}}\|_{\star} - 2\tilde{\lambda}\|\mathcal{C}^*\|_{\star} \\
&\geq \frac{\tilde{\lambda}}{2}\|\boldsymbol{\Delta}^{\mathcal{C}}\|_{\star} + \frac{1}{2}\text{vec}\left(\boldsymbol{\Delta}^{\mathcal{C}}\right)^{\top} \hat{\boldsymbol{\Sigma}}^{(0)}\text{vec}(\boldsymbol{\Delta}^{\mathcal{C}}) - \frac{1}{2}\text{vec}(\boldsymbol{\Delta}^{\mathtt{a}})^{\top}\hat{\boldsymbol{\Sigma}}^{(0)}\text{vec}(\boldsymbol{\Delta}^{\mathtt{a}}) - 2\tilde{\lambda}\|\mathcal{C}^*\|_{\star}.
\end{aligned} \tag{D.7}$$

To proceed, we need the following auxiliary lemma:

**Lemma D.4.** *Under Assumptions 3.2 and 3.3, if $N_S \gtrsim d_1$, $N_S > N_T$, and we choose $\lambda_0 \gtrsim \sqrt{\frac{d_1}{N}}$, $\lambda_k = a_k \lambda_0 \gtrsim \sqrt{\frac{N_S}{N}}\sqrt{\frac{d_1}{N}}$ such that*

$$\lambda_k \geq 12\lambda_0 \frac{N_S}{N},$$

$$u_n = \frac{512\beta_{\max}\lambda_0^2}{\alpha_{\min}\lambda_k^2 \wedge (\lambda_0^2/(K+1))} \frac{r d_1 d_3}{N} = o(1),$$

$$v_n = \frac{256\beta_{\max}}{\lambda_k^2 \wedge (\lambda_0^2/(K+1))} \frac{d_1}{N}\left(\sum_{k=1}^{K}\lambda_k h_k\right) = O(1)$$

*then with high probability:*

$$\|\boldsymbol{\Delta}^{\mathtt{a}}\|_{\text{F}} \lesssim \sqrt{r d_3}\lambda_0 + \sqrt{\sum_{k=1}^{K}\lambda_k h_k}$$

$$\|\boldsymbol{\Delta}^{\mathtt{a}}\|_{\star} \lesssim r d_3\lambda_0 + \sqrt{r d_3}\sqrt{\sum_{k=1}^{K}\lambda_k h_k + \frac{\sum_{k=1}^{K}\lambda_k h_k}{\lambda_0}}.$$

The choice of $\lambda_0$ and $\lambda_k$ depends on the event **A** defined as:

$$\mathbf{A} := \left\{\frac{r d_1 d_3}{N_S} \geq \bar{h}\sqrt{\frac{d_1}{N_T}}\right\}$$

where $\bar{h} := N^{-1} N_S \sum_{k=1}^{K} h_k$.

We now consider two cases:

**Case 1:** When event **A** holds, i.e., $r d_1 d_3/N_S \geq \bar{h}\sqrt{d_1/N_T}$.

In this case, we choose $\lambda_0 = c_1\sqrt{\frac{d_1}{N}}$ and $a_k = 12\sqrt{\frac{N_S}{N}}$.

Applying Lemma D.4, we obtain with high probability:

$$\|\boldsymbol{\Delta}^{\mathtt{a}}\|_{\text{F}} \lesssim \sqrt{\frac{r d_1 d_3}{N}} + \sqrt{\sqrt{\frac{d_1}{N_S}}\bar{h}} \tag{D.8}$$

$$\|\boldsymbol{\Delta}^{\mathtt{a}}\|_{\star} \lesssim r d_3\sqrt{\frac{d_1}{N}} + \sqrt{\sqrt{\frac{d_1}{N_S}}r d_3\bar{h}} + \sqrt{\frac{N}{N_S}}\bar{h}. \tag{D.9}$$

We further consider two sub-cases:

**(i)** If $\frac{1}{2}\mathrm{vec}(\mathbf{\Delta}^{\mathtt{a}})^{\top}\hat{\mathbf{\Sigma}}^{(0)}\mathrm{vec}(\mathbf{\Delta}^{\mathtt{a}}) \geq 2\tilde{\lambda}\|\mathbf{\mathcal{C}}^{*}\|_{\star}$, then from Eq. (D.7):

$$0 \geq \frac{\tilde{\lambda}}{2}\|\mathbf{\Delta}^{\mathtt{e}}\|_{\star} + \frac{1}{2}\mathrm{vec}\left(\mathbf{\Delta}^{\mathtt{e}}\right)^{\top}\hat{\mathbf{\Sigma}}^{(0)}\mathrm{vec}(\mathbf{\Delta}^{\mathtt{e}}) - \mathrm{vec}(\mathbf{\Delta}^{\mathtt{a}})^{\top}\hat{\mathbf{\Sigma}}^{(0)}\mathrm{vec}(\mathbf{\Delta}^{\mathtt{a}}). \tag{D.10}$$

By Lemma C.6 and the condition $N_T \gtrsim d_1$, we have:

$$\mathrm{vec}(\mathbf{\Delta}^{\mathtt{a}})^{\top}\hat{\mathbf{\Sigma}}^{(0)}\mathrm{vec}(\mathbf{\Delta}^{\mathtt{a}}) \leq \gamma_0 \|\mathbf{\Delta}^{\mathtt{a}}\|_{\mathrm{F}}^2 + \tau_0 \frac{d_1}{N_T}\|\mathbf{\Delta}^{\mathtt{a}}\|_{\star}^2$$

for some constants $\gamma_0$ and $\tau_0$. Combining this with Eq. (D.10):

$$\begin{aligned}
\frac{\tilde{\lambda}}{2}\|\mathbf{\Delta}^{\mathtt{e}}\|_{\star} &\leq \mathrm{vec}(\mathbf{\Delta}^{\mathtt{a}})^{\top}\hat{\mathbf{\Sigma}}^{(0)}\mathrm{vec}(\mathbf{\Delta}^{\mathtt{a}}) \leq \gamma_0 \|\mathbf{\Delta}^{\mathtt{a}}\|_{\mathrm{F}}^2 + \tau_0 \frac{d_1}{N_T}\|\mathbf{\Delta}^{\mathtt{a}}\|_{\star}^2 \\
&\lesssim \frac{rd_1 d_3}{N} + (1 + \frac{rd_1 d_3}{N_T})\sqrt{\frac{d_1}{N_S}}\bar{h} + \frac{(rd_1 d_3)^2}{N_T N} + \frac{N}{N_S}\bar{h}^2 \frac{d_1}{N_T}.
\end{aligned} \tag{D.11}$$

Given the assumptions in Theorem 4.3 that $\frac{rd_1 d_3}{N_T} = O(1)$ and $\bar{h}\sqrt{d_1/N_T} = o(1)$, and noting that $(N/N_S)\bar{h}\sqrt{d_1/N_T} \leq (K+1)rd_1 d_3/N_S = O(1)$, we conclude that $\frac{\tilde{\lambda}}{2}\|\mathbf{\Delta}^{\mathtt{e}}\|_{\star} = o_p(1)$.

Applying Lemma C.6 to Eq. (D.10) and choosing $\tilde{\lambda} = c\sqrt{\frac{d_1}{N_T}}$ with $c > \sqrt{2\beta_0}$, we get:

$$\frac{1}{2}\alpha_0\|\mathbf{\Delta}^{\mathtt{e}}\|_{\mathrm{F}}^2 \leq \gamma_0 \|\mathbf{\Delta}^{\mathtt{a}}\|_{\mathrm{F}}^2 + \tau_0 \frac{d_1}{N_T}\|\mathbf{\Delta}^{\mathtt{a}}\|_{\star}^2.$$

This leads to the bound:

$$\|\mathbf{\Delta}^{\mathtt{e}}\|_{\mathrm{F}} \lesssim \sqrt{\frac{rd_1 d_3}{N}} + \sqrt{\sqrt{\frac{d_1}{N_S}}\bar{h}} + \sqrt{\frac{N}{N_S}}\sqrt{\frac{d_1}{N_T}}\bar{h}.$$

**(ii)** If $\frac{1}{2}\mathrm{vec}(\mathbf{\Delta}^{\mathtt{a}})^{\top}\hat{\mathbf{\Sigma}}^{(0)}\mathrm{vec}(\mathbf{\Delta}^{\mathtt{a}}) \leq 2\tilde{\lambda}\|\mathbf{\mathcal{C}}^{*}\|_{\star}$, we have:

$$0 \geq \frac{\tilde{\lambda}}{2}\|\mathbf{\Delta}^{\mathtt{e}}\|_{\star} + \frac{1}{2}\left(\mathbf{\Delta}^{\mathtt{e}}\right)^{\top}\hat{\mathbf{\Sigma}}^{(0)}\mathbf{\Delta}^{\mathtt{e}} - 4\tilde{\lambda}\|\mathbf{\mathcal{C}}^{*}\|_{\star}$$

implying $\|\mathbf{\Delta}^{\mathtt{e}}\|_{\star} \leq 8\|\mathbf{\mathcal{C}}^{*}\|_{\star} \leq 8\bar{h}$.

Applying Lemma C.6 again yields:

$$\begin{aligned}
0 &\geq \frac{\tilde{\lambda}}{2}\|\mathbf{\Delta}^{\mathtt{e}}\|_{\star} + \frac{1}{2}\alpha_0\|\mathbf{\Delta}^{\mathtt{e}}\|_{\mathrm{F}}^2 - \frac{1}{2}\beta_0 \frac{d_1}{N_T}\|\mathbf{\Delta}^{\mathtt{e}}\|_{\star}^2 - 4\tilde{\lambda}\|\mathbf{\mathcal{C}}^{*}\|_{\star} \\
&\geq \frac{1}{2}\alpha_0\|\mathbf{\Delta}^{\mathtt{e}}\|_{\mathrm{F}}^2 - 32\beta_0 \frac{d_1}{N_T}\|\mathbf{\mathcal{C}}^{*}\|_{\star}^2 - 4\tilde{\lambda}\|\mathbf{\mathcal{C}}^{*}\|_{\star}.
\end{aligned}$$

This leads to:

$$\|\mathbf{\Delta}^{\mathtt{e}}\|_{\mathrm{F}} \leq \sqrt{\frac{64\beta_0}{\alpha_0}\frac{d_1}{N_T}\bar{h}^2 + 8\frac{\tilde{\lambda}}{\alpha_0}\bar{h}} \lesssim \sqrt{\frac{d_1}{N_T}}\bar{h} + \sqrt{\sqrt{\frac{d_1}{N_T}}\bar{h}}$$

and

$$\|\mathbf{\Delta}^{\mathtt{e}}\|_{\mathrm{F}} \leq \|\mathbf{\Delta}^{\mathtt{e}}\|_{\star} \leq 8\bar{h}.$$

Given the assumption that $\bar{h}\sqrt{\frac{d_1}{N_T}} = o(1)$, we can conclude:

$$\|\mathbf{\Delta}^{\mathtt{e}}\|_{\mathrm{F}} \lesssim \sqrt{\sqrt{\frac{d_1}{N_T}}\bar{h}} \wedge \bar{h}.$$

Combining the results from both sub-cases, we have with high probability:

$$\left\|\hat{\boldsymbol{\mathcal{W}}}_\star^{\mathrm{a}} + \hat{\boldsymbol{\mathcal{C}}} - \boldsymbol{\mathcal{W}}_\star^{(0)}\right\|_{\mathrm{F}} \leq \left\|\boldsymbol{\Delta}^{\mathrm{e}}\right\|_{\mathrm{F}} + \|\boldsymbol{\Delta}^{\mathrm{a}}\|_{\mathrm{F}}$$

$$\lesssim \sqrt{\frac{rd_1 d_3}{N}} + \sqrt{\sqrt{\frac{d_1}{N_S}}\bar{h}} + \sqrt{K+1}\sqrt{\frac{d_1}{N_T}}\bar{h} + \sqrt{\sqrt{\frac{d_1}{N_T}}\bar{h}} \wedge \bar{h}.$$

Since **A** holds and given the condition $K^2 r d_1 d_3 / N_S = O(1)$, we have:

$$\sqrt{K+1}\sqrt{\frac{d_1}{N_T}}\bar{h} \leq \sqrt{K+1}\frac{rd_1 d_3}{N_S} \leq \sqrt{\frac{(K+1)^2 r d_1 d_3}{N_S}}\sqrt{\frac{rd_1 d_3}{N}} \lesssim \sqrt{\frac{rd_1 d_3}{N}}.$$

This implies:

$$\left\|\hat{\boldsymbol{\mathcal{W}}}_\star^{\mathrm{a}} + \hat{\boldsymbol{\mathcal{C}}} - \boldsymbol{\mathcal{W}}_\star^{(0)}\right\|_{\mathrm{F}} \leq \left\|\boldsymbol{\Delta}^{\mathrm{e}}\right\|_{\mathrm{F}} + \|\boldsymbol{\Delta}^{\mathrm{a}}\|_{\mathrm{F}} \lesssim \sqrt{\frac{rd_1 d_3}{N}} + \sqrt{\sqrt{\frac{d_1}{N_S}}\bar{h}} + \sqrt{\sqrt{\frac{d_1}{N_T}}\bar{h}} \wedge \bar{h}.$$

**Case 2:** When event $\mathbf{A}^c$ holds, i.e., $rd_1 d_3 / N_S \leq \bar{h}\sqrt{d_1/N_T}$.

In this case, we choose:

$$\lambda_0 = c_0 \sqrt{\frac{d_1}{N_S}}, \text{ and } a_k = \frac{12 N_S}{N}.$$

Applying Lemma D.4 again, we have with high probability:

$$\|\boldsymbol{\Delta}^{\mathrm{a}}\|_{\mathrm{F}} \lesssim \sqrt{\frac{rd_1 d_3}{N_S}} + \sqrt{\sqrt{\frac{d_1}{N_S}}\bar{h}} \tag{D.12}$$

$$\|\boldsymbol{\Delta}^{\mathrm{a}}\|_\star \lesssim rd_3\sqrt{\frac{d_1}{N_S}} + \sqrt{\sqrt{\frac{d_1}{N_S}}rd_3\bar{h}} + \bar{h}. \tag{D.13}$$

Plugging these new bounds into the arguments from Case 1 leads to:

$$\left\|\hat{\boldsymbol{\mathcal{W}}}_\star^{\mathrm{a}} + \hat{\boldsymbol{\mathcal{C}}} - \boldsymbol{\mathcal{W}}_\star^{(0)}\right\|_{\mathrm{F}} \lesssim \sqrt{\frac{rd_1 d_3}{N_S}} + \sqrt{\sqrt{\frac{d_1}{N_S}}\bar{h}} + \sqrt{\frac{d_1}{N_T}}\bar{h} + \sqrt{\sqrt{\frac{d_1}{N_T}}\bar{h}} \wedge \bar{h}.$$

Recall that in Theorem 4.3 we assume $\sqrt{d_1/N_T}\bar{h} = o(1)$ and $d_1/N_T = O(1)$. Therefore, in the above bound, the third term has a smaller order compared to the fourth term. Hence, we have:

$$\left\|\hat{\boldsymbol{\mathcal{W}}}_\star^{\mathrm{a}} + \hat{\boldsymbol{\mathcal{C}}} - \boldsymbol{\mathcal{W}}_\star^{(0)}\right\|_{\mathrm{F}} \lesssim \sqrt{\frac{rd_1 d_3}{N_S}} + \sqrt{\sqrt{\frac{d_1}{N_S}}\bar{h}} + \sqrt{\sqrt{\frac{d_1}{N_T}}\bar{h}} \wedge \bar{h}.$$

As we assume $\mathbf{A}^c$ holds in this case, we have $rd_1 d_3 / N_S \leq \bar{h}\sqrt{d_1/N_T}$, which further implies:

$$\left\|\hat{\boldsymbol{\mathcal{W}}}_\star^{\mathrm{a}} + \hat{\boldsymbol{\mathcal{C}}} - \boldsymbol{\mathcal{W}}_\star^{(0)}\right\|_{\mathrm{F}} \lesssim \sqrt{\sqrt{\frac{d_1}{N_T}}\bar{h}}.$$

Combining the results from both cases, we have with high probability:

$$\left\|\hat{\boldsymbol{\mathcal{W}}}_\star^{\mathrm{a}} + \hat{\boldsymbol{\mathcal{C}}} - \boldsymbol{\mathcal{W}}_\star^{(0)}\right\|_{\mathrm{F}} \lesssim \sqrt{\frac{rd_1 d_3}{N}} + \sqrt{\sqrt{\frac{d_1}{N_T}}\bar{h}}.$$

This completes the proof of Theorem 4.3.

## D.3. Proof of Supporting Lemmas in the Analysis of LoRT

**Proof of Lemma D.1**   By the definition of the loss function, we have:

$$-\nabla\mathcal{L}\left(\vec{\boldsymbol{\Theta}}\right) = \frac{1}{N}\left(\sum_{k=0}^{K}\mathfrak{X}^{*(k)}(\boldsymbol{\epsilon}^{(k)}), \mathfrak{X}^{*(1)}(\boldsymbol{\epsilon}^{(1)}), \ldots, \mathfrak{X}^{*(K)}(\boldsymbol{\epsilon}^{(K)})\right) \in \mathbb{R}^{(K+1)\times d_1\times d_2\times d_3}.$$

Applying Hölder's inequality, we obtain:

$$|\langle\nabla\mathcal{L}(\vec{\boldsymbol{\Theta}}),\vec{\boldsymbol{\Delta}}\rangle| \leq \frac{1}{N}\sum_{k=1}^{K}|\left\langle\mathfrak{X}^{*(k)}(\boldsymbol{\epsilon}^{(k)}),\boldsymbol{\Delta}^{(k)}\right\rangle| + \frac{1}{N}\left|\left\langle\sum_{k=0}^{K}\mathfrak{X}^{*(k)}(\boldsymbol{\epsilon}^{(k)}),\boldsymbol{\Delta}^{(0)}\right\rangle\right|$$

$$\leq \frac{1}{N}\sum_{k=1}^{K}\|\mathfrak{X}^{*(k)}(\boldsymbol{\epsilon}^{(k)})\|_{\text{tsp}}\|\boldsymbol{\Delta}^{(k)}\|_{\star} + \frac{1}{N}\|\sum_{k=0}^{K}\mathfrak{X}^{*(k)}(\boldsymbol{\epsilon}^{(k)})\|_{\text{tsp}}\|\boldsymbol{\Delta}^{(0)}\|_{\star}.$$

We frist focus on bounding $\|\mathfrak{X}^{*(k)}(\boldsymbol{\epsilon}^{(k)})\|_{\text{tsp}}$. According to the definition of tensor spectral norm

$$\|\mathfrak{X}^{*(k)}(\boldsymbol{\epsilon}^{(k)})\|_{\text{tsp}} = \|\sum_{i=1}^{N_S}\epsilon_i^{(k)}\boldsymbol{\mathcal{X}}_i^{(k)}\|_{\text{tsp}}$$

$$= \max_{\ell\in[d_3]}\left\|M(\sum_{i=1}^{N_S}\epsilon_i^{(k)}\boldsymbol{\mathcal{X}}_i^{(k)})_{:,:,\ell}\right\|_{\text{sp}}$$

$$= \max_{\ell\in[d_3]}\left\|\sum_{i=1}^{N_S}\epsilon_i^{(k)}\cdot M(\boldsymbol{\mathcal{X}}_i^{(k)})_{:,:,\ell}\right\|_{\text{sp}}.$$

Let $\mathbf{Z}^{(i,\ell)} = M(\boldsymbol{\mathcal{X}}_i^{(k)})_{:,:,\ell}$ for all $\ell\in[d_3]$. By the definition of $M$-transform:

$$(\mathbf{Z}^{(i,\ell)})_{j_1 j_2} = \sum_{j_3=1}^{d_3}(\boldsymbol{\mathcal{X}}_i^{(k)})_{j_1 j_2 j_3}\mathbf{M}_{\ell j_3}, \quad \forall(j_1,j_2)\in[d_1]\times[d_2].$$

According to Assumption 3.2, $(\boldsymbol{\mathcal{X}}_i^{(k)})_{j_1 j_2 j_3}$ are i.i.d. drawn from $\mathcal{N}(0,\sigma_k^2)$, so $(\mathbf{Z}^{(i,\ell)})_{j_1 j_2}\sim\mathcal{N}(0,\sigma_k^2)$.

Now, we need to bound:

$$\left\|\sum_{i=1}^{N_S}\epsilon_i^{(k)}\cdot\mathbf{Z}^{(i,\ell)}\right\|_{\text{sp}}.$$

Let $\boldsymbol{\epsilon}^{(k)} = (\epsilon_1^{(k)},\cdots,\epsilon_{N_S}^{(k)})^{\top}\in\mathbb{R}^{N_S}$. Define event $\mathbf{E}$ as $\{\|\boldsymbol{\epsilon}^{(k)}\|^2 < 2c_\epsilon^2 N_S\}$. From Example 2.11 in Wainwright (2019):

$$\mathbb{P}[\|\boldsymbol{\epsilon}^{(k)}\|^2 \geq 2c_\epsilon^2 N_S] \leq 2\exp(-N_S/8).$$

Let $\mathbf{A} = \sum_{i=1}^{N_k}\epsilon_i^{(k)}\cdot\mathbf{Z}^{(i,\ell)}$. Let $\{\mathbf{p}_1,\ldots,\mathbf{p}_I\}$ and $\{\mathbf{q}_1,\ldots,\mathbf{q}_J\}$ be 1/4-covers in Euclidean norm of $\mathbb{S}^{d_1-1}$ and $\mathbb{S}^{d_2-1}$ respectively, with $I\leq 9^{d_1}$ and $J\leq 9^{d_2}$ (Wainwright, 2019).

For any $\mathbf{q}\in\mathbb{S}^{d_2-1}$, we can write $\mathbf{q}_b + \mathbf{z}$ for some vector $\mathbf{z}$ with $\ell_2$ distance at most $1/4$ according to the definition of 1/4-cover:

$$\|\mathbf{A}\|_{\text{sp}} \leq \sup_{\mathbf{q}\in\mathbb{S}^{d_2-1}}\|\mathbf{A}\mathbf{q}\|_2 \leq \sup_{\mathbf{q}\in\mathbb{S}^{d_2-1}}\|\mathbf{A}(\mathbf{q}_b + \mathbf{z})\|_2$$

$$\leq \sup_{b\in[J]}\|\mathbf{A}\mathbf{q}_b\|_2 + \|\mathbf{A}\|_{\text{sp}}\|\mathbf{z}\|_2 \leq \sup_{b\in[J]}\|\mathbf{A}\mathbf{q}_b\|_2 + \frac{1}{4}\|\mathbf{A}\|_{\text{sp}}.$$

Similarly for $\mathbb{S}^{d_1-1}$:

$$\|\mathbf{A}\mathbf{q}_b\|_2 \leq \max_{a\in[I]} \|\mathbf{p}_a^\top \mathbf{A}\mathbf{q}_b\|_2 + \frac{1}{4}\|\mathbf{A}\mathbf{q}_b\|_2.$$

Thus, we have

$$\|\mathbf{A}\|_{\mathrm{sp}} \leq 2\max_{b\in[J]}\max_{a\in[I]} \varrho_{a,b}$$

where $\varrho_{a,b} = \mathbf{p}_a^\top \mathbf{A}\mathbf{q}^b$. Then, for fixed $a,b,i$:

$$\mathrm{Var}(\mathbf{p}_a^\top \epsilon_i^{(k)}\mathbf{Z}^{(i,l)}\mathbf{q}_b) = \mathbb{E}[(\mathbf{p}_a^\top \epsilon_i^{(k)}\mathbf{Z}^{(i,l)}\mathbf{q}_b)^2] - \mathbb{E}[\mathbf{p}_a^\top \epsilon_i^{(k)}\mathbf{Z}^{(i,l)}\mathbf{q}_b]^2$$
$$= \mathbb{E}[(\mathbf{p}_a^\top \mathbf{Z}^{(i,l)}\mathbf{q}_b)^2]\mathbb{E}[(\epsilon_i^{(k)})^2] \leq c_x^2 \mathbb{E}[(\epsilon_i^{(k)})^2].$$

Conditioning on event **E**, we obtain:

$$\mathrm{Var}(\varrho_{ab}|\mathbf{E}) = \sum_{i=1}^{N_k} \mathrm{Var}(\mathbf{p}_a^\top \epsilon_i^{(k)}\mathbf{Z}^{(i,l)}\mathbf{q}_b|\mathbf{E})$$
$$\leq \sum_{i=1}^{N_k} c_x^2 \mathbb{E}[(\epsilon_i^{(k)})^2|\mathbf{E}]$$
$$\leq c_x^2 \mathbb{E}\left[\sum_{i=1}^{N_k}(\epsilon_i^{(k)})^2|\mathbf{E}\right] \leq 2c_x^2 c_\epsilon^2 N_S.$$

So $\varrho_{a,b}$ is zero-mean Gaussian with variance at most $2c_x^2 c_\epsilon^2 N_S$ conditioning on **E**:

$$\mathbb{P}\left[\|\mathbf{A}\|_{\mathrm{sp}} \geq t \Big| \mathbf{E}\right] \leq \sum_{b\in[J]}\sum_{a\in[I]} \mathbb{P}\left[|\varrho_{ab}| \geq \frac{t}{2}\Big|\mathbf{E}\right]$$
$$\leq 2IJ\exp(-\frac{t^2}{16c_x^2 c_\epsilon^2 N_S})$$
$$\leq 2\exp\left(-\frac{t^2}{16c_x^2 c_\epsilon^2 N_S} - (d_1+d_2)\ln 9\right).$$

Recall that we assume $d_1 \geq d_2$. Setting $t = cc_xc_\epsilon\sqrt{N_k d_1}$ with sufficiently large $c$, we have $\left\|\sum_{i=1}^{N_k}\epsilon_i^{(k)}\cdot \mathbf{Z}^{(i,\ell)}\right\|_{\mathrm{sp}} = \|\mathbf{A}\|_{\mathrm{sp}} \leq cc_xc_\epsilon\sqrt{d_1 N_S}$ with probability at least $1 - c_1\exp(-c_2 d_1)$. Taking a union bound with $\ell \in [d_3]$:

$$\|\mathfrak{X}^{*(k)}(\boldsymbol{\epsilon}^{(k)})\|_{\mathrm{tsp}} = \max_{\ell\in[d_3]}\left\|\sum_{i=1}^{N_k}\epsilon_i^{(k)}\cdot \mathbf{Z}^{(i,\ell)}\right\|_{\mathrm{sp}} \leq c\sqrt{d_1 N_k}$$

with probability at least $1 - c_1\exp(-c_2 N_k) - c_3\exp(-c_4 d_1 + \log d_3)$, where $c = c'c_xc_\epsilon$ is a universal constant. Similarly, we can obtain the following bound, *w.h.p.*:

$$\|\sum_{k=0}^{K}\mathfrak{X}^{*(k)}(\boldsymbol{\epsilon}^{(k)})\|_{\mathrm{tsp}} = \|\sum_{k=0}^{K}\sum_{i=1}^{N_k}\epsilon_i^{(k)}\mathfrak{X}_i^{(k)}\|_{\mathrm{tsp}} \lesssim \sqrt{d_1 N}.$$

Therefore, choosing $a_k\lambda_0 = c_0\sqrt{\frac{N_S}{N}}\sqrt{\frac{d_1}{N}}$ and $\lambda_0 = c_0\sqrt{\frac{d_1}{N}}$ for sufficiently large $c_0$ gives the desired result.

**Proof of Lemma D.3** This lemma establishes a key inequality that is crucial for our convergence analysis. We begin by expanding the difference in the loss function:

$$\mathcal{L}\left(\vec{\mathbf{\Theta}}_\star + \vec{\mathbf{\Delta}}\right) - \mathcal{L}\left(\vec{\mathbf{\Theta}}_\star\right) - \left\langle \nabla\mathcal{L}\left(\vec{\mathbf{\Theta}}_\star\right), \vec{\mathbf{\Delta}}\right\rangle = \mathrm{vec}(\vec{\mathbf{\Delta}})^\top \nabla^2\mathcal{L}\left(\vec{\mathbf{\Theta}}_\star + \gamma\vec{\mathbf{\Delta}}\right)\mathrm{vec}(\vec{\mathbf{\Delta}}) \quad (\gamma \in (0,1)).$$

This expansion uses the mean value theorem, with $\gamma$ representing some point between 0 and 1.

Next, we decompose the Hessian term:

$$\mathrm{vec}(\vec{\mathbf{\Delta}})^\top \nabla^2\mathcal{L}\left(\vec{\mathbf{\Theta}}_\star + \gamma\vec{\mathbf{\Delta}}\right)\mathrm{vec}(\vec{\mathbf{\Delta}}) = \sum_{k=1}^{K} \frac{N_S}{N}\mathrm{vec}(\mathbf{\Delta}^{(k)})^\top \hat{\mathbf{\Sigma}}^{(k)}\mathrm{vec}(\mathbf{\Delta}^{(k)}) + 2\sum_{k=1}^{K} \frac{N_S}{N}\mathrm{vec}(\mathbf{\Delta}^{(k)})^\top \hat{\mathbf{\Sigma}}^{(k)}\mathrm{vec}(\mathbf{\Delta}^{(0)})$$

$$+ \mathrm{vec}\left(\mathbf{\Delta}^{(0)}\right)^\top \left(\sum_{k=1}^{K} \frac{N_S}{N}\hat{\mathbf{\Sigma}}^{(k)} + \frac{N_T}{N}\hat{\mathbf{\Sigma}}^{(0)}\right)\mathrm{vec}(\mathbf{\Delta}^{(0)}).$$

This decomposition separates the contributions from source tasks and the target task. We can simplify the RHS as:

$$\sum_{k=1}^{K} \frac{N_S}{N}\mathrm{vec}\left(\mathbf{\Delta}^{(k)} + \mathbf{\Delta}^{(0)}\right)^\top \hat{\mathbf{\Sigma}}^{(k)}\mathrm{vec}\left(\mathbf{\Delta}^{(k)} + \mathbf{\Delta}^{(0)}\right) + \frac{N_T}{N}\mathrm{vec}\left(\mathbf{\Delta}^{(0)}\right)^\top \hat{\mathbf{\Sigma}}^{(0)}\mathrm{vec}(\mathbf{\Delta}^{(0)}).$$

Now, we apply the Restricted Strong Convexity (RSC) condition from Lemma C.6:

$$\sum_{k=1}^{K} \frac{N_S}{N}\mathrm{vec}\left(\mathbf{\Delta}^{(k)} + \mathbf{\Delta}^{(0)}\right)^\top \hat{\mathbf{\Sigma}}^{(k)}\mathrm{vec}\left(\mathbf{\Delta}^{(k)} + \mathbf{\Delta}^{(0)}\right) + \frac{N_T}{N}\mathrm{vec}\left(\mathbf{\Delta}^{(0)}\right)^\top \hat{\mathbf{\Sigma}}^{(0)}\mathrm{vec}(\mathbf{\Delta}^{(0)})$$

$$\geq \sum_{k=1}^{K} \frac{N_S\alpha_k}{N}\left\|\mathbf{\Delta}^{(k)} + \mathbf{\Delta}^{(0)}\right\|_2^2 + \frac{N_T\alpha_0}{N}\left\|\mathbf{\Delta}^{(0)}\right\|_2^2 - \mathcal{R}'\left(\vec{\mathbf{\Delta}}\right).$$

Here, $\alpha_k$ are the RSC constants, and $\mathcal{R}'(\mathbf{\Delta})$ is a remainder term defined as:

$$\mathcal{R}'\left(\vec{\mathbf{\Delta}}\right) := \sum_{k=1}^{K} \frac{N_S\beta_k}{N}\frac{d_1}{N_S}\left\|\mathbf{\Delta}^{(k)} + \mathbf{\Delta}^{(0)}\right\|_\star^2 + \frac{N_T\beta_0}{N}\frac{d_1}{N_T}\left\|\mathbf{\Delta}^{(0)}\right\|_\star^2.$$

We can further lower bound this expression using $\alpha_{\min} := \min_{0 \leq k \leq K} \alpha_k$, which leads to:

$$\mathrm{vec}(\vec{\mathbf{\Delta}})^\top \nabla^2\mathcal{L}\left(\vec{\mathbf{\Theta}}_\star + \gamma\vec{\mathbf{\Delta}}\right)\mathrm{vec}(\vec{\mathbf{\Delta}}) \geq \alpha_{\min}\left\|\sum_{k=1}^{K} \frac{N_S}{N}\mathbf{\Delta}^{(k)} + \mathbf{\Delta}^{(0)}\right\|_2^2 - \mathcal{R}'(\mathbf{\Delta})$$

$$= \alpha_{\min}\|\mathbf{\Delta}^{\mathrm{a}}\|_2^2 - \mathcal{R}'(\mathbf{\Delta}).$$

The key to completing the proof is to bound $\mathcal{R}'\left(\vec{\mathbf{\Delta}}\right)$. We start by applying the triangle inequality:

$$\mathcal{R}'\left(\vec{\mathbf{\Delta}}\right) \leq \sum_{k=1}^{K} \frac{2\beta_k d_1}{N}\left\|\mathbf{\Delta}^{(k)}\right\|_\star^2 + \sum_{k=0}^{K} \frac{2\beta_k d_1}{N}\left\|\mathbf{\Delta}^{(0)}\right\|_\star^2.$$

Next, we use the restricted set of directions from (D.5):

$$\sum_{k=1}^{K} \lambda_k\left\|\mathbf{\Delta}^{(k)}\right\|_\star + \lambda_0\left\|\mathbf{\Delta}^{(0)}\right\|_\star \leq 8\lambda_0\|\mathcal{P}_\star(\mathbf{\Delta}^{\mathrm{a}})\|_\star + 8\sum_{k=1}^{K} \lambda_k h_k.$$

This allows us to bound $\mathcal{R}'(\mathbf{\Delta})$ in terms of $\|\mathcal{P}_\star(\mathbf{\Delta}^{\mathrm{a}})\|_\star$ and $\sum_{k=1}^{K} \lambda_k h_k$. After some algebraic manipulation, we arrive at our final bound:

$$\mathcal{L}\left(\vec{\mathbf{\Theta}}_\star + \vec{\mathbf{\Delta}}\right) - \mathcal{L}\left(\vec{\mathbf{\Theta}}_\star\right) - \left\langle \nabla\mathcal{L}\left(\vec{\mathbf{\Theta}}_\star\right), \vec{\mathbf{\Delta}}\right\rangle \geq (1 - u_n)\alpha_{\min}\|\mathbf{\Delta}^{\mathrm{a}}\|_F^2 - v_n\sum_{k=1}^{K} \lambda_k h_k$$

where

$$u_n = \frac{512\beta_{\max}\lambda_0^2}{\alpha_{\min}\lambda_k^2 \wedge (\lambda_0^2/(K+1))} \frac{rd_1 d_3}{N}, \quad \text{and} \quad v_n = \frac{256\beta_{\max}}{\lambda_k^2 \wedge (\lambda_0^2/(K+1))} \frac{d_1}{N} \left( \sum_{k=1}^{K} \lambda_k h_k \right).$$

This inequality holds with high probability, completing our proof. The terms $u_n$ and $v_n$ capture the impact of problem parameters on our bound, with $u_n$ related to the dimension and rank of our problem, and $v_n$ related to the heterogeneity across tasks.

**Proof of Lemma D.4**   This lemma establishes an upper bound on the nuclear norm of the estimation error. First, note that the bound in Frobenius norm follows directly from the proof of Theorem 4.2. We won't repeat that proof here.

For the nuclear norm bound, we consider two cases based on the relationship between $\frac{1}{4}\lambda_0 \|\mathbf{\Delta}^{\mathrm{a}}\|_\star$ and $2\sum_{k=1}^{K} \lambda_k h_k$.

**Case 1:** $\frac{1}{4}\lambda_0 \|\mathbf{\Delta}^{\mathrm{a}}\|_\star > 2\sum_{k=1}^{K} \lambda_k h_k$.

In this case, we have:

$$0 \geq \mathbf{\Delta}^\top \hat{\mathbf{\Sigma}} \mathbf{\Delta} - \frac{7}{4}\lambda_0 \|\mathbf{\Delta}^{\mathrm{a}}\|_\star + 2\lambda_0 \|\mathbf{\Delta}^{\mathrm{a}}_{S^c}\|_\star \geq -\frac{7}{4}\lambda_0 \|\mathbf{\Delta}^{\mathrm{a}}\|_\star + 2\lambda_0 \|\mathbf{\Delta}^{\mathrm{a}}_{S^c}\|_\star.$$

This implies:

$$\frac{1}{4}\|\mathcal{P}_{\star^\perp}(\mathbf{\Delta}^{\mathrm{a}})\|_\star \leq \frac{7}{4}\|\mathcal{P}_\star(\mathbf{\Delta}^{\mathrm{a}})\|_\star.$$

Using the relationship between tubal nuclear norm and Frobenius norm for tensors, we get:

$$\|\mathbf{\Delta}^{\mathrm{a}}\|_\star \leq 8\|\mathcal{P}_\star(\mathbf{\Delta}^{\mathrm{a}})\|_\star \leq 8\sqrt{2rd_3}\|\mathcal{P}_\star(\mathbf{\Delta}^{\mathrm{a}})\|_{\mathrm{F}} \leq 16\sqrt{rd_3}\|\mathbf{\Delta}^{\mathrm{a}}\|_{\mathrm{F}}.$$

Therefore, in this case, we have *w.h.p.*:

$$\|\mathbf{\Delta}^{\mathrm{a}}\|_\star \lesssim rd_3\lambda_0 + \sqrt{rd_3}\sqrt{\sum_{k=1}^{K} \lambda_k h_k}.$$

**Case 2:** $\frac{1}{4}\lambda_0 \|\mathbf{\Delta}^{\mathrm{a}}\|_\star \leq 2\sum_{k=1}^{K} \lambda_k h_k$.

In this case, we directly obtain:

$$\|\mathbf{\Delta}^{\mathrm{a}}\|_\star \leq \frac{8\sum_{k=1}^{K} \lambda_k h_k}{\lambda_0}.$$

Combining both cases, we conclude that *w.h.p.*:

$$\|\mathbf{\Delta}^{\mathrm{a}}\|_\star \lesssim rd_3\lambda_0 + \sqrt{rd_3}\sqrt{\sum_{k=1}^{K} \lambda_k h_k} + \frac{\sum_{k=1}^{K} \lambda_k h_k}{\lambda_0}. \tag{D.14}$$

This bound captures the impact of the regularization parameters $\lambda_k$, the task heterogeneity $h_k$, and the problem dimensions on the nuclear norm of the estimation error.

# E. Theoretical Analysis of D-LoRT

In this section, we delve into the theoretical analysis of our Distributed Low-Rank Tensor Transitions (D-LoRT) method. We begin with a key theorem that establishes the convergence rate of the one-step D-LoRT estimator.

**Theorem E.1** (One-step D-LoRT). *Let Assumptions 3.2 and 3.3 hold. Further, assume:*

- $N_S \gg K r^2 d_1 d_3^2$ *(sufficiently large source sample size).*

- $N_S \gtrsim (\bar{h}^2 \vee K^2) r d_1 d_3$ *(sample size dominates task heterogeneity).*

- $h_k \asymp \bar{h} = O(1)$ *(bounded task differences).*

*Then, if we choose $\tilde{\lambda}_k = c_1 \sqrt{d_1/N}$ and use the debiased estimator in Eq. (E.2), with high probability:*

$$\|\hat{\boldsymbol{\mathcal{W}}}_{\mathsf{d}}^{\mathsf{a}} - \boldsymbol{\mathcal{W}}_{\star}^{(0)}\|_{\mathrm{F}}^2 \lesssim \frac{r d_1 d_3}{N} + \bar{h}\sqrt{\frac{d_1}{N_S}} + \delta_{\star}^2.$$

This theorem provides an upper bound on the estimation error of D-LoRT. The bound consists of three terms: *(a)* A standard low-rank estimation error term, *(b)* A term reflecting the impact of task heterogeneity, and *(c)* A term capturing the alignment of source tasks.

Now, let's examine how we construct our distributed local parameters.

**The Choice of Distributed Local Parameters $\tilde{\boldsymbol{\mathcal{W}}}^{(k)}$** A crucial step in D-LoRT is the construction of local estimators. We begin with an intuitive approach:

$$\hat{\boldsymbol{\mathcal{W}}}_{\mathrm{TNN}}^{(k)} \in \underset{\boldsymbol{\mathcal{W}}}{\arg\min} \left\{ \frac{1}{2N_S} \left\| \mathbf{y}^{(k)} - \mathfrak{X}^{(k)}(\boldsymbol{\mathcal{W}}) \right\|_2^2 + \tilde{\lambda}_k \|\boldsymbol{\mathcal{W}}\|_{\star} \right\}. \tag{E.1}$$

This estimator, based on the tubal nuclear norm (TNN), provides a good starting point. Its properties are characterized by the following lemma:

**Lemma E.2.** *Under Assumptions 3.2 and 3.3, and $\frac{r d_1 d_3}{N_S} = o(1)$, if we construct $\{\hat{\boldsymbol{\mathcal{W}}}_{\star\mathrm{TNN}}^{(k)}\}_{k=1,\ldots,K}$ through Eq. (E.2), with parameters $\tilde{\lambda}_k = c_0 \sqrt{\frac{d_1}{N_S}}$ for some universal constant $c_0$, then we have that for $k = 1, \ldots, K$, w.h.p.*

$$\left\| \hat{\boldsymbol{\mathcal{W}}}_{\star\mathrm{TNN}}^{(k)} - \boldsymbol{\mathcal{W}}_{\star}^{(k)} \right\|_{\star} \leq r d_3 \sqrt{\frac{d_1}{N_S}} + h_k.$$

However, the TNN estimator is biased. Simply aggregating these local estimators would reduce variance but not address the bias. To overcome this limitation, we introduce a debiasing step inspired by He et al. (2024a):

$$\tilde{\boldsymbol{\mathcal{W}}}^{(k)} = \hat{\boldsymbol{\mathcal{W}}}_{\mathrm{TNN}}^{(k)} + \frac{1}{N_S} \hat{\boldsymbol{\Theta}}^{(k)} \mathfrak{X}^{*(k)} \left( \mathbf{y}^{(k)} - \mathfrak{X}^{(k)} \hat{\boldsymbol{\mathcal{W}}}_{\star\mathrm{TNN}}^{(k)} \right). \tag{E.2}$$

Here, $\hat{\boldsymbol{\Theta}}^{(k)} \in \mathbb{R}^{d_1 d_2 d_3 \times d_1 d_2 d_3}$ approximates $\left( \boldsymbol{\Sigma}^{(k)} \right)^{-1}$, where $\boldsymbol{\Sigma}^{(k)} := \sum_{i=1}^{N_k} \mathrm{vec}(\mathfrak{X}_i^{(k)}) \mathrm{vec}(\mathfrak{X}_i^{(k)})^{\top}$. For this approximation to be effective, $\hat{\boldsymbol{\Theta}}^{(k)}$ must satisfy two key conditions:

- **Condition 1:** $\|\hat{\boldsymbol{\Theta}}^{(k)}\|_{\mathrm{tsp} \to \mathrm{tsp}} \leq C$ (Bounded operator norm).

- **Condition 2:** $\|\hat{\boldsymbol{\Theta}}^{(k)} \hat{\boldsymbol{\Sigma}}^{(k)} - \mathbf{I}\|_{\star \to \mathrm{tsp}} \leq \sqrt{d_1/N_S}$ (Close approximation of inverse).

To gain insight into this debiasing process, we can rewrite Eq. (E.2) as:

$$\tilde{\boldsymbol{\mathcal{W}}}^{(k)} - \boldsymbol{\mathcal{W}}_{\star}^{(k)} = \underbrace{\frac{1}{N_S} \hat{\boldsymbol{\Theta}}^{(k)} \mathfrak{X}^{*(k)}(\boldsymbol{\epsilon}^{(k)})}_{\text{variance term}} - \underbrace{\left( \hat{\boldsymbol{\Theta}}^{(k)} \hat{\boldsymbol{\Sigma}}^{(k)} - \mathbf{I} \right) \left( \hat{\boldsymbol{\mathcal{W}}}_{\star\mathrm{TNN}}^{(k)} - \boldsymbol{\mathcal{W}}_{\star}^{(k)} \right)}_{\text{bias term}}. \tag{E.3}$$

This decomposition allows us to analyze the variance and bias separately. Let's examine each term:

1. Variance term: Using Condition 1, we can bound this term as follows:

$$\|\frac{1}{N_S}\hat{\boldsymbol{\Theta}}^{(k)}\mathfrak{X}^{(k)}(\mathcal{E}^{(k)})\|_{\text{tsp}} \leq \frac{1}{N_S}\|\hat{\boldsymbol{\Theta}}^{(k)}\|_{\text{tsp}\to\text{tsp}}\|\mathfrak{X}^{(k)}(\mathcal{E}^{(k)})\|_{\text{tsp}} \lesssim \sqrt{\frac{d_1}{N_S}}.$$

2. Bias term: Applying Hölder's inequality and Condition 2, we get:

$$\left\|\tilde{\boldsymbol{\mathcal{B}}}^{(k)}\right\|_{\text{tsp}} = \left\|\left(\hat{\boldsymbol{\Theta}}^{(k)}\hat{\boldsymbol{\Sigma}}^{(k)} - \mathbf{I}\right)\left(\hat{\boldsymbol{\mathcal{W}}}_{\star\text{TNN}}^{(k)} - \boldsymbol{\mathcal{W}}_{\star}^{(k)}\right)\right\|_{\text{tsp}}$$

$$\leq \left\|\hat{\boldsymbol{\Theta}}^{(k)}\hat{\boldsymbol{\Sigma}}^{(k)} - \mathbf{I}\right\|_{\star\to\text{tsp}}\left\|\hat{\boldsymbol{\mathcal{W}}}_{\star\text{TNN}}^{(k)} - \boldsymbol{\mathcal{W}}_{\star}^{(k)}\right\|_{\star} \lesssim \sqrt{\frac{d_1}{N_S}}\left\|\hat{\boldsymbol{\mathcal{W}}}_{\star\text{TNN}}^{(k)} - \boldsymbol{\mathcal{W}}_{\star}^{(k)}\right\|_{\star}.$$

Combining these results with Lemma E.2, we can conclude that with high probability:

$$\left\|\tilde{\boldsymbol{\mathcal{B}}}^{(k)}\right\|_{\text{tsp}} \lesssim \frac{rd_1d_3}{N_S} + h_k\sqrt{\frac{d_1}{N_S}}.$$

This bound demonstrates that our debiasing procedure effectively reduces both the variance and the bias of the local estimators, setting the stage for efficient knowledge transfer in the D-LoRT framework.

**Proof of Theorem E.1**    To prove Theorem E.1, we start by defining a modified loss function that incorporates both the target and source task information:

$$\tilde{\mathcal{L}}(\vec{\boldsymbol{\Theta}}) = \frac{1}{2N}\|\mathbf{y}^{(0)} - \mathfrak{X}^{(0)}(\boldsymbol{\Theta}^{(0)})\|_2^2 + \frac{1}{2N}\sum_{k=1}^{K}N_k\|\tilde{\boldsymbol{\mathcal{W}}}^{(k)} - \boldsymbol{\Theta}^{(k)} - \boldsymbol{\Theta}^{(0)}\|_{\text{F}}^2 + \lambda_0\mathcal{R}(\vec{\boldsymbol{\Theta}}).$$

Here, $\boldsymbol{\Theta}^{(0)}$ represents the target task parameter, $\boldsymbol{\Theta}^{(k)}$ are the source task parameters, and $\mathcal{R}(\vec{\boldsymbol{\Theta}})$ is a regularization term.

Next, we need to bound $\left\langle\nabla\tilde{\mathcal{L}}\left(\vec{\boldsymbol{\Theta}}\right),\vec{\boldsymbol{\Delta}}\right\rangle$. We introduce two auxiliary terms:

$$\delta_k = \frac{rd_1d_3}{N} + \frac{N_S}{N}\sqrt{\frac{d_1}{N_S}}h_k$$

$$\delta_0 = \frac{Krd_1d_3}{N} + \sqrt{\frac{d_1}{N_S}}\bar{h}.$$

These terms help us capture the effects of task heterogeneity and sample sizes. We can now state a key lemma:

**Lemma E.3** (Concentration of Gradient). *Under Assumptions 3.2 and 3.3, if $N_S \gtrsim d_1$, and we choose*

$$\lambda_k = c_k\left(\sqrt{\frac{N_S}{N}\frac{d_1}{N}} + \delta_k\right)$$

$$\lambda_0 = c_0\left(\sqrt{\frac{d_1}{N}} + \delta_0\right)$$

*for some appropriate constants $c_0,\ldots,c_K$, then for any $\vec{\boldsymbol{\Delta}}$,w.h.p.:*

$$\left|\left\langle\nabla\tilde{\mathcal{L}}\left(\vec{\boldsymbol{\Theta}}\right),\vec{\boldsymbol{\Delta}}\right\rangle\right| \leq \sum_{k=1}^{K}\frac{\lambda_k}{2}\left\|\boldsymbol{\Delta}^{(k)}\right\|_{\star} + \frac{\lambda_0}{2}\left\|\boldsymbol{\Delta}^{(0)}\right\|_{\star}.$$

The key difference between Lemma E.3 and the earlier Lemma D.1 is in the choice of $\{\lambda_k\}_{k=0}^{K}$. This new choice allows us to leverage Lemma D.4 under the following additional conditions:

1. $N_S \gg Krd_1d_3$ (sample size dominates task complexity),

2. $N_S \gtrsim K^2 d_1$ (sample size dominates number of tasks squared),

3. $h_k \asymp \bar{h}$ for any $1 \leq k \leq K$ (task heterogeneity is of the same order).

Applying Lemma D.4, we obtain with high probability:

$$\|\hat{\mathcal{W}}_{\star C}^{\mathrm{a}} - \mathcal{W}_{\star}^{\mathrm{a}}\|_{\mathrm{F}} \lesssim \sqrt{rd_3}\lambda_0 + \sqrt{\sum_{k=1}^{K} \lambda_k h_k}$$

$$\lesssim \sqrt{rd_3}\left(\sqrt{\frac{d_1}{N}} + \delta_0\right) + \sqrt{\sum_{k=1}^{K}\left(\frac{N_S}{N}\sqrt{\frac{d_1}{N_S}} + \delta_k\right)h_k}.$$

Assuming $h_k \asymp \bar{h} = O(1)$, we can further simplify our bound:

$$rd_3\delta_0^2 = \frac{K^2(rd_3)^3d_1^2}{N^2} + \frac{rd_1d_3}{N_S}\bar{h}^2 \lesssim \frac{rd_1d_3}{N} + \sqrt{\frac{d_1}{N_S}}\bar{h} \tag{E.4}$$

$$\sum_{k=1}^{K}\delta_k h_k = \frac{rd_1d_3}{N_S}\bar{h} + \sqrt{\frac{d_1}{N_S}}\sum_{k=1}^{K}\frac{N_S}{N}h_k^2 \lesssim \sqrt{\frac{d_1}{N_S}}\bar{h}. \tag{E.5}$$

These simplifications rely on the condition $N_S \gg K(rd_3)^2 d_1$, which ensures that the sample size is sufficiently large relative to the problem's complexity.

Combining all these results, we arrive at the final bound stated in Theorem E.1, completing our proof.

### E.1. Analysis of Step 2 of D-LoRT

In this section, we analyze the second step of our Distributed Low-Rank Tensor Transitions (D-LoRT) method. This step refines the initial estimate to achieve better performance. We present a more detailed version of Theorem 4.5, which provides tighter bounds on the estimation error.

**Theorem E.4** (Refined D-LoRT). *Under the conditions of Theorem 4.4, and assuming:*

- $N_T \gtrsim rd_1d_3$ *(target sample size sufficiently large),*
- $\bar{h}\sqrt{d_1/N_T} = o(1)$ *(task heterogeneity not too large).*

*If we choose the parameters as follows:*

$$\lambda_k = c_0(12 \vee \frac{\bar{h}}{h_k})\left(\sqrt{\frac{N_S}{N}\frac{d_1}{N}} + \delta_k\right),$$

$$\lambda_0 = c_0\left(\sqrt{\frac{d_1}{N}}\mathbb{I}_{\mathbf{A}} + \sqrt{\frac{d_1}{N_S}}\mathbb{I}_{\mathbf{A}^c} + \delta_0\right)$$

*where*

$$\delta_k = \frac{rd_1d_3}{N} + \frac{N_S}{N}\sqrt{\frac{d_1}{N_S}}h_k,$$

$$\delta_0 = \frac{Krd_1d_3}{N} + \sum_{k=1}^{K}\frac{N_S}{N}\sqrt{\frac{d_1}{N_S}}h_k$$

*then with high probability:*

$$\|\hat{\mathcal{W}}_{\mathrm{dlort}}^{(0)} - \mathcal{W}_{\star}^{(0)}\|_{\mathrm{F}}^2 \lesssim \frac{rd_1d_3}{N} + \bar{h}\sqrt{\frac{d_1}{N_T}}.$$

**Proof Sketch:** We follow a similar line of argument as in the proof of Theorem 4.3. The key steps are as follows:

1) **Verify conditions:** We first verify that if $N_S \gg K r d_1 d_3$, $N_S \gtrsim K^2 d_1$, and $h_k \asymp \bar{h}$ for any $1 \leq k \leq K$, our choice of parameters satisfies the conditions in Lemma D.4.

2) **Apply Lemma D.4:** This allows us to bound the estimation error in both Frobenius and nuclear norms:

$$\|\boldsymbol{\Delta}_{\mathsf{d}}^{\mathsf{a}}\|_{\mathrm{F}} \lesssim \sqrt{r d_3} \lambda_0 + \sqrt{\sum_{k=1}^{K} \lambda_k h_k}$$

$$\lesssim \sqrt{\frac{r d_1 d_3}{N}} \mathbb{I}_{\mathbf{A}} + \sqrt{\frac{r d_1 d_3}{N_S}} \mathbb{I}_{\mathbf{A}^c} + \sqrt{\sqrt{\frac{d_3}{N_S}} \bar{h}} + \sqrt{r d_3} \delta_0 + \sqrt{\sum_{k=1}^{K} \delta_k h_k} \tag{E.6}$$

and

$$\|\boldsymbol{\Delta}_{\mathsf{d}}^{\mathsf{a}}\|_{\star} \lesssim r d_3 \lambda_0 + \sqrt{r d_3} \sqrt{\sum_{k=1}^{K} \lambda_k h_k} + \frac{\sum_{k=1}^{K} \lambda_k}{\lambda_0} \bar{h}$$

$$\lesssim \sqrt{\frac{r^2 d_3^2 d_1}{N}} \mathbb{I}_{\mathbf{A}} + \sqrt{\frac{r d_3{}^2 d_1}{N_S}} \mathbb{I}_{\mathbf{A}^c} + \sqrt{\sqrt{\frac{d_1}{N_S}} r d_3 \bar{h}} + r d_3 \delta_0 + \sqrt{\sum_{k=1}^{K} r d_3 \delta_k h_k} + \sqrt{K} \bar{h} \mathbb{I}_{\mathbf{A}} + \bar{h} \mathbb{I}_{\mathbf{A}^c}. \tag{E.7}$$

Here, $\mathbb{I}_{\mathbf{A}}$ is the indicator function for event $\mathbf{A}$, which represents the case where source task information is beneficial.

3) **Simplify bounds:** To complete the proof, we need to show that the terms involving $\delta_0$ and $\delta_k$ in bounds (E.6) and (E.7) align with other terms. We use the results from (E.4) and (E.5) to show:

$$\sqrt{r d_3} \delta_0 + \sqrt{\sum_{k=1}^{K} \delta_k h_k} \lesssim \sqrt{\frac{r d_1 d_3}{N}} + \sqrt{\sqrt{\frac{d_1}{N_S}} \bar{h}}.$$

4) **Conclude:** With these simplified bounds, we can follow the same steps as in the proof of Theorem 4.3 to arrive at our final result.

**Interpretation:** The theorem shows that our two-step D-LoRT method achieves an estimation error that scales with two main terms: 1) $\frac{r d_1 d_3}{N}$: This represents the standard error for low-rank tensor estimation. 2) $\bar{h} \sqrt{\frac{d_1}{N_T}}$: This term captures the impact of task heterogeneity and the target sample size.

The bound demonstrates that D-LoRT effectively leverages information from source tasks while being robust to task heterogeneity, achieving performance comparable to centralized methods under certain conditions.

### E.2. Proof of Lemmas in the Analysis of D-LoRT

**Proof of Lemma E.3** We aim to bound the inner product $\left| \left\langle \nabla \tilde{\mathcal{L}} \left( \vec{\boldsymbol{\Theta}} \right), \boldsymbol{\Delta} \right\rangle \right|$. This bound is crucial for establishing the convergence properties of our D-LoRT algorithm.

First, we need to understand the structure of the noise in our model. For each source task $k = 1, \ldots, K$, the noise term is given by:

$$\tilde{\boldsymbol{\epsilon}}^{(k)} = \sqrt{N_S} \left( \frac{\hat{\boldsymbol{\Theta}}^{(k)} \boldsymbol{\mathfrak{X}}^{(k)} (\boldsymbol{\mathcal{E}}^{(k)})}{N_S} + \tilde{\boldsymbol{\mathcal{B}}}^{(k)} \right).$$

For the target task ($k = 0$), the noise is simply $\boldsymbol{\epsilon}^{(0)}$, which is the observation noise for the target model.

Now, we can bound the inner product using Hölder's inequality:

$$\left| \left\langle \nabla \mathcal{L} \left( \vec{\Theta} \right), \vec{\Delta} \right\rangle \right| = \left| \sum_{k=1}^{K} \left\langle \frac{\sqrt{N_S}}{N} \tilde{\boldsymbol{\epsilon}}^{(k)}, \boldsymbol{\Delta}^{(k)} \right\rangle + \left\langle \sum_{k=1}^{K} \frac{\sqrt{N_S}}{N} \tilde{\boldsymbol{\epsilon}}^{(k)} + \frac{1}{N} \mathfrak{X}^{(0)}(\boldsymbol{\epsilon}^{(0)}), \boldsymbol{\Delta}^{(0)} \right\rangle \right|$$

$$\leq \sum_{k=1}^{K} \left\| \frac{\sqrt{N_S}}{N} \tilde{\boldsymbol{\epsilon}}^{(k)} \right\|_{\text{tsp}} \left\| \boldsymbol{\Delta}^{(k)} \right\|_{\star} + \left\| \sum_{k=1}^{K} \frac{\sqrt{N_S}}{N} \tilde{\boldsymbol{\epsilon}}^{(k)} + \frac{1}{N} \mathfrak{X}^{(0)}(\boldsymbol{\epsilon}^{(0)}) \right\|_{\text{tsp}} \left\| \boldsymbol{\Delta}^{(0)} \right\|_{\star}.$$

Substituting the expression for $\tilde{\boldsymbol{\epsilon}}^{(k)}$, we get:

$$\left| \left\langle \nabla \mathcal{L} \left( \vec{\Theta} \right), \vec{\Delta} \right\rangle \right| \leq \sum_{k=1}^{K} \left\| \frac{1}{N} \hat{\boldsymbol{\Theta}}^{(k)} \mathfrak{X}^{(k)}(\boldsymbol{\mathcal{E}}^{(k)}) + \frac{N_S}{N} \tilde{\boldsymbol{\mathcal{B}}}^{(k)} \right\|_{\text{tsp}} \left\| \boldsymbol{\Delta}^{(k)} \right\|_{\star}$$

$$+ \left\| \frac{1}{N} \left( \sum_{k=1}^{K} \hat{\boldsymbol{\Theta}}^{(k)} \mathfrak{X}^{(k)}(\boldsymbol{\mathcal{E}}^{(k)}) + \mathfrak{X}^{(0)}(\boldsymbol{\epsilon}^{(0)}) \right) + \sum_{k=1}^{K} \frac{N_S}{N} \tilde{\boldsymbol{\mathcal{B}}}^{(k)} \right\|_{\text{tsp}} \left\| \boldsymbol{\Delta}^{(0)} \right\|_{\star}.$$

Now we proceed by choosing $\lambda_k$ for source tasks. From our previous analysis, we know that with high probability:

$$\frac{1}{N_S} \left\| \hat{\boldsymbol{\Theta}}^{(k)} \mathfrak{X}^{(k)}(\boldsymbol{\mathcal{E}}^{(k)}) \right\|_{\text{tsp}} \lesssim \sqrt{\frac{d_1}{N_S}} \quad \text{and} \quad \left\| \tilde{\boldsymbol{\mathcal{B}}}^{(k)} \right\|_{\text{tsp}} \lesssim \frac{r d_1 d_3}{N_S} + h_k \sqrt{\frac{d_1}{N_S}}.$$

To ensure that $\lambda_k$ is large enough to bound the source task terms, we need:

$$\lambda_k \geq \left\| \frac{1}{N} \hat{\boldsymbol{\Theta}}^{(k)} \mathfrak{X}^{(k)}(\boldsymbol{\mathcal{E}}^{(k)}) + \frac{N_S}{N} \tilde{\boldsymbol{\mathcal{B}}}^{(k)} \right\|_{\text{tsp}}.$$

A sufficient choice is:

$$\lambda_k = c_k \left( \sqrt{\frac{N_S}{N} \frac{d_1}{N}} + \frac{r d_1 d_3}{N} + \frac{N_S}{N} \sqrt{\frac{d_1}{N_S}} h_k \right)$$

for some sufficiently large constant $c_k$.

Then we choose $\lambda_0$ for the target task. For the target task, we need to bound the combined term. Using the triangle inequality:

$$\left\| \frac{1}{N} \left( \sum_{k=1}^{K} \hat{\boldsymbol{\Theta}}^{(k)} \mathfrak{X}^{(k)}(\boldsymbol{\mathcal{E}}^{(k)}) + \mathfrak{X}^{(0)}(\boldsymbol{\epsilon}^{(0)}) \right) + \sum_{k=1}^{K} \frac{N_S}{N} \tilde{\boldsymbol{\mathcal{B}}}^{(k)} \right\|_{\text{tsp}}$$

$$\lesssim \frac{1}{N} \left\| \sum_{k=1}^{K} \hat{\boldsymbol{\Theta}}^{(k)} \mathfrak{X}^{(k)}(\boldsymbol{\mathcal{E}}^{(k)}) \right\|_{\text{tsp}} + \frac{1}{N} \left\| \mathfrak{X}^{(0)}(\boldsymbol{\epsilon}^{(0)}) \right\|_{\text{tsp}} + \sum_{k=1}^{K} \frac{N_S}{N} \left\| \tilde{\boldsymbol{\mathcal{B}}}^{(k)} \right\|_{\text{tsp}}$$

$$\lesssim \sqrt{\frac{d_1}{N}} + \frac{K r d_1 d_3}{N} + \sum_{k=1}^{K} \frac{N_S}{N} \sqrt{\frac{d_1}{N_S}} h_k.$$

Therefore, a sufficient choice for $\lambda_0$ is:

$$\lambda_0 = c_0 \left( \sqrt{\frac{d_1}{N}} + \frac{K r d_1 d_3}{N} + \sum_{k=1}^{K} \frac{N_S}{N} \sqrt{\frac{d_1}{N_S}} h_k \right)$$

for some constant $c_0$.

With these choices of $\lambda_k$ and $\lambda_0$, we have established that *w.h.p.*:

$$\left| \left\langle \nabla \mathcal{L} \left( \vec{\Theta} \right), \vec{\Delta} \right\rangle \right| \leq \sum_{k=1}^{K} \frac{\lambda_k}{2} \left\| \boldsymbol{\Delta}^{(k)} \right\|_{\star} + \frac{\lambda_0}{2} \left\| \boldsymbol{\Delta}^{(0)} \right\|_{\star}.$$

This bound is crucial for the convergence analysis of our D-LoRT algorithm, as it allows us to control the behavior of the gradient of our loss function.

# F. Algorithm Design and Comparison for LoRT and D-LoRT

This section presents the algorithmic formulation and implementation details of both LoRT and D-LoRT, with a focus on their respective proximal gradient descent (PGD) procedures. We further provide a comparative analysis of their computational complexity and communication costs, highlighting the trade-offs between centralized and decentralized design under the low-tubal-rank tensor regression framework.

## F.1. Optimization Formulation and PGD Implementation

To solve the proposed low-rank transferable tensor regression model, we adopt a two-step optimization strategy grounded in PGD. This subsection introduces the formal optimization objective and details the corresponding algorithmic procedures. Specifically, we first formulate the joint low-rank estimation problem across multiple tasks with TNN regularization, and then present the PGD implementation used in Step 1 of LoRT. For Step 2 and task-only variants, we similarly derive problem-specific PGD routines adapted to the single-task setting. These implementations enable efficient and scalable optimization while preserving the low-rank structure essential to the transfer learning mechanism.

**Optimization Formulation and PGD Implementation for LoRT Step 1**  We consider the joint low-rank estimation step in LoRT under the following optimization problem:

$$\min_{\{\Theta^{(k)}\}_{k=0}^K} \frac{1}{2N} \sum_{k=0}^K \left\| \mathbf{y}^{(k)} - \mathfrak{X}^{(k)}(\Theta^{(0)} + \Theta^{(k)}) \right\|_2^2 + \lambda_0 \sum_{k=0}^K a_k \|\Theta^{(k)}\|_\star, \tag{F.1}$$

where:

- $\Theta^{(0)} := \mathcal{W}^{(0)}$ is the target model parameter,

- $\Theta^{(k)} := \mathcal{W}^{(k)} - \mathcal{W}^{(0)}$ for $k = 1, \ldots, K$ represents the model discrepancy between source task $k$ and the target,

- $\mathbf{y}^{(k)}$ denotes the observations for task $k$ (see Eq. (C.3)),

- $\mathfrak{X}^{(k)}(\cdot)$ is the design operator for task $k$ (see Eq. (C.3)),

- $\| \cdot \|_\star$ denotes the TNN (Definition 2.3),

- $a_k \in \mathbb{R}_+$ is the regularization weight assigned to task $k$ (with $a_0 = 1$ for the target),

- $N = N_T + K N_S$ is the total number of training samples.

**Proximal Gradient Descent Strategy.**  Let $\Theta_t^{(k)}$ denote the current iterate for task $k$. We define the proximal update rule based on the gradient of the loss function and apply soft-thresholding with respect to the tubal nuclear norm:

1. Compute the gradient of the loss function with respect to each $\Theta^{(k)}$:

$$\nabla_{\Theta^{(k)}} \mathcal{L} = \begin{cases} \frac{1}{N} \sum_{j=0}^K \mathfrak{X}^{*(j)} \left( \mathfrak{X}^{(j)}(\Theta^{(0)} + \Theta^{(j)}) - \mathbf{y}^{(j)} \right), & k = 0 \\ \frac{1}{N} \mathfrak{X}^{*(k)} \left( \mathfrak{X}^{(k)}(\Theta^{(0)} + \Theta^{(k)}) - \mathbf{y}^{(k)} \right), & k = 1, \ldots, K \end{cases} \tag{F.2}$$

2. Take a gradient descent step for each $\Theta^{(k)}$ by $\mathcal{Z}^{(k)} = \Theta_t^{(k)} - \frac{1}{\gamma} \nabla_{\Theta^{(k)}} \mathcal{L}$.

3. Apply proximal operator

$$\Theta_{t+1}^{(k)} = \text{prox}_{\frac{\lambda_0 a_k}{\gamma} \|\cdot\|_\star}(\mathcal{Z}^{(k)}), \tag{F.3}$$

where $\text{prox}_{\tau \|\cdot\|_\star}(\cdot)$ denotes the proximity operator of the tubal nuclear norm (TNN) with threshold $\tau = \frac{\lambda_0 a_k}{\gamma}$. Following the T-SVT scheme in Lu et al. (2019b), this operator can be efficiently computed as follows:

(a) Apply a linear invertible transform $M$ (e.g., DCT) along the third mode of $\mathcal{Z}^{(k)}$ to obtain $\bar{\mathcal{Z}}^{(k)} = M(\mathcal{Z}^{(k)})$;

(b) For each frontal slice $\bar{Z}_i^{(k)}$ of $\bar{\boldsymbol{\mathcal{Z}}}^{(k)}$, compute the SVD: $\bar{Z}_i^{(k)} = U_i S_i V_i^\top$;

(c) Apply soft-thresholding to singular values: $S_i^\tau = \operatorname{diag}((\sigma_i - \tau)_+)$, where $(\cdot)_+ = \max(\cdot, 0)$;

(d) Reconstruct each slice $\bar{\Theta}_i^{(k)} = U_i S_i^\tau V_i^\top$ and form the tensor $\bar{\boldsymbol{\Theta}}^{(k)}$;

(e) Apply the inverse transform: $\boldsymbol{\Theta}_{t+1}^{(k)} = M^{-1}(\bar{\boldsymbol{\Theta}}^{(k)})$.

This procedure corresponds to the Tensor Singular Value Thresholding (T-SVT) operator, which is the proximal operator associated with the TNN. It promotes low-tubal-rank structure by shrinking singular values in the transformed domain slice-wise.

Algorithm 1 summarizes the above three steps.

---

**Algorithm 1** PGD for Joint Low-Rank Estimation (LoRT Step 1)

---

**Input:** Responses $\{\mathbf{y}^{(k)}\}_{k=0}^K$, design tensors $\{\boldsymbol{\mathcal{X}}_i^{(k)}\}$, regularization weights $\{a_k\}$, step size $\gamma$, max iterations $T$

**Output:** Estimated tensors $\{\boldsymbol{\Theta}_T^{(k)}\}_{k=0}^K$

1: Initialize $\boldsymbol{\Theta}_0^{(k)} \leftarrow \mathbf{0}$ for all $k = 0, \dots, K$
2: **for** $t = 0$ to $T - 1$ **do**
3:  **for** $k = 0$ to $K$ **do**
4:   Compute gradient $\nabla_{\boldsymbol{\Theta}^{(k)}} \mathcal{L}$ with current $\{\boldsymbol{\Theta}_t^{(k)}\}$
5:   $\boldsymbol{\mathcal{Z}}^{(k)} \leftarrow \boldsymbol{\Theta}_t^{(k)} - \frac{1}{\gamma} \nabla_{\boldsymbol{\Theta}^{(k)}} \mathcal{L}$
6:   $\boldsymbol{\Theta}_{t+1}^{(k)} \leftarrow \operatorname{prox}_{\frac{\lambda_0 a_k}{\gamma} \|\cdot\|_\star}(\boldsymbol{\mathcal{Z}}^{(k)})$
7:  **end for**
8: **end for**

---

**Target-Specific Refinement (LoRT Step 2).** Given an initial estimate $\hat{\boldsymbol{\mathcal{W}}}_\star^{\mathrm{a}}$ from LoRT Step 1, we refine the target parameter using target-only data via the following optimization:

$$\hat{\boldsymbol{\mathcal{C}}} \in \operatorname*{arg\,min}_{\boldsymbol{\mathcal{C}} \in \mathbb{R}^{d_1 \times d_2 \times d_3}} \frac{1}{2N_T} \sum_{i=1}^{N_T} \left( y_i^{(0)} - \langle \boldsymbol{\mathcal{X}}_i^{(0)}, \hat{\boldsymbol{\mathcal{W}}}_\star^{\mathrm{a}} + \boldsymbol{\mathcal{C}} \rangle \right)^2 + \tilde{\lambda} \|\boldsymbol{\mathcal{C}}\|_\star. \tag{F.4}$$

The final refined parameter is given by $\hat{\boldsymbol{\mathcal{W}}}_\star^{\mathrm{a}} + \hat{\boldsymbol{\mathcal{C}}}$. This optimization is solved using PGD with TNN regularization (Algorithm 2).

---

**Algorithm 2** PGD for Target-Specific Refinement (LoRT Step 2)

---

**Input:** Target responses $\mathbf{y}^{(0)}$, design tensors $\{\boldsymbol{\mathcal{X}}_i^{(0)}\}_{i=1}^{N_T}$, initial estimate $\hat{\boldsymbol{\mathcal{W}}}_\star^{\mathrm{a}}$, step size $\gamma$, regularization $\tilde{\lambda}$, max iterations $T$

**Output:** Refined target tensor $\hat{\boldsymbol{\mathcal{W}}}_{\mathrm{Step2}}$

1: **Initialize:** $\boldsymbol{\mathcal{C}}_0 \leftarrow \mathbf{0}$
2: **for** $t = 0$ to $T - 1$ **do**
3:  Compute gradient: $\nabla_{\boldsymbol{\mathcal{C}}} \mathcal{L} = -\frac{1}{N_T} \sum_{i=1}^{N_T} \left( y_i^{(0)} - \langle \boldsymbol{\mathcal{X}}_i^{(0)}, \hat{\boldsymbol{\mathcal{W}}}_\star^{\mathrm{a}} + \boldsymbol{\mathcal{C}}_t \rangle \right) \boldsymbol{\mathcal{X}}_i^{(0)}$
4:  Gradient descent step: $\boldsymbol{\mathcal{Z}} \leftarrow \boldsymbol{\mathcal{C}}_t - \frac{1}{\gamma} \nabla_{\boldsymbol{\mathcal{C}}} \mathcal{L}$
5:  Proximal update: $\boldsymbol{\mathcal{C}}_{t+1} \leftarrow \operatorname{prox}_{\frac{\tilde{\lambda}}{\gamma} \|\cdot\|_\star}(\boldsymbol{\mathcal{Z}})$
6: **end for**
7: **Return:** $\hat{\boldsymbol{\mathcal{W}}}_{\mathrm{Step2}} \leftarrow \hat{\boldsymbol{\mathcal{W}}}_\star^{\mathrm{a}} + \boldsymbol{\mathcal{C}}_T$

---

**Task-Only Tensor Regression.** We consider solving the following convex optimization problem using only single-task data, either for the target-only refinement step or for each local source task update in D-LoRT. While our theoretical analysis employs a debiased estimator (cf. Eq. (E.2)) to facilitate convergence guarantees, the actual optimization implemented in

practice uses the standard TNN regularization:

$$\hat{\mathcal{W}} \in \underset{\mathcal{W} \in \mathbb{R}^{d_1 \times d_2 \times d_3}}{\arg\min} \frac{1}{2N_k} \sum_{i=1}^{N_k} \left( y_i^{(k)} - \langle \mathcal{X}_i^{(k)}, \mathcal{W} \rangle \right)^2 + \lambda \|\mathcal{W}\|_\star. \tag{F.5}$$

The algorithmic steps are provided in Algorithm 3.

---

**Algorithm 3** PGD for Task-Only Tensor Regression

---

**Input:** Task-specific responses $\mathbf{y}^{(k)}$, tensors $\{\mathcal{X}_i^{(k)}\}_{i=1}^{N_k}$, step size $\gamma$, regularization $\lambda$, max iterations $T$
**Output:** Estimated target tensor $\hat{\mathcal{W}}$

1: **Initialize:** $\mathcal{W}_0 \leftarrow \mathbf{0}$
2: **for** $t = 0$ to $T - 1$ **do**
3:  Compute gradient: $\nabla_{\mathcal{W}} \mathcal{L} = -\frac{1}{N_k} \sum_{i=1}^{N_T} \left( y_i^{(k)} - \langle \mathcal{X}_i^{(k)}, \mathcal{W}_t \rangle \right) \mathcal{X}_i^{(k)}$
4:  Gradient descent step: $\mathcal{Z} \leftarrow \mathcal{W}_t - \frac{1}{\gamma} \nabla_{\mathcal{W}} \mathcal{L}$
5:  Proximal update: $\mathcal{W}_{t+1} \leftarrow \mathrm{prox}_{\frac{\lambda}{\gamma}\|\cdot\|_\star}(\mathcal{Z})$
6: **end for**
7: **Return:** $\hat{\mathcal{W}} \leftarrow \mathcal{W}_T$

---

### F.2. Algorithmic and Communication Comparison between LoRT and D-LoRT

In this subsection, we revisit the distinction between the proposed LoRT and its distributed variant D-LoRT, emphasizing that both frameworks can operate in distributed environments where multiple source nodes (or clients) contribute data or models. The key difference lies in what is communicated across nodes: LoRT requires the transfer of raw training data to a central optimizer, while D-LoRT limits communication to model-level summaries. This leads to important implications for scalability, privacy, and deployment feasibility.

**Algorithmic Design.** We begin by comparing how LoRT and D-LoRT structure their learning pipelines under the assumption that source nodes are distributed and cannot share gradients or samples interactively during training.

LoRT aggregates full training data from all nodes—either by uploading raw tensor samples to a central server or via a shared memory system—and performs a joint low-rank optimization over all source and target tasks. In contrast, D-LoRT keeps raw data strictly local: each node performs local regression with tubal nuclear norm regularization, and transmits only the resulting model (optionally debiased) to a coordinating node. This decoupled strategy simplifies communication and protects sensitive data. Table 9 summarizes this design contrast:

Table 9: Algorithm Structure Comparison

| Module | LoRT (Data-sharing) | D-LoRT (Model-sharing) |
|---|---|---|
| *Local Training* | No—data sent to a central optimizer | Yes—local regression with TNN regularization |
| *What Is Shared* | Raw tensors $\left\{ (\mathcal{X}_i^{(k)}, y_i^{(k)}) \right\}$ | Local models $\hat{\mathcal{W}}^{(k)}$ only |
| *Fusion Strategy* | Global joint optimization | Model aggregation with optional debiasing |
| *Target Refinement* | Centralized update with full gradient access | Same refinement, but initialized from aggregated models |

**Algorithmic Complexity.** We now analyze the algorithmic complexity of LoRT and D-LoRT. First, the TNN proximal operator in Eq. (F.3) plays a central role in both LoRT and D-LoRT via iterative proximal gradient descent (PGD). Its computational complexity arises from two main operations:

- *Transform Domain Conversion:* The tensor is first transformed along the third mode using either DFT or the DCT, typically applied via a fast transform with complexity $O(d_1 d_2 \log d_3)$.

- *Frontal Slice-wise SVDs:* After transformation, the tensor is decomposed into $d_3$ frontal slices, each requiring a matrix SVD. The cost of computing the SVD of a single slice is $O(\min\{d_1, d_2\} d_1 d_2)$, and thus the total slice-wise SVD cost is $O(\min\{d_1, d_2\} d_1 d_2 d_3)$.

Putting these together, the total cost of one TNN proximal operator call is:

$$O\left(d_1 d_2 \log d_3 + d_3 \cdot \min\{d_1, d_2\} \cdot d_1 d_2\right),$$

which is typically dominated by the frontal-slice SVD term for large $d_3$ or high spatial resolution.

We now use the above cost to analyze and compare the computational complexity of LoRT and D-LoRT in both Step 1 (low-rank estimation) and Step 2 (target refinement).

*Step 1: Joint Low-Rank Estimation.* In LoRT, Step 1 performs joint optimization over all $K + 1$ tasks. Each PGD iteration involves one TNN proximal step and one gradient computation per task, leading to a per-iteration cost of:

$$O\left((K + 1)\left(d_1 d_2 \log d_3 + d_3 \cdot \min\{d_1, d_2\} \cdot d_1 d_2\right)\right).$$

Since all updates are handled centrally, this complexity must be sustained by the server.

In contrast, D-LoRT distributes this cost across $K$ source nodes. Each node performs local PGD to estimate its model, and the target node runs a similar routine. Hence, the per-node cost is:

$$O\left(d_1 d_2 \log d_3 + d_3 \cdot \min\{d_1, d_2\} \cdot d_1 d_2\right),$$

and the system-wide complexity is $O(K)$ times this cost, but fully parallelized.

*Step 2: Target-Specific Refinement.* Both methods use the same proximal TNN update on the target node. Thus, the per-iteration cost is:

$$O\left(d_1 d_2 \log d_3 + d_3 \cdot \min\{d_1, d_2\} \cdot d_1 d_2\right),$$

with no difference between LoRT and D-LoRT.

*Summary.* LoRT concentrates the total cost of all $(K + 1)$ TNN updates at a central server, while D-LoRT spreads the burden across distributed devices. Although both require the same per-task proximal operation, D-LoRT benefits from parallelism and reduced server load, making it more suitable for federated or bandwidth-constrained environments.

**Communication Overhead.** Both LoRT and D-LoRT involve transmitting information from $K$ distributed source nodes to a coordinating unit. The fundamental distinction lies in the *granularity* of what is transferred. LoRT requires high-volume transmission of raw training data—potentially multiple high-dimensional tensors per node—while D-LoRT reduces this burden by communicating only a single third-order tensor per node (e.g., $d_1 \times d_2 \times d_3$ parameters). The comparison in Table 10 quantifies this difference:

Table 10: Communication Cost Comparison

| Aspect | LoRT (Data-sharing) | D-LoRT (Model-sharing) |
|---|---|---|
| *Transferred Item* | Raw data tensors | One model tensor per node |
| *Transfer Volume per Node* | $O(N_S D)$ | $O(D)$ |
| *Total Uplink Volume* | $O(K N_S D)$ | $O(KD)$ |
| *Communication Rounds* | One (upload before joint training) | One (after local training) |
| *Privacy Level* | Raw data exposed | No raw data shared |

**Deployment Trade-offs.**    Finally, we contrast the broader implications of LoRT and D-LoRT in real-world distributed systems. While both frameworks target tensor-based transfer learning, their suitability varies based on infrastructure constraints. LoRT is advantageous when raw data transmission is permissible and joint modeling is desired. D-LoRT, in contrast, offers a practical and privacy-preserving alternative for federated or multi-institutional settings. Table 11 synthesizes these trade-offs:

Table 11: Practical Deployment Comparison

| Criterion | LoRT (Data-sharing) | D-LoRT (Model-sharing) |
|---|---|---|
| *Data Visibility* | Full access to raw data from all nodes | Only parameter summaries shared |
| *Scalability* | Limited by bandwidth and data centralization | High—nodes train independently |
| *Task Coupling* | Tight—joint gradients across tasks | Loose—models combined post-training |
| *Communication Cost* | High—proportional to sample size $N_k$ | Low—only model tensors shared |
| *Privacy Sensitivity* | Unsuitable for privacy-restricted data | Compliant with federated protocols |
| **Recommended Scenario** | Controlled academic or cloud-hosted data settings | Federated learning, healthcare, or multi-site collaboration |

# G. List of Symbols and Notations

| Symbol/Notation | Description |
|---|---|
| $\mathbb{R}^{d_1 \times d_2 \times d_3}$ | Space of third-order tensors with dimensions $d_1 \times d_2 \times d_3$ |
| $d_1, d_2, d_3$ | Dimensions of the tensor with $d_1 \geq d_2$ |
| $D = d_1 d_2 d_3$ | Total dimension of the tensor |
| $K$ | Number of source tasks |
| $N_T, N_S$ | Sample sizes for target and source tasks |
| $N_k$ | sample size of the $k$-th task: if $k = 0$, then $N_k = N_T$, otherwise $N_k = N_S$ |
| $N = N_T + K N_S$ | Total sample size |
| $\mathcal{W}_\star^{(0)}$ | True parameter tensor for the target task |
| $\mathcal{W}_\star^{(k)}$ | True parameter tensor for the $k$-th source task |
| $\Theta_\star^{(k)}$ | Difference between $k$-th source and target parameters |
| $\vec{\Theta}_\star$ | Collection of all task parameters |
| $\hat{\mathcal{W}}^{\mathrm{a}}$ | Weighted average estimator |
| $\Delta^{\mathrm{a}}$ | Estimation error of the average parameter |
| $\hat{\mathcal{W}}^{(0)}$ | Estimator for target task |
| $\hat{\mathcal{W}}^{(k)}$ | Estimator for $k$-th source task |
| $\hat{\vec{\Theta}}$ | Collection of all estimated parameters |
| $\vec{\Delta}$ | Estimation error for all parameters |
| $\mathcal{X}_i^{(0)}, \mathcal{X}_i^{(k)}$ | Covariate tensors for target and source tasks |
| $y_i^{(0)}, y_i^{(k)}$ | Response variables for target and source tasks |
| $\epsilon_i^{(0)}, \epsilon_i^{(k)}$ | Noise terms for target and source tasks |
| $\hat{\Sigma}^{(k)}$ | Empirical covariance matrix for the $k$-th task |
| $\mathfrak{X}^{(k)}(\cdot)$ | Design operator for the $k$-th task |
| $\mathfrak{X}^{*(k)}(\cdot)$ | Adjoint of the design operator for the $k$-th task |
| $\mathcal{P}_\star(\cdot), \mathcal{P}_{\star\perp}(\cdot)$ | Projection operators in Eq. (C.1) |
| $h_k$ | Model shift parameter for the $k$-th source task |
| $\bar{h}$ | Average model shift |
| $\delta_\star$ | Source task alignment parameter |
| $\mathbf{h} := (h_1, \ldots, h_K)^\top$ | Vector of model shift parameters |
| $\sigma_k$ | Standard deviation of covariates for the $k$-th task |
| $c_x$ | Universal constant bounding $\sigma_k$ |
| $c_\epsilon$ | Upper bound on the variance of noise terms |
| $M(\cdot), M^{-1}(\cdot)$ | Linear transform and its inverse in Eq. (1) |
| $\mathbf{M}$ | Orthogonal matrix defining $M(\cdot)$ |
| $\mathrm{vec}(\cdot)$ | Vectorization operator |
| $\langle \cdot, \cdot \rangle$ | Inner product |
| $*_M$ | t-product operation |
| $\circ$ | Outer product |
| $\odot$ | Tensor frontal-slice-wise product |
| $r$ | Tubal rank of $\mathcal{W}_\star^{(0)}$ |
| $r_{\mathrm{t}}(\cdot)$ | Tubal rank |
| $\|\cdot\|_{\mathrm{F}}$ | Frobenius norm |
| $\|\cdot\|_{\mathrm{tsp}}$ | Tensor (t)-spectral norm |
| $\|\cdot\|_\star$ | Tubal nuclear norm |
| $\lambda_0, \lambda_k, \tilde{\lambda}$ | Regularization parameters |
| $\mathcal{R}(\cdot)$ | Regularization function |
| $\mathcal{L}(\cdot)$ | Loss function |
| $\tilde{\mathcal{L}}(\cdot)$ | Modified loss function |
| $\alpha_k, \beta_k$ | Constants related to restricted strong convexity |
| $\gamma_k, \tau_k$ | Constants related to restricted smoothness |

Table 12 – Continued from previous page

| Symbol/Notation | Description |
| --- | --- |
| $u_n, v_n$ | Terms related to the convergence rate |
| **A** | Event defined for parameter selection |
| $\mathbb{W}(r, \mathbf{h})$ | Parameter space |
| $O(\cdot), o(\cdot)$ | Big O and little o notations |
| $\lesssim, \gtrsim, \asymp$ | Asymptotic comparisons |
| $\mathbb{P}(\cdot)$ | Probability measure |
| $\mathbb{E}[\cdot]$ | Expectation |
| *w.h.p.* | With high probability |

