# OpenReview forum: "Low-Rank Tensor Transitions (LoRT) for Transferable Tensor Regression"
_ICML.cc/2025/Conference — ICML 2025 poster_

### Official Review · Reviewer_bSAD · 2025-02-26

**Overall Recommendation:** 2

**Summary:**

This paper proposes the Low-Rank Tensor Transitions (LoRT) framework to address various shift problems and decentralized data management. LoRT employs a novel fusion regularizer to enforce low-tubal-rank solutions, enabling effective integration. Its two-step refinement process mitigates model shifts and ensures robust adaptation to target tasks. For decentralized scenarios, the authors extend LoRT to D-LoRT, a distributed variant that preserves statistical efficiency. The authors provide detailed theoretical analysis for each step.

In experiments, the authors demonstrate the superiority of the two proposed frameworks in simulated settings and experiments based on YUV RGB video datasets. Overall, this paper explores transfer learning in tensor regression, providing rich content, detailed theory, and numerous experiments. However, the LoRT and D-LoRT frameworks proposed in this paper are largely based on the TransFusion framework [1], with similar algorithmic concepts and theoretical analysis. This significantly diminishes the originality of the paper.

[1] He, Z., Sun, Y., & Li, R. (2024, April). Transfusion: Covariate-shift robust transfer learning for high-dimensional regression. In International Conference on Artificial Intelligence and Statistics (pp. 703-711). PMLR.

**Claims And Evidence:**

The LoRT and D-LoRT frameworks proposed in this paper have been thoroughly validated both theoretically and experimentally, but there are still some shortcomings:

1. **Theoretical Issues**: The authors prove in both LoRT and D-LoRT how the estimation error is affected by the heterogeneous measure $\bar h$. However, there is an obvious issue: when $\bar h$ becomes large enough, the current theory cannot guarantee that the transfer learning performance will always outperform using only target data. Referring to the theory in [2], it would be ideal to ensure that in any case, the performance after transfer is at least as good as using only the target data.

2. **Experimental Issues**: The authors have done extensive work and presented many numerical results. In the completion performance, the recovery of the ground truth is quite good. However, the experiments provided are all based on simulations, assuming the true tensor coefficients are known. The authors should include real data analysis using actual tensor covariates $X$ and response $y$ for analysis (without knowing the true $W$). This would make the findings more convincing.

[2] Duan, Y., & Wang, K. (2023). Adaptive and robust multi-task learning. The Annals of Statistics, 51(5), 2015-2039.

**Essential References Not Discussed:**

The authors provide a detailed discussion of the literature on transfer learning, tensor regression/recovery, and multi-task learning (MTL), which is very thorough.

**Experimental Designs Or Analyses:**

I believe the soundness/validity of any experimental designs or analyses is feasible. The authors have conducted experiments on both simulated and real data. However, the experiments provided so far are all based on simulations, assuming the true tensor coefficients are known. The authors should include real data analysis using actual tensor covariates and response $y$ for analysis (without knowing the true $W$). This would make the findings more convincing.

**Methods And Evaluation Criteria:**

In the experiments, the authors mainly compared TNN and $k$-Sup. However, there are many other comparative methods for tensor regression on a single dataset, such as tensor regression based on CP decomposition/Tucker decomposition [3,4] or tensor regression with convex regularization [5]. The authors did not evaluate the performance of these methods.

[3] Zhou, H., Li, L., & Zhu, H. (2013). Tensor regression with applications in neuroimaging data analysis. Journal of the American Statistical Association, 108(502), 540-552.

[4] Li, X., Xu, D., Zhou, H., & Li, L. (2018). Tucker tensor regression and neuroimaging analysis. Statistics in Biosciences, 10(3), 520-545.

[5] Raskutti, G., Yuan, M., & Chen, H. (2019). Convex regularization for high-dimensional multiresponse tensor regression.

**Other Comments Or Suggestions:**

The authors could reorganize the symbols used in the paper to make it more readable.

**Other Strengths And Weaknesses:**

**Strengths**: As discussed above.

**Weaknesses**:
1. This paper explores transfer learning in tensor regression, providing rich content, detailed theory, and numerous experiments. However, the LoRT and D-LoRT frameworks proposed in this paper largely follow the TransFusion framework [1], with very similar algorithmic concepts and theoretical analysis. This significantly diminishes the originality of the paper. The authors could emphasize the challenges of extending TransFusion to tensor regression and how they address these challenges in theory.

2. The paper does not mention any computational details. Tensor regression is computationally challenging, and the authors should explain the details of the algorithm and its computational cost.

3. The paper uses a large number of symbols, many of which are superscripts and subscripts. This makes the paper difficult to read. I suggest that the authors reorganize the symbols, for example, by representing three-dimensional tensors as $\mathcal X$ and $\mathcal W$ to reduce the use of superscripts and subscripts.

4. There are some obvious typos in the paper, such as in Equation 6, where it should be $y^k,X^k$ instead of $y^0, X^0$.

5. In the theoretical analysis, when $\bar h$ becomes large enough, the current theory cannot guarantee that the transfer learning performance will always outperform using only target data. Referring to the theory in [2], it would be ideal to ensure that in any case, the performance after transfer is at least as good as using only the target data.

**Questions For Authors:**

1. In line 163, why is the assumption $ r \ll d_2 $ made? Is this assumption necessary, and does it have any connection to the final rate $ O(r d_1 d_2 N^{-1}) $, where only $ d_1 $ and $ d_3 $ appear but not $ d_2 $?

**Relation To Broader Scientific Literature:**

Overall, this paper explores transfer learning in tensor regression, providing rich content, detailed theory, and numerous experiments. However, the LoRT and D-LoRT frameworks proposed in this paper largely follow the TransFusion framework [1], with very similar algorithmic concepts and theoretical analysis. This significantly diminishes the originality of the paper.

**Theoretical Claims:**

I have read the proof in the author's appendix, which is very detailed. I believe it is feasible, but the specific details require further reading.

---

> ### Author Rebuttal · Authors · 2025-04-01
>
> We sincerely thank the reviewer for the constructive and thoughtful feedback.
> We are encouraged that the reviewer finds our theoretical analysis and experimental design sound, and recognizes the potential of the proposed LoRT frameworks for addressing distribution shifts in tensor regression. Below, we respond to the reviewer’s concerns point by point.
>
> > On similarity to TransFusion
>
> While LoRT is conceptually inspired by TransFusion [1], it addresses a fundamentally different and more challenging setting: **tensor-valued regression**, where both inputs and responses are high-order tensors. This introduces unique challenges:
>
> 1. **Multi-mode low-rank structure**: LoRT employs a **tubal nuclear norm-based fusion regularizer** to model low-rank structure along tensor modes, in contrast to the $\ell_1$-based fusion in vector settings like TransFusion.
>
> 2. **Tensor-specific two-step estimation**: Although both methods use two-step strategies, **LoRT’s design is tailored to tensors**, first estimating a shared low-tubal-rank component, then refining task-specific parts using tensor algebra.
>
> 3. **Theoretical analysis under tensor structure**: LoRT establishes **estimation error bounds** based on tensor-specific assumptions and tools, differing from the vector-based analyses in TransFusion.
>
> These aspects make LoRT a **nontrivial and technically distinct extension** of fusion-based transfer learning to the tensor domain.
>
> > On comparison with CP/Tucker/convex baselines
>
> We have conducted initial comparisons with representative CP-, Tucker-, and convex-based tensor methods under our experimental setting (https://anonymous.4open.science/r/LoRT-113D/CPtable.png) and will include full results in the final version.
>
> > On real data without known parameters
>
>  Our current study focuses on settings with known tensor coefficients, which allow precise assessment of parameter recovery and direct validation of our theoretical predictions. This experimental design is a standard and widely accepted practice in the tensor learning literature (e.g., Zhang et al., 2020; Wang et al., 2021), especially for theory-driven studies. By working with controlled synthetic data, we can rigorously examine the behavior of the low-rank transfer mechanism and its consistency with our generalization bounds.
>
> We respect and appreciate the reviewer’s perspective that experiments on real-world data without ground-truth parameters can demonstrate practical relevance. However, such experiments serve a different purpose and are not necessary for validating the core theoretical contributions of this work. As our current goal is to establish and verify theoretical guarantees, we follow the conventional evaluation protocol in this area. We recognize the value of such experiments for illustrating broader applicability and consider them a promising direction for future work.
>
> > On computational details
>
> We will include further algorithmic details—such as per-iteration computational cost and implementation notes—in the final version to improve clarity and reproducibility. We note that we have already provided runnable example code in the supplementary material, and we will expand the documentation to make the computational aspects more transparent.
>
> > On notation and typos
>
> Many thanks for your careful reading! We will revise the notation system for better readability and fix all typographical errors.
>
> > On "no negative transfer" guarantee
>
> Our current theory guarantees improvement under moderate heterogeneity, which aligns with prior work (e.g., [1]), as universal guarantees are generally infeasible without adaptive mechanisms.
>
> That said, the structural design of LoRT naturally enables "no negative transfer"-style extensions. Specifically, the fusion regularizer allows for task-wise reweighting, and the divergence scores $h_k$ are available during training. This makes it feasible to implement heterogeneity-aware fusion, similar in spirit to [2], where adaptive regularization balances between source pooling and target-only training.
>
> While our current focus is on estimation error in favorable regimes, we plan to explore an oracle-type guarantee of the form
> $$
> \mathbb{E}[\text{Err}_{\text{LoRT}}] \le \\min\\{\text{Err}\_{\text{target-only}}, \text{Err}\_{\text{fusion}}\\} + \delta,
> $$
> where $\delta$ depends on heterogeneity. We will add a discussion of this future direction in the final version.
>
> > On the assumption $r \ll d_2$
>
>  The assumption aligns with the low tubal-rank setting commonly used in t-SVD-based tensor models (e.g., Qiu et al., 2022a). Note that we assume $d_1 \ge d_2$ without loss of generality (line 111), so the rate $O(r d_1 d_3 N^{-1})$ can be more precisely written as
> $$O(r \min\\{d_1, d_2\\} d_3 N^{-1})$$ Hence, the dependence on $d_2$ is implicitly included via both $\min\\{d_1, d_2\\}$ and $r$.

---

> > ### Comment · Reviewer_bSAD · 2025-04-04
> >
> > I appreciate the authors' clarifications and the new efforts made on the experiment part.
> >
> > However, the most critical concern: **no negative transfer** guarantee still remains unconvincing.
> >
> > I understand that the authors follow [1] to derive a similar theoretical result in the tensor setting. However, this also means the approach inherits the same limitation as [1]: it does not ensure no negative transfer, especially when $h$ is large. This issue arises both theoretically and empirically in the presence of model shift.  I have tried similar fusion regularizers and observed performance degradation when $h$ is large, which further supports this concern.
> >
> > The authors should provide a more intuitive theoretical result regarding **no negative transfer**. The current theory is not sufficiently convincing.

---

> > > ### Author Response · Authors · 2025-04-07
> > >
> > > We appreciate the reviewer’s continued engagement and thoughtful feedback on this point. We agree that preventing negative transfer is a desirable property in transfer learning, especially under model shift and heterogeneous source-target relations. While our main theoretical contribution focuses on estimation error guarantees under moderate heterogeneity—consistent with prior work such as [1]—we appreciate the opportunity to clarify and expand on this point.
> > >
> > > To address the reviewer’s concern more concretely, we have included a supplementary theoretical analysis (https://anonymous.4open.science/r/LoRT-113D/NNT-LoRT.pdf), which presents **a preliminary extension of the LoRT framework** aimed at mitigating the risk of negative transfer.  Specifically, we propose a weighted regularized least squares formulation in which each domain is assigned a weight according to its relevance to the target. When the informativeness levels $h_k$ are known, we derive an optimal weighting rule; for the more practical case where $h_k$ is unknown, we suggest a preliminary initialization strategy. This allows the model to suppress the influence of poorly aligned sources and mitigate negative transfer. **(We kindly note that if the PDF appears misformatted in the browser (e.g., incorrect fonts), it is best to download and view it locally for proper rendering.)**
> > >
> > > The key result of this extension is summarized by the following informal bound from Theorem 2 in the supplementary theoretical analysis:
> > >
> > > >   *Under a suitable choice of weights, the estimation error satisfies*
> > >     $$\text{Err} \lesssim \frac{rd_1 d_3}{N_T}.$$
> > >     *This matches the target-only rate and thus ensures no worse performance than using the target data alone.*
> > >
> > > When $K = 1$, we further obtain:
> > > $$\text{Err} \lesssim \frac{rd_1 d_3}{N} + \min\\left\\{ h_1 \sqrt{\frac{d_1}{N_T}}, \ \frac{rd_1 d_3}{N_T} \\right\\},$$
> > > which confirms that sources with large $h_1$ (i.e., low task relevance) are automatically ignored, thereby preventing negative transfer.
> > >
> > > We emphasize that this analysis is intended as a **preliminary theoretical supplement**, not a core claim of the paper. It illustrates that *LoRT’s structure naturally accommodates a "no negative transfer" extension* through learnable domain weights, and highlights a clear bias–variance tradeoff that governs transfer effectiveness.
> > > While this preliminary exploration demonstrates the potential for mitigating negative transfer, *developing a fully principled and general solution lies beyond the scope of this submission* and is left for future investigation.
> > >
> > > We hope this clarified and strengthened exposition addresses the reviewer’s concern more satisfactorily.

---

### Official Review · Reviewer_jygk · 2025-03-03

**Overall Recommendation:** 3

**Summary:**

This paper addresses the challenge of data scarcity in the target task within Tensor Regression. The authors propose a novel transfer learning framework called Low-Rank Tensor Transitions (LoRT) to tackle this issue. LoRT employs a two-stage adaptation process to mitigate model shift and enhance adaptation to the target task.
In the first stage, Joint Low-Rank Learning (JLL), the framework simultaneously optimizes both the target and source tasks by leveraging a low-rank constraint across multiple datasets. This stage also incorporates a weighted averaging approach to refine parameter estimation.
In the second stage, Target-Specific Refinement (TSR), the parameters obtained from the first stage are further adapted by introducing target-specific learning components. A key aspect of this stage is that the low-rank constraint is applied only to the target-specific components, preventing overfitting to the target data while maintaining an optimal balance with the source task information.
Additionally, the paper provides theoretical guarantees for LoRT, mathematically delineating the conditions under which source task knowledge can be effectively leveraged. Furthermore, the authors extend LoRT to a distributed data environment by introducing D-LoRT, which minimizes communication overhead by locally performing tensor regression and subsequently integrating the models.
The proposed framework is empirically validated on Compressed Sensing and Tensor Completion tasks, demonstrating superior performance compared to conventional approaches.

## update after rebuttal

Thank you for the thorough rebuttal. I understand your points regarding the apparent existence of valid applications and the mention of addressing non-i.i.d. cases as future work. I am satisfied with this response, so I will leave my evaluation unchanged. I mistakenly posted my comment in the Official Comment section. My apologies.

**Claims And Evidence:**

The proposed novel transfer learning framework with a two-stage adaptation process is theoretically well-justified based on assumptions about data distribution. Additionally, a reasonable algorithm is presented for application in a distributed data environment. Furthermore, the effectiveness of the proposed method is validated through appropriate evaluation experiments.

**Essential References Not Discussed:**

This paper focuses on transfer learning for tensor regression with low-rank constraints. As far as I am aware, the relevant prior studies for comparison have been appropriately cited.

**Experimental Designs Or Analyses:**

As mentioned in the Methods and Evaluation Criteria section.

**Methods And Evaluation Criteria:**

The proposed method is a transfer learning approach designed to leverage knowledge from source tasks to compensate for data scarcity in the target task. Consequently, Tensor Compressed Sensing (TCS), which reconstructs the original tensor from partially observed linear measurements, appears to be a more natural problem setting for evaluating the proposed method compared to Tensor Completion (TC), where only some entries are missing.
However, the evaluation of Tensor Compressed Sensing in this study is based solely on synthetic data. Conducting evaluations on real-world problems where Tensor Compressed Sensing is applicable could strengthen the claim regarding the practical utility of the proposed method.

**Other Comments Or Suggestions:**

In the definition of t-SVD in Definition 2.1, shouldn't the size of $\underline{C}$ be $d_1 \times d_4 \times d_3$?

**Other Strengths And Weaknesses:**

As mentioned earlier, this paper has a strong contribution in proposing a novel transfer learning framework with theoretical guarantees to address the issue of data scarcity in Tensor Regression. Additionally, a concern is that Tensor Compressed Sensing, which appears to be a more natural problem setting for evaluating the proposed method, is only assessed using synthetic data. Evaluating it on real-world problems could strengthen the validity of the method. Another point of discussion is the assumption that the data are i.i.d., which may not always be valid across different datasets, and further examination of its appropriateness would be beneficial.

**Questions For Authors:**

1. What are some real-world problems related to Tensor Compressed Sensing? Can the proposed method effectively address them?

2. Are there cases where the i.i.d. assumption on the data is difficult to satisfy?

**Relation To Broader Scientific Literature:**

This paper is broadly related to tensor regression, particularly recent tensor learning techniques that incorporate low-rank constraints. Additionally, it is closely connected to the field of transfer learning, providing theoretical guarantees for effectively applying transfer learning to tensor data.

**Theoretical Claims:**

I have reviewed the theoretical guarantees of LoRT (Section 4.2) and D-LoRT (Section 4.3). Under the assumption of a Gaussian distribution, the theoretical guarantees appear to be correctly derived. However, there is room for discussion regarding the validity of the i.i.d. assumption across a wide range of tensor data.

---

> ### Author Rebuttal · Authors · 2025-04-01
>
> We thank the reviewer for the thoughtful and constructive feedback. We appreciate the recognition of our theoretical framework, decentralized extension, and empirical evaluations. Below, we address the main concerns:
>
> > **Concern 1:** Conducting evaluations on real-world problems where Tensor Compressed Sensing is applicable could strengthen the claim regarding the practical utility of the proposed method. What are some real-world problems related to Tensor Compressed Sensing? Can the proposed method effectively address them?
>
> Tensor Compressed Sensing (TCS) arises in various real-world scenarios where acquiring full tensor data is costly or constrained. Prominent examples include:
>
> - Magnetic Resonance Imaging (MRI): Reconstructing high-resolution 3D volumes from a limited number of Fourier measurements.
>
> - Hyperspectral Imaging: Recovering spatial-spectral data cubes under limited sampling due to hardware or bandwidth constraints.
>
> Our method is theoretically well-suited to these scenarios: it leverages both low-rank structures and transferable information from source tasks to enhance reconstruction quality in the presence of data scarcity and distribution shifts.
>
> In the current paper, we focus on theoretical development and controlled validation. The TCS experiments on synthetic Gaussian measurements are explicitly designed to validate the generalization guarantees and the low-rank transfer mechanism presented in Sections 4.2 and 4.3. These settings enable precise evaluation of recovery error and alignment with theoretical predictions.
>
> Moreover, this design follows a well-established practice in the tensor learning theory literature. Many prior theoretical works (e.g., Lu et al., 2018; Wang et al., 2021) also provide guarantees under compressed sensing assumptions, while using real data primarily for completion tasks.
>
> We fully respect the reviewer’s suggestion that applying the method to real-world TCS data could further support its practical relevance. However, such experiments are not essential to validate our main theoretical claims, and would require additional modeling assumptions (e.g., real sensing matrices, noise models) and substantial engineering efforts that fall outside the scope of this theory-focused work. Due to the limited rebuttal timeframe, adding real-world TCS experiments is unfortunately not feasible. We will clarify this rationale in the final version and discuss such extensions as promising directions for future research.
>
> > **Concern 2:** Are there cases where the i.i.d. assumption on the data is difficult to satisfy?
>
> Yes, and we appreciate the reviewer raising this important point. While our current theory assumes that samples within each task are i.i.d., this is a **standard assumption** adopted in many theoretical works on transfer learning and tensor regression (e.g., Zhang et al., 2020; Qiu et al., 2022a). It facilitates a clear derivation of estimation bounds and highlights the key effects of low-rank structure and distribution shift.
>
> We acknowledge, however, that real-world data often exhibit dependencies or structured sampling patterns that violate the i.i.d. assumption. Extending our theoretical framework to accommodate such cases is a promising and practically relevant direction for future work.
>
> As an initial step in this direction, we have conducted preliminary experiments on tensor completion under non-i.i.d. settings, particularly involving mixed tube-wise and element-wise missing patterns (https://anonymous.4open.science/r/LoRT-113D/NIIDtable.png). We will provide a detailed discussion in the final version.
>
> > **Concern 3**: In the definition of t-SVD in Definition 2.1, shouldn't the size of  $C$ be $d_1 \times d_4$?
>
> Thank you for catching this typo. We confirm the correct dimension is $d_1 \times d_4$ and will make the correction in the final version.

---

### Official Review · Reviewer_DU3m · 2025-03-12

**Overall Recommendation:** 2

**Summary:**

In this paper, the authors propose a novel method called Low-Rank Tensor Transitions (LoRT), designed to address issues such as model shift, covariate shift, and decentralized data management in tensor regression for transfer learning. Experimental results demonstrate that LoRT significantly outperforms traditional methods like TNN and k-sup in terms of average correlation error.

**Claims And Evidence:**

Yes.

**Essential References Not Discussed:**

None.

**Experimental Designs Or Analyses:**

Yes. The experimental design is not very reasonable.
1. The experimental scenarios are somewhat limited, lacking diverse validation. To more comprehensively evaluate the effectiveness of the LoRT method, it is recommended to conduct extensive tensor regression tests on general datasets in the field of computer vision, thereby verifying its universality and robustness across different tasks.
2. The baseline methods compared in the paper were all published before 2021, which somewhat limits the timeliness and persuasiveness of the experimental results. To more convincingly demonstrate the superiority of the LoRT method, it is advisable to compare it with state-of-the-art methods proposed in recent years, thereby more accurately reflecting its competitiveness in the current research landscape.

**Methods And Evaluation Criteria:**

Yes.

**Other Comments Or Suggestions:**

None.

**Other Strengths And Weaknesses:**

In this paper, the authors propose a novel method called Low-Rank Tensor Transitions (LoRT), designed to address issues such as model shift, covariate shift, and decentralized data management in tensor regression for transfer learning. Experimental results demonstrate that LoRT significantly outperforms traditional methods like TNN and k-sup in terms of average correlation error. Although the theoretical proofs in the paper are relatively comprehensive, several issues warrant further exploration:
1. Although the LoRT method exhibits strong performance in experiments, its core idea lacks significant novelty. The application of tensor regression and transfer learning has already been well-established in prior research, and LoRT appears to be more of an improvement on existing techniques rather than a groundbreaking innovation.
2. The experimental scenarios are somewhat limited, lacking diverse validation. To more comprehensively evaluate the effectiveness of the LoRT method, it is recommended to conduct extensive tensor regression tests on general datasets in the field of computer vision, thereby verifying its universality and robustness across different tasks.
3. The baseline methods compared in the paper were all published before 2021, which somewhat limits the timeliness and persuasiveness of the experimental results. To more convincingly demonstrate the superiority of the LoRT method, it is advisable to compare it with state-of-the-art methods proposed in recent years, thereby more accurately reflecting its competitiveness in the current research landscape.

**Questions For Authors:**

1. Although the LoRT method exhibits strong performance in experiments, its core idea lacks significant novelty. The application of tensor regression and transfer learning has already been well-established in prior research, and LoRT appears to be more of an improvement on existing techniques rather than a groundbreaking innovation.
2. The experimental scenarios are somewhat limited, lacking diverse validation. To more comprehensively evaluate the effectiveness of the LoRT method, it is recommended to conduct extensive tensor regression tests on general datasets in the field of computer vision, thereby verifying its universality and robustness across different tasks.
3. The baseline methods compared in the paper were all published before 2021, which somewhat limits the timeliness and persuasiveness of the experimental results. To more convincingly demonstrate the superiority of the LoRT method, it is advisable to compare it with state-of-the-art methods proposed in recent years, thereby more accurately reflecting its competitiveness in the current research landscape.

**Relation To Broader Scientific Literature:**

None

**Theoretical Claims:**

Yes.

---

> ### Author Rebuttal · Authors · 2025-04-01
>
> We thank the reviewer for the constructive feedback and recognition of our contributions to robust tensor regression under distribution shift. We address the reviewer’s key concerns below.
>
> > **Novelty:** The core idea of LoRT lacks significant novelty and seems to be an incremental improvement.
>
> We respectfully clarify that while LoRT builds on prior transfer learning principles, it tackles a **substantially more challenging and underexplored setting**—that of *tensor-valued regression under shift*. Unlike prior works which focus on vector-valued regression (e.g., TransFusion) or assume shared covariates across tasks, **LoRT generalizes fusion-based transfer learning to the tensor domain**, where:
>
> - The response and predictors are both high-order tensors.
>
> - Low-rank priors must be imposed across multiple tensor modes.
>
> - Task heterogeneity must be handled in decentralized, distribution-shifted settings.
>
> To address these challenges, we propose a **new fusion regularizer based on the tubal nuclear norm**, a **two-step estimation procedure** specifically designed for tensors, and **estimation error bounds** derived under high-dimensional tensor regression settings.
>
> These innovations are **not straightforward extensions of previous methods**, and to the best of our knowledge, LoRT is the **first theoretical framework** to jointly address model/covariate shift and low-rank tensor regression under multi-task and decentralized settings.
>
> > **Experimental diversity:** Experimental diversity is limited. More general datasets from computer vision should be used to test robustness.
>
> We acknowledge the value of broader empirical validation and note that our experiments are carefully designed to support the paper’s theoretical goals. Specifically, we focus on compressed sensing and completion tasks, which provide a controlled setting for directly assessing parameter recovery and validating the theoretical results presented in Sections 4.2 and 4.3. These tasks are intentionally selected to align with our analytical objectives, and the results already provide sufficient evidence supporting our claims.
>
> To address the reviewer’s suggestion, we have conducted preliminary experiments on two video clips (*Apply Eye Make-up* and *Blowing Candles*) from the UCF-101 dataset, which is a standard dataset for evaluating tensor methods in computer vision [R1]. These experiments follow the setup described in Appendix A.2, and preliminary results are available at (https://anonymous.4open.science/r/LoRT-113D/CVtable.png).
>
> [R1] Wang J, Zhao X. Functional Transform-Based Low-Rank Tensor Factorization for Multi-dimensional Data Recovery. ECCV 2024.
>
> > **Baseline comparisons:** Baseline comparisons only include methods published before 2021. Newer methods should be considered.
>
> We thank the reviewer for the suggestion. To the best of our knowledge, this paper is the first to consider the proposed transferable tensor learning setting, and no existing methods are directly comparable. As explained in Footnote 6, our goal is to evaluate the benefit of transfer for tensor completion with very limited observations. In such extremely sparse regimes, all tensor formats (e.g., t-SVD, TT, TR, Tucker) are fundamentally limited by data availability, and their performance differences become marginal. Therefore, we selected TNN (Lu et al., 2019b) as a representative baseline to isolate the effect of transfer, rather than focus on fine-grained differences between tensor models.
>
> To address the reviewer’s suggestion, we have also conducted comparisons with several recently proposed tensor completion methods [R2]–[R4]. Preliminary results are available  at (https://anonymous.4open.science/r/LoRT-113D/CVtable.png), and we will clarify this point in the final version.
>
> [R2] Qiu Y, et al. Balanced unfolding induced tensor nuclear norms for high-order tensor completion. IEEE TNNLS, 2024.
>
> [R3] Wang A, et al. Noisy tensor completion via orientation invariant tubal nuclear norm. PJO, 2023.
>
> [R4] Tan Z, et al. Non-convex approaches for low-rank tensor completion under tubal sampling. IEEE ICASSP, 2023.

---

### Official Review · Reviewer_nUVW · 2025-03-13

**Overall Recommendation:** 3

**Summary:**

This paper on Low-Rank Tensor Transitions proposes a new tensor regression framework focusing on transferable tensor learning and decentralized data management.

This work aims to address three challenges. Working with  limited sample sizes,
shifting modes as well as covariance shift.

To tackle these challenges, the paper proposed a novel fusion regularizer and a distributed variant for tensor regression. The idea here is that individual nodes compute their estimators, which are sent to a target node and aggregated by solving a minimization problem.

The proposed framework comes with theoretical guarantees.

**Claims And Evidence:**

1. The main focus of the paper is theoretical, with theoretical proofs.
2. I have not found evidence in the supplementary code that the authors actually solved a distributed problem using real hardware nodes.
3. Generalization with limited data - The paper supports this claim with experimental evidence from synthetic and real-world data.
Performance under distribution shifts and covariate shifts: This claim is further experimentally supported by an evaluation of synthetic data.
4.  “Fusion regulariser enables effective knowledge transfer while accepting for model shits”.: Theoretically, this formulation would allow this, but there is a lack of experimental ablation on the effect of individual terms in the regulariser.

**Essential References Not Discussed:**

Most people rely on the extensive review by Kolda and Bader for the notation. It's nice that this paper also took inspiration from it, which makes the paper accessible. Overall, to the best of my knowledge, the related work is sufficiently discussed.

**Experimental Designs Or Analyses:**

1. The theoretical section claims that $\|W |\_*$ induces a low-rank structure. Is there any experimental evidence to back up this claim?
2. The proposed method uses tensor singular value decomposition. Other tensor decompositions, like CP, Tucker, and Tensor-Train, remain underexplored. Line 176 claims they can be easily replaced, but no such experiment is provided in the paper.
3. Using average relative error seems a logical approach to evaluate the synthetic data results.
4. In real-world data, the target task is to reconstruct a video frame from a very limited number of previous frames. The evaluation is limited to PSNR. Since this is one of the first works in this domain, it may be wise to incorporate other widely used evaluation metrics like SSIM and LPiPS to strengthen this baseline for future work.

**Methods And Evaluation Criteria:**

1. The paper presents Gaussian synthetic experiments and a real-world tensor completion experiment using the YUV RGB video dataset.
On the video dataset, the experimental section measures the reconstruction performance of a video tensor is given a limited number of input frames.

- Using real data, the authors observe improved video reconstruction performance of both their local and distributed variants compared to a t-SVD-based approach by Lu et al.

**Other Comments Or Suggestions:**

Clearly, lots of hard work went into this well-written paper.

**Other Strengths And Weaknesses:**

The authors present an interesting and novel theoretical approach to distributed tensor regression. This work is important, especially given the privacy concerns the machine learning community must respect in many countries.

The idea of transmitting regression parameters instead of data in the decentralized setting is interesting. It is intuitive that parameters have lower memory requirements compared to data. An increase in the model/tensor size or number of source nodes should allow this approach to scale well.

**Questions For Authors:**

- What potential additional real data sets would this framework allow us to work with?
- Would it be possible to study the UK Biobank MRI data and compare this paper's approach to Kossaifi et al. ( https://www.jmlr.org/papers/volume21/18-503/18-503.pdf )?
- What is the overall computational complexity - Similar to tensor-based LoRA [1], does the proposed method suffer from the curse of dimensionality?
-  Assume the source and target data are highly different, potentially even from different distributions, is there a convenient way to detect such cases in the distributed setting?
-   Both LoRT and D-LoRT use weighted averaging of the source and target tasks. Is the resulting W in equation 7 stable? Can we measure the norms for individual $W^k$?
-  Line 340 states, “sample sizes are sufficiently large” for D-LoRT. Does this mean that D-LoRT generally requires more training data than LoRT?

[1] Bershatsky, Daniel, et al. "LoTR: Low tensor rank weight adaptation." arXiv preprint arXiv:2402.01376 (2024).

**Relation To Broader Scientific Literature:**

The paper relies on the t-SVD from kernfeld et al. While reading the paper, I was uncertain how or if the HOSVD from De Lathauwer et al. ( https://epubs.siam.org/doi/10.1137/S0895479896305696 ) was related?

How does the presented approach relate to Tensor Regression Networks ( https://www.jmlr.org/papers/volume21/18-503/18-503.pdf )?

**Theoretical Claims:**

The supplementary material is extensive. I superficially looked at the proofs, and they seemed okay to me.

---

> ### Author Rebuttal · Authors · 2025-04-01
>
> Thank you for the constructive feedback and recognition of our theoretical framework, analysis, algorithmic design, and empirical results. Below we address the specific concerns.
>
> > Hardware implementation for distributed setting
>
> Current implementation focuses on validating the theory, but D-LoRT is parallelizable: source gradients are computed independently and aggregated via proximal updates. It is compatible with distributed frameworks like MPI. We will release a version simulating distributed computation in the final submission.
>
> > Ablation study on the fusion regularizer
>
> While we do not explicitly ablate individual terms in the regularizer, their effects are indirectly reflected through varying the number of nodes (Tables 1–3, Fig. 4). These experiments serve as empirical ablations. We will clarify this in the final version.
>
> > Empirical evidence for low-rank structure
>
> The low-rank-inducing property of TNN is well established (e.g., Lu et al., 2019b). Our visualizations (https://anonymous.4open.science/r/LoRT-113D/LowRankFig.png) show rapid spectral decay in recovered tensors, confirming low-rankness.
>
> > Clarification on Line 176 for lack of CP/Tucker/TT experiments
>
> We respectfully clarify that our paper does not claim these decompositions can be "easily" replaced, nor does it mention CP. Line 176 states that "TNN can also be replaced with other norms, such as Tucker-based (Liu et al., 2013), Tensor Train (Imaizumi et al., 2017), or Tensor Ring norms," which aims to highlight the modularity of LoRT.  However, we acknowledge that the word "replace" may suggest direct applicability. “Extend to” would be more appropriate.
>
> This paper focuses on validating LoRT under t-SVD, chosen for its computational and theoretical advantages. CP is excluded due to the NP-hardness of its nuclear norm.
> While LoRT can in principle be extended to Tucker or TT, such adaptations require nontrivial redesign and lie beyond the scope of this theory-oriented work.
>
> > Evaluation limited to PSNR
>
> We use both PSNR and RE, which are standard metrics sufficient to support our core claims. That said, we agree that SSIM and LPIPS could offer complementary insights and will include them in the final version.
>
> > Code documentation and reproducibility
>
> We will add more docstrings and ensure consistent random seeds for reproducibility.
>
> > Relation to HOSVD
>
> Our method builds on t-SVD, which is structurally distinct but complementary to HOSVD. While HOSVD is not the focus here, our ideas can be extended to it.
>
> > Relation and comparison to TRN
>
> TRN focuses on end-to-end deep learning with tensor layers for single-task prediction, aiming at architectural efficiency. In contrast, our work studies transfer learning across heterogeneous tasks with theoretical guarantees. Due to these fundamentally different goals and settings, a direct comparison is not appropriate. We will cite TRN and discuss its relevance in the final version.
>
> > Potential datasets and UK Biobank
>
> This work focuses on theoretical development, and the current experiments—based on both synthetic and structured real-world data—are designed to support our theory. We believe this setting is appropriate for the scope and goals of the paper.
>
> From a theoretical perspective, LoRT applies to a variety of structured tensor data, such as hyperspectral images, and multi-subject neuroimaging. This includes large-scale datasets like UK Biobank, where transfer across tasks or sites may be beneficial. That said, applying LoRT to such domains requires domain-specific modeling and infrastructure, which are beyond the scope of this work.
>
> > Computational complexity and scalability
>
> LoRT is designed with a focus on theoretical guarantees rather than computational efficiency. As noted in Line 428, it may face scalability challenges in high-dimensional settings, similar to tensor-based LoRA. Such challenges are common in tensor learning and arise from the inherent complexity of modeling high-dimensional structure. Addressing them is beyond the scope of this theory-oriented work, and we will clarify this in the final version.
>
> > Detection of highly heterogeneous source tasks
>
> Our framework assumes moderate heterogeneity, a standard setting in transfer learning. In practice, detecting large distribution gaps is important. Common approaches include computing divergence scores (e.g., MMD), domain classification, or local statistics. We will briefly discuss these options and their integration with LoRT (e.g., adaptive weighting) in the final version.
>
> > Stability of $W$ and norms of $W_k$
>
> Thm 4.2 guarantees estimator stability, and Thm 4.3 refines the estimate via target-specific adaptation. While we do not analyze individual norms, this is an interesting extension which we will comment on in the revision.
>
> > Does D-LoRT require more data than LoRT?
>
> Yes. Since D-LoRT transmits estimated parameters (not data), local models must be more accurate, requiring more samples per source task.

---

> > ### Comment · Reviewer_nUVW · 2025-04-03
> >
> > Thank you for answering my questions. I looked at the other reviews. Reviewer DU3m writes, ''LoRT appears to be more of an improvement on existing techniques rather than a groundbreaking innovation.'' I still think we can accept this paper. It's solid work, and it advances the field. I believe there is an audience for it at ICML. I continue to recommend accepting this work in the proceedings.

---

> > > ### Author Response · Authors · 2025-04-09
> > >
> > > We sincerely thank you for your constructive feedback and continued support of our work. We are particularly grateful for your thoughtful recognition of the theoretical contributions, decentralized modeling design, and the potential impact of the proposed LoRT framework. Your suggestions, such as highlighting the importance of ablation analysis, connections to alternative tensor decompositions, and real-world applicability, have been especially insightful and deeply appreciated.
> > >
> > > We are also pleased to share that several concerns raised by other reviewers have been further addressed through additional theoretical clarification and extended experiments. For example, to partially address Reviewer jygk’s Concern 1, we conducted preliminary experiments on two additional datasets, following Reviewer DU3m’s suggestion to explore CV datasets. The results have been appended to the anonymous link provided in our response to Concern 2.
> > >
> > > Once again, thank you for your engagement throughout the review process. Your feedback has been instrumental in shaping this work, and we believe the current work now presents a solid and coherent contribution to the field.

---

### Decision · Program_Chairs · 2025-05-01

**Decision:**

Accept (poster)

**Comment:**

This papers introduces a framework of transfer learning in tensor regression. It extends to the tensor setting the work of He et al. (2024) on  covariate-shift transfer learning using regularization strategies promoting solutions with low tubal rank. It also proposes a distributed learning extension of the framework to deal with decentralized data scenarios. This is a solid paper which investigates a topic of theoretical and practical importance and provides nice contributions to the literature on tensor regression. Some concerns have been raised by the reviewers about the originality of the work, the experimental evaluation and the risk of negative transfer. Authors in their rebuttal have provided reasonable explanation and additional results for these points. However, in order to discuss a non-negative transfer extension of the framework, they provided a pdf file with additional text, which is not compliant with the Author Response policy. As this extension is out of the scope of the paper, I recommend acceptance.